# Scalable Quasi-Bayesian Inference for Instrumental Variable Regression

**Ziyu Wang**[1,*], **Yuhao Zhou**[1,*], **Tongzheng Ren**[2], **Jun Zhu**[1,‡]
[1] Dept. of Comp. Sci. and Tech., BNRist Center, State Key Lab for Intell. Tech. & Sys.,
Institute for AI, Tsinghua-Bosch Joint Center for ML, Tsinghua University
[2] Department of Computer Science, UT Austin
`{wzy196,yuhaoz.cs}@gmail.com, tongzheng@utexas.edu, dcszj@tsinghua.edu.cn`

## Abstract

Recent years have witnessed an upsurge of interest in employing flexible machine learning models for instrumental variable (IV) regression, but the development of uncertainty quantification methodology is still lacking. In this work we present a scalable quasi-Bayesian procedure for IV regression, building upon the recently developed kernelized IV models. Contrary to Bayesian modeling for IV, our approach does not require additional assumptions on the data generating process, and leads to a scalable approximate inference algorithm with time cost comparable to the corresponding point estimation methods. Our algorithm can be further extended to work with neural network models. We analyze the theoretical properties of the proposed quasi-posterior, and demonstrate through empirical evaluation the competitive performance of our method.

## 1 Introduction

Instrumental variable (IV) regression is a standard approach for estimating causal effect from confounded observational data. In the presence of confounding, any regression method estimating $\mathbb{E}(\mathbf{y} \mid \mathbf{x})$ cannot recover the causal relation $f^\dagger$ between the outcome $\mathbf{y}$ and the treatment $\mathbf{x}$, since the residual $\mathbf{u} = \mathbf{y} - f^\dagger(\mathbf{x})$ is correlated with $\mathbf{x}$ due to the unobserved confounders. IV regression enables identification of the causal effect through the introduction of *instruments*, variables $\mathbf{z}$ that are known to influence $\mathbf{y}$ only through $\mathbf{x}$.

IV regression is widely used in economics [1], epidemiology [2] and clinical research [3], but modeling nonlinear effect in IV regression can be challenging. Recent years have seen great development in adopting modern machine learning models for IV regression [4–8]. However, there is still a lack of uncertainty quantification measures for these flexible IV models. Uncertainty quantification is especially important for IV analysis, since unlike in standard supervised learning scenarios, we do not have (unconfounded) validation data, from which we could deduce the error pattern of the estimated model. Moreover, the instrument of choice may be weak, meaning that it only provides limited information for $\mathbf{x}$; in such cases point estimators suffer from high variance [9]. This problem is exacerbated in the nonparametric setting, where IV estimation is usually an ill-posed inverse problem, where instruments provide vanishing information for the higher-order nonlinear effects [10].

The IV setting brings unique challenges for uncertainty quantification. For example, while it is natural to consider a Bayesian approach, specification of the likelihood requires knowledge of the entire data generating process, which is typically unavailable in IV regression. Consequently, most, if not all, existing work on Bayesian IV [11–15] assumes the following data generating process:

---

[*]ZW and YZ contribute equally. [‡]JZ is the corresponding author. An extended version is available at `https://arxiv.org/abs/2106.08750v2`.

35th Conference on Neural Information Processing Systems (NeurIPS 2021).

$\mathbf{x} = g(\mathbf{z}) + \mathbf{u}_1$, $\mathbf{y} = f(\mathbf{x}) + \mathbf{u}_2$, where $(\mathbf{u}_1, \mathbf{u}_2)$ are correlated and independent of $\mathbf{z}$. These methods then conduct posterior inference on $g, f$ as well as the *generative model* for the unobserved confounders $(\mathbf{u}_1, \mathbf{u}_2)$. However, the additive error in $\mathbf{x}$ is an unnecessary assumption for most point estimation procedures, and is difficult to check in high dimensions. Furthermore, the need to model the generating process of $(\mathbf{u}_1, \mathbf{u}_2)$ introduces an extra risk of model misspecification, and Bayesian inference on the generative model is computationally expensive, especially on complex high-dimensional datasets. None of these issues present if only point estimation is needed.

For the above reasons, it is appealing to turn to an alternative *quasi-Bayesian* approach [16–18]. Quasi-Bayesian analysis views IV estimation as a generalized method-of-moments (GMM) procedure, and defines the quasi-posterior as a Gibbs distribution constructed from a chosen prior and violation of the moment constraints. It does not require full knowledge of the data generating process, and thus does not suffer from the aforementioned drawbacks of Bayesian inference. However, computation of the quasi-posterior is non-trivial, as its density contains a conditional expectation term $\mathbb{E}(\mathbf{y} - f(\mathbf{x}) \mid \mathbf{z})$, which itself needs to be estimated from data. This makes both approximate inference and theoretical analysis difficult. So far, quasi-Bayesian analysis for nonparametric IV is only developed on classical models such as wavelet basis [17, 18] or Nadaraya-Watson smoothing [19], while adoption of flexible machine learning models such as kernel machines or neural networks remains an open challenge. Moreover, numerical study has been limited in previous work, so little is known about the empirical performance of the quasi-Bayesian approach.

In this work, we present a novel quasi-Bayesian procedure for IV regression, building upon the recent development in kernelized IV models [7, 20]. We employ a Gaussian process (GP) prior and construct a quasi-likelihood using a kernel conditional expectation estimator. We establish theoretical properties of the resultant quasi-posterior, proving its consistency and showing that it may quantify instrument strength. Furthermore, inspired from the minimax formulation of IV estimation [5, 8, 20, 21], we design a principled approximate inference algorithm using random feature expansion and a novel adaptation of the "randomized prior trick" [22]. The algorithm has the form of stochastic gradient descent-ascent and is thus scalable and easy to implement. It can also be adapted to work with flexible neural network models, in which case its behavior can be formally justified by analyzing the neural networks in the kernel regime. Empirical evaluation shows that the proposed method produces informative uncertainty estimates, scales to high-dimensional and complex nonlinear problems, and is especially advantageous when the instrument is weak.

The rest of the paper is organized as follows: In Section 2 we set up the problem. We then derive the quasi-posterior and analyze its theoretical properties in Section 3, and present the approximate inference algorithm in Section 4. Section 5 reviews related work, and Section 6 presents numerical studies. Finally, we discuss conclusion and future work in Section 7.

## 2 Notations and Setup

**Notations** We use boldface $(\mathbf{x}, \mathbf{y}, \mathbf{z})$ to represent random variables on the space $\mathcal{X} \times \mathcal{Y} \times \mathcal{Z}$, regular font $(x, y, z)$ to denote deterministic values. $[n]$ denotes the set $\{1, 2, \cdots, n\}$. $\{(x_i, y_i, z_i) : i \in [n]\}$ indicates the training data. We use the notations $X := (x_1, \ldots, x_n) \in \mathcal{X}^n$, $f(X) := (f(x_1), \ldots, f(x_n))$; likewise for $Y, Z$. For finite-dimensional vectors $\theta, \theta' \in \mathbb{R}^m$, we use $\|\theta\|_2, \langle \theta, \theta' \rangle_2$ to denote the Euclidean norm and inner product, respectively. For any operator $A : H_1 \to H_2$ between Hilbert spaces $H_1$ and $H_2$, we denote its adjoint by $A^* : H_2 \to H_1$. When $H_1 = H_2$ and $\lambda \in \mathbb{R}$, we use the notation $A_\lambda := A + \lambda I$ for simplicity.

**IV regression** Denote the treatment and response variables as $\mathbf{x}, \mathbf{y}$, the instrument as $\mathbf{z}$, and the true *structural function* of interest as $f^\dagger$. Consider the data generating process $\mathbf{y} = f^\dagger(\mathbf{x}) + \mathbf{u}$, where the unobserved $\mathbf{u}$ satisfies $\mathbb{E}(\mathbf{u} \mid \mathbf{z}) = 0$, but may correlate with $\mathbf{x}$. Then $f^\dagger$ satisfies

$$\mathbb{E}(\mathbf{y} - f^\dagger(\mathbf{x}) \mid \mathbf{z}) = 0 \text{ a.s. } [P(dz)], \tag{1}$$

where $P$ denotes the data distribution. This *conditional moment restriction* (CMR) formulation is the standard definition in literature [e.g., 23, 18], and is used in the recent work on machine learning models for IV. It connects to GMM as (1) can be viewed as a continuum of generalized moment constraints. Note that (1) does not place any structural constraint on the conditional distribution $p(\mathbf{x} \mid \mathbf{z})$, such as additive noise; hence, it does not require full knowledge of the data generating

process. Also, as discussed in, e.g., Hartford et al. [4], the setup can also be extended to incorporate observed confounders $\mathbf{v}$, by including $\mathbf{v}$ in both $\mathbf{x}$ and $\mathbf{z}$.

**Kernelized IV and a dual view**  Let $\mathcal{H}, \mathcal{I}$ be suitably chosen function spaces on $\mathcal{X}, \mathcal{Z}$, respectively. The generalized moment constraint (1) motivates the use of the following objective

$$\min_{f \in \mathcal{H}} \quad \mathcal{L}(f) := d_n^2(\hat{E}_n f - \hat{b}) + \bar{\lambda}\Omega(f), \tag{2}$$

where $\hat{E}_n : \mathcal{H} \to \mathcal{I}$ is an empirical approximation to the conditional expectation operator $E : f \mapsto \mathbb{E}(f(\mathbf{x}) \mid \mathbf{z} = \cdot)$, $\hat{b}$ is an estimator of $\mathbb{E}(\mathbf{y} \mid \mathbf{z} = \cdot)$, $\{d_n\}$ is a sequence of suitable (semi-)norm on $\mathcal{I}$, and $\Omega : \mathcal{H} \to \mathbb{R}$ is a regularization term.

When $\mathcal{H}, \mathcal{I}$ are reproducing kernel Hilbert spaces (RKHS) with corresponding kernels $k_x, k_z$, it is natural to set $\Omega(f) = \frac{1}{2}\|f\|_{\mathcal{H}}^2$, and define $\hat{b}$ as the result of kernel ridge regression on $\mathbf{y}$ with respect to $\mathbf{z}$. In this case $\hat{E}_n$ can be defined with the empirical *kernel conditional expectation* operator: let $C_{zx} = \mathbb{E}(k(\mathbf{x}, \cdot) \otimes k(\mathbf{z}, \cdot)), C_{zz} = \mathbb{E}(k(\mathbf{z}, \cdot) \otimes k(\mathbf{z}, \cdot))$. Assuming $E$ maps all $f \in \mathcal{H}$ to $Ef \in \mathcal{I}$, we have [24]

$$C_{zz}Ef = C_{zx}f, \quad \forall f \in \mathcal{H}.$$

This motivates the use of $\hat{E}_n := \hat{C}_{zz,\bar{\nu}}^{-1}\hat{C}_{zx}$ where $\hat{C}_{zz} := \frac{1}{n}\sum_{i=1}^n k(z_i, \cdot) \otimes k(z_i, \cdot)$, $\hat{C}_{zx}$ is defined similarly, and $\bar{\nu}$ is a regularization hyperparameter.[2] The choice of $d_n$ is flexible; the dual IV formulation uses $d_n^2(g) := \frac{1}{2n}\sum_{j=1}^n g(z_j)^2 + \frac{\bar{\nu}}{2}\|g\|_{\mathcal{I}}^2 = \frac{1}{2}\langle g, \hat{C}_{zz,\bar{\nu}}g\rangle_{\mathcal{I}}$. Introducing the evaluation operator $S_z : \mathcal{I} \to \mathbb{R}^n, S_z g := (g(z_1), \ldots, g(z_n))$, we have $\hat{b} = \hat{C}_{zz,\bar{\nu}}^{-1}\frac{S_z^* Y}{n}$, and

$$\mathcal{L}(f) = \frac{1}{2}\left\|\hat{C}_{zz,\bar{\nu}}^{-1/2}\left(\hat{C}_{zx}f - \frac{S_z^* Y}{n}\right)\right\|_{\mathcal{I}}^2 + \frac{\bar{\lambda}}{2}\|f\|_{\mathcal{H}}^2 \tag{3}$$

$$= \max_{g \in \mathcal{I}} \frac{1}{n}\sum_{j=1}^n \left((f(x_j) - y_j)g(z_j) - \frac{g(z_j)^2}{2}\right) - \frac{\bar{\nu}}{2}\|g\|_{\mathcal{I}}^2 + \frac{\bar{\lambda}}{2}\|f\|_{\mathcal{H}}^2, \tag{4}$$

where (4) holds because of the Fenchel duality $\frac{1}{2}u^2 = \sup_{v \in \mathbb{R}}\left(uv - \frac{1}{2}v^2\right)$ and the equality $\mathbb{E}(u(\mathbf{x}, \mathbf{y})g(\mathbf{z})) = \mathbb{E}(\mathbb{E}(u(\mathbf{x}, \mathbf{y}) \mid \mathbf{z})g(\mathbf{z}))$; see [25, 8, 21, 26]. The dual formulation (4) circumvents the need to compute $\hat{E}_n$ directly, an operator between the typically infinite-dimensional spaces $\mathcal{H}$ and $\mathcal{I}$, and leads to a scalable estimation procedure based on stochastic gradient descent-ascent (SGDA). It can also be generalized to work with deep neural networks (DNNs) instead of kernel machines, by replacing the RKHS regularizer with a suitable regularizer for DNNs, although theoretical analysis for the resulted algorithm requires separate effort [8, 26]. As we shall see, (4) will also enable the construction of a scalable approximate inference algorithm.

## 3   Quasi-Bayesian Analysis of Dual IV

Introduce the notations $S_x : \mathcal{H} \to \mathbb{R}^n, S_x f := (f(x_1), \ldots, f(x_n))$, so that $\hat{C}_{zx} = \frac{1}{n}S_z^* S_x, \hat{C}_{zz} = \frac{1}{n}S_z^* S_z$. Define $\lambda := n\bar{\lambda}, \nu = n\bar{\nu}$. Now we can re-express (3) in an equivalent form as:

$$\bar{\mathcal{L}}(f) := \frac{n}{\lambda}\mathcal{L}(f) = \frac{1}{2}(f(X) - Y)^\top (\lambda^{-1}L)(f(X) - Y) + \frac{1}{2}\|f\|_{\mathcal{H}}^2, \tag{5}$$

where $L := \frac{1}{n}S_z \hat{C}_{zz,\bar{\nu}}^{-1} S_z^*$ is a linear map from $\mathbb{R}^n$ to $\mathbb{R}^n$, and thus can be identified with an $n \times n$ matrix. Since the first term above is equivalent to the log density of the multivariate normal distribution $\mathcal{N}(Y \mid f(X), \lambda L^{-1})$, we can view (5) as the objective of a kernel ridge regression problem, which has a data-dependent noise covariance $\lambda L^{-1}$.[3] The connection between kernel ridge regression and Gaussian process regression [28] thus motivates the use of the *quasi-posterior*

---

[2]With an abuse of notation, $k$ refers to the reproducing kernel of the corresponding RKHS ($k_x$ for $\mathcal{H}$ or $k_z$ for $\mathcal{I}$) whenever the denotation is clear.

[3]Here we assume the invertibility of $L$ for brevity. Alternatively, observe that (5) defines a linear inverse problem with the finite-dimensional observation operator $f \mapsto \sqrt{L}f(X)$ and noise variance $\lambda^{-1}I$, and we can follow [27, Chapter 6] to derive the same quasi-posterior.

$\Pi(df \mid \mathcal{D}^{(n)})$, defined through the following Radon-Nikodym derivative w.r.t. the standard GP prior $\Pi = \mathcal{GP}(0, k_x)$:

$$\frac{d\Pi(\cdot \mid \mathcal{D}^{(n)})}{d\Pi}(f) \propto \exp\left(-\frac{1}{2}(f(X) - Y)^\top(\lambda^{-1}L)(f(X) - Y)\right) = \exp\left(-\frac{n}{\lambda}d_n^2(\hat{E}_n f - \hat{b})\right). \quad (6)$$

Note that contrary to standard Bayesian modeling, we do not assume $Y \sim \mathcal{N}(f(X), \lambda L^{-1})$ is part of the true data generating process, nor does the theoretical analysis below rely on it. Instead, the quasi-posterior (6) should be interpreted as a Gibbs distribution which trades off between the properly scaled *evidence* $\lambda^{-1}nd_n^2(\hat{E}_n f - \hat{b})$, which characterizes the estimated violation of the GMM constraint (1), and our *prior belief* $\Pi(df)$. This trade-off is most clear from the well-known variational characterization of the Gibbs distribution [29],

$$\Pi(\cdot \mid \mathcal{D}^{(n)}) = \arg\min_{\Psi} \mathbb{E}_{f \sim \Psi}[\lambda^{-1}nd_n^2(\hat{E}_n f - \hat{b})] + \mathrm{KL}(\Psi \parallel \Pi). \quad (7)$$

Nonetheless, the fictitious data generating process $f \sim \mathcal{GP}(0, k), Y \sim \mathcal{N}(f(X), \lambda L^{-1})$ is useful for deriving the quasi-posterior, since its conditional distribution $p_{\mathrm{fic}}(df \mid Y)$ coincides with (6) [27, Chapter 6]. In the probability space of this fictitious data generating process, for any finite set of test inputs $x_*$, we have

$$p_{\mathrm{fic}}(f(x_*), Y) \sim \mathcal{N}\left(0, \begin{bmatrix} K_{**} & K_{*x} \\ K_{x*} & K_{xx} + \lambda L^{-1} \end{bmatrix}\right),$$

where $K_{(\cdot)}$ denote the corresponding Gram matrices with subscript $*$ denoting $x_*$ and $x$ denoting $X$ (so, e.g., $K_{*x} := k(x_*, X)$). Thus by the Gaussian conditioning formula, we have

$$\Pi(f(x_*) \mid \mathcal{D}^{(n)}) = p_{\mathrm{fic}}(f(x_*) \mid Y) = \mathcal{N}(m, S), \quad (8)$$

$$\text{where} \quad m := K_{*x}(\lambda I + LK_{xx})^{-1}LY, \quad (9)$$

$$S := K_{**} - K_{*x}L(\lambda I + K_{xx}L)^{-1}K_{x*}, \quad (10)$$

$$L = K_{zz}(K_{zz} + \nu I)^{-1}. \quad (11)$$

In the above $K_{zz} := k(Z, Z)$ denotes the Gram matrix, and (11) follows from the definition $L = n^{-1}S_z(n^{-1}S_z^*S_z + \bar{\nu}I)^{-1}S_z^*$, the Woodbury identity, and the observation that $S_zS_z^* = K_{zz}$.

**Theoretical Analysis** For the quasi-posterior $\Pi(\cdot \mid \mathcal{D}^{(n)})$ to be a useful measure of uncertainty, it needs to satisfy the following informal criteria: as $n \to \infty$, we expect

(C1) $\Pi(\cdot \mid \mathcal{D}^{(n)})$ will exclude incorrect solutions;

(C2) In cases of non-identification, $\Pi(\cdot \mid \mathcal{D}^{(n)})$ will not exclude any valid solution in the model.

In the following, we formalize these criteria, and demonstrate that with appropriately chosen hyper-parameters, the quasi-posterior satisfies both criteria. We will work with the following assumptions:

**Assumption 3.1.** *The restriction of the conditional expectation operator $f \mapsto \mathbb{E}(f(\mathbf{x}) \mid \mathbf{z} = \cdot)$ on $\mathcal{H}$, denoted as $E$, has its image contained in $\mathcal{I}$. $E : \mathcal{H} \to \mathcal{I}$ is bounded.*

Assumption 3.1 is intuitive, and is also slightly more general than some previous work [20, 26] that require the conditional expectation operator to be bounded on the entire hypothesis space, which typically corresponds to the "sample space" of the GP prior in our setting, and is much larger than $\mathcal{H}$ (see Appendix A). Nonetheless, we note that Hypothesis 4 in Singh et al. [7] may be more general, although they impose extra smoothness assumptions on $E$.

**Assumption 3.2.** *The true structural function $f^\dagger(x)$ is such that, there exists a sequence $\{\tau_m : m \in \mathbb{N}\}$ satisfying $\tau_m \to 0$, and*

$$m\tau_m^2 \geq \inf_{f_m^\dagger \in \mathcal{H}:\|f^\dagger - f_m^\dagger\| \leq \tau_m} \|f_m^\dagger\|_{\mathcal{H}}^2 - \log \Pi(\{f : \|f\| \leq \tau_m\}), \quad (12)$$

*where $\|\cdot\|$ denotes the sup norm.*

Assumption 3.2 requires that $f^\dagger$ can be well approximated in $\mathcal{H}$; this is more general than previous work [e.g., 20, 7] that require $f^\dagger \in \mathcal{H}$, but is typical in the GP literature [30, 31]. The sequence $\{\tau_m\}$ is determined by the complexity of $\mathcal{H}$ and its ability for approximating $f^\dagger$, and is usually the optimal posterior contraction rate for Gaussian process regression on the unconfounded dataset $\{(x_i, f^\dagger(x_i) + \epsilon_i) : i \in [m]\}$ [31].

**Assumption 3.3.** *$\mathcal{X}$ and $\mathcal{Z}$ are Polish spaces. The kernels $k_x, k_z$ are continuous, $\sup_{x \in \mathcal{X}} k(x,x) + \sup_{z \in \mathcal{Z}} k(z,z) \leq \kappa^2$, and Mercer's representations [32] of $k_x$ and $k_z$ exist. The random variable $\mathbf{y} - f^\dagger(\mathbf{x})$ is sub-exponential.*

Assumption 3.3 imposes technical conditions frequently assumed in literature [e.g., 33]. The requirements on the kernels can be satisfied by e.g. continuous kernels on compact subsets of $\mathbb{R}^d$.

Given the assumptions above, we characterize (C1) by showing that as $n \to \infty$, the posterior places vanishing mass on the region of functions that violates the GMM constraints (1). Concretely,

**Theorem 3.1** (Proof in Appendix B). *There exist a constant $M > 0$ depending on the data distribution and the kernels of choice, and a sufficiently slowly growing sequence $\{\gamma_n\} \to \infty$ (e.g., we can always have $\gamma_n \leq \log \log n$) such that when taking $\bar\lambda = n^{-1/2}, \bar\nu = \min\{\tau^2_{\lceil \sqrt{n} \rceil}, n^{-1/2}\gamma_n\}$, it has*

$$\mathbb{E}_{\mathcal{D}^{(n)}} \Pi(\{f : \mathbb{E}^2(f(\mathbf{x}) - \mathbf{y} \mid \mathbf{z}) > M\tau^2_{\lceil \sqrt{n} \rceil}\} \mid \mathcal{D}^{(n)}) \to 0. \tag{13}$$

*Remark* 3.1. The above result should primarily be interpreted as a consistency result, although it also provides a posterior contraction rate [34] in the order of $\tau_{\lceil \sqrt{n} \rceil}$ in the semi-norm $f \mapsto \|Ef\|_{L_2(P(dz))}$, where $\{\tau_m\}$ is defined in Assumption 3.2. As an example, suppose the regularity of $f^\dagger$ is similar to the Matérn-1/2 RKHS; then for suitable kernels we have $\tau_{\lceil \sqrt{n} \rceil} = O(n^{-1/8})$, see Remark A.3.

The rate is suboptimal, and can be immediately improved if we impose further regularity assumption on $k_z$: if we assume the critical radius of the local Rademacher complexity [35] of the unit-norm ball $\mathcal{I}_1$ is $\delta_n \asymp \tau_n \asymp n^{-\frac{b}{2(b+1)}}$, a rate of $O(n^{-\frac{b-1}{2(b+1)}})$ for the semi-norm can be established following similar arguments. This is still worse than [26], which provides the rate $O(\max\{\tau_n, \delta_n\})$ for what corresponds to our posterior mean estimator. Nonetheless, their result is also generally suboptimal, because the choice of $\delta_n \asymp \tau_n$ is actually suboptimal when the IV regression problem is ill-posed. To date, minimax optimal rates have only been established under additional assumptions on the relations between the conditional expectation operator and the model for $f$. Our extended version establishes such results, for both this semi-norm and more intuitive norms such as $L_2(P(dx))$.

*Remark* 3.2. The use of an increasing $\lambda = n\bar\lambda$ is a technical artifact due to our minimal assumptions on $k_z$. It is common to impose extra regularization in nonparametric IV, due to the need to estimate $E$ from data [19], and, in the quasi-Bayesian setting, also due to the additional technical challenges [18]. However, this is not necessary for our method if additional assumptions are imposed, as we show in the extended version. In practice, our hyperparameter selection procedure does not produce $\lambda$ with significant growth.

The following proposition characterizes (C2):

**Proposition 3.1** (Proof in Appendix B). *The scaled log quasi-likelihood estimate $\ell_n(f) = d_n^2(\hat{E}_n f - \hat{b})$ satisfies*

$$\Pi\left(\left\{\forall f : \forall \delta > 0, \lim_{n \to \infty} \mathbb{P}_{\mathcal{D}^{(n)}}(|\ell_n(f) - \mathbb{E}_{\mathbf{z} \sim P(z)}(\mathbb{E}(f(\mathbf{x}) - \mathbf{y} \mid \mathbf{z})^2)| > \delta) = 0\right\}\right) = 1.$$

In words, for $\Pi$-almost every $f$, the log quasi-likelihood estimate scaled by $n^{-1}$ converges to $\mathbb{E}_{\mathbf{z} \sim P(z)}(\mathbb{E}(f(\mathbf{x}) - \mathbf{y} \mid \mathbf{z})^2)$, which characterizes the violation of (1). While the restriction to the probability-1 subset can be concerning in the nonparametric setting [34], the proof only depends on $f$ satisfying similar approximability conditions to $f^\dagger$; see (45).

*Remark* 3.3. To understand why the proposition characterizes (C2), suppose there are multiple $f$ that satisfy (1). Then all of them will eventually have significantly higher log likelihood than functions violating (1): the difference is $\Theta(\lambda^{-1}n)$. If these functions have a similar level of regularity in the sense that their *concentration functions*, which is the right-hand side of (12), have the same asymptotics as $\epsilon \to 0$, Borell's inequality [e.g., 34, Proposition 11.17] implies that the posterior mass of small $\|\cdot\|$-norm balls around them will also have the same asymptotics.

Finally, we provide the following (over-)simplified example, which compares the behavior of the quasi-posterior and bootstrap in the context of (C2):

*Example* 3.1. Suppose $\mathbf{z}$ is completely non-informative so that $\mathbb{E}(f(\mathbf{x}) \mid \mathbf{z}) \equiv \mathbb{E}f(\mathbf{x})$ for all $f$; and suppose the estimated conditional expectation $\hat{E}_n$ is sufficiently accurate, so that we replace it

with $E$.[4] In this case bootstrap on (3) will always return the point estimator $f \equiv 0$ due to the non-zero regularization on $\|f\|_{\mathcal{H}}$, while the quasi-posterior behaves like the prior, correctly reflecting the complete lack of evidence in data.

While this example is oversimplified, and in practice the estimation error of $E$ plays an important role, it is known that bootstrap uncertainty estimates can be problematic given weak IVs [36–38]. We also observe similar failures for bootstrap in the experiments (see Section 6.1).

## 4 Scalable Approximate Inference via a Randomized Prior Trick

We now turn to approximate inference with parametric models such as random feature expansion or wide NNs. Scalable inference for the IV quasi-posterior appears difficult, since for any $f$, computing the quasi-likelihood involves computing $\hat{E}_n f$, which in turn requires either inverting an $n \times n$ Gram matrix, judging from (5) and (11), or solving an optimization problem specific to $f$ from (4). Nonetheless, we show that it is possible, by extending the "randomized prior" trick for Gaussian process regression [22] to work with (quasi-)likelihoods with an optimization formulation as in (4).

Our algorithm works with random feature models. A random feature model for $k_z$ approximates $k_z(z, z') \approx \tilde{k}_{z,m}(z, z') := \frac{1}{m}\phi_{z,m}(z)^\top \phi_{z,m}(z')$, where $\phi_{z,m}$ takes value in $\mathbb{R}^m$. Then the map $\varphi \mapsto \frac{1}{\sqrt{m}}\varphi^\top \phi_{z,m}(\cdot) =: g(\cdot; \varphi)$ parameterizes an approximate RKHS $\tilde{\mathcal{H}}$; and for all $c > 0$, the random function $g(\cdot; \varphi)$, where $\varphi \sim \mathcal{N}(0, cI)$, is distributed as $\mathcal{GP}(0, c\tilde{k}_{z,m})$. The notations related to $k_x$ are similar and thus omitted. Now we can state the objective function:

**Proposition 4.1** (Proof in Appendix C.1)**.** *Let* $\phi_0 \sim \mathcal{N}(0, \lambda\nu^{-1}I), \theta_0 \sim \mathcal{N}(0, I), \tilde{y}_i \sim \mathcal{N}(y_i, \lambda)$. *Then the optima* $\theta^*$ *of*

$$\min_{\theta \in \mathbb{R}^m} \max_{\phi \in \mathbb{R}^m} \sum_{i=1}^n \left( (f(x_i; \theta) - \tilde{y}_i)g(z_i; \phi) - \frac{g(z_i; \phi)^2}{2} \right) - \frac{\nu}{2}\|\phi - \phi_0\|_2^2 + \frac{\lambda}{2}\|\theta - \theta_0\|_2^2 \quad (14)$$

*parameterizes a random function which follows the quasi-posterior distribution* (6)*, where the kernels are replaced by the random feature approximations.*

Given the above proposition, we can sample from the random feature-approximated quasi-posterior by solving (14) with stochastic gradient descent-ascent; the approximation errors will be analyzed in the following. The objective (14) is closely related to (4); as we show in Appendix C.1.1, it is equivalent to

$$\min_{f \in \tilde{\mathcal{H}}} \max_{g \in \tilde{\mathcal{I}}} \sum_{i=1}^n \left( (f(x_i) - \tilde{y}_i)g(z_i) - \frac{g(z_i)^2}{2} \right) - \frac{\nu}{2}\|g - g_0\|_{\tilde{\mathcal{I}}}^2 + \frac{\lambda}{2}\|f - f_0\|_{\tilde{\mathcal{H}}}^2, \quad (15)$$

which differs from (4) only in the regularizers: instead of regularizing the norm of $f$ and $g$, (15) encourages the functions to stay close to randomly sampled *anchors* [39]. Alternatively, we can view (15) as *perturbing* the point estimator (4), so that it has a covariance matching that of the quasi-posterior. A similar relation is also observed in the original randomized prior trick, which transforms GP regression to the optimization problem $\min_{f \in \tilde{\mathcal{H}}} \sum_{i=1}^n (f(x_i) - \tilde{y}_i)^2 + \lambda\|f - f_0\|_2^2$. In both cases, the resultant algorithm for approximate inference has the same time complexity as ensemble training for point estimation.

While the algorithm can be directly applied to neural network models as in [22], we follow [40] and modify the objective, to account for the difference between the neural tangent kernel (NTK) [41] of a wide neural network architecture and the NNGP kernel of the corresponding infinite-width Bayesian neural network [42–44]. Concretely, we modify (14) as

$$\min_\theta \max_\phi \sum_{i=1}^n \left( (\tilde{f}_\theta(x_i) - \tilde{y}_i)\tilde{g}_\phi(z_i) - \frac{\tilde{g}_\phi(z_i)^2}{2} \right) - \frac{\nu}{2}\|\phi - \phi_0\|_2^2 + \frac{\lambda}{2}\|\theta - \theta_0\|_2^2, \quad (16)$$

where $\tilde{g}_\phi(z) := g(z; \phi) - g(z; \phi_0) + \tilde{g}_0(z)$, $\tilde{g}_0(z) := \sqrt{\frac{\lambda}{\nu}}\langle \bar{\phi}_0, \frac{\partial g}{\partial \phi}\big|_{\phi=\phi_0}(z)\rangle$,

---

[4]This setting could be realistic in the sample splitting setup considered in [7]. Note that as long as we have finite samples for the estimation of $f$ (the "second stage"), we still need to have $\bar{\lambda} > 0$.

and $\phi_0$ denotes the initial value of $\phi$, and $\bar{\phi}_0 \sim \mathcal{N}(0, I)$ is a set of randomly initialized NN parameters independent of $\phi_0$; and $\tilde{f}_\theta$ is defined similarly.

We only give a formal justification for the modification, *under the assumption*[5] that the NNs remain in the kernel regime throughout training, so that $g(z; \phi) - g(z; \phi_0) = \langle \phi - \phi_0, \frac{\partial g(z)}{\partial \phi}|_{\phi_0} \rangle_2$ [47]. Thus for the purpose of analyzing $g(\cdot; \phi) - g(\cdot; \phi_0)$, we can view $g$ as a random feature model with the parameterization $\phi \mapsto \langle \phi, \frac{\partial g(z)}{\partial \phi}|_{\phi_0} \rangle_2$. Thus by the argument in Appendix C.1.1, we can show that the weight regularizer $\|\phi - \phi_0\|_2$ is equivalent to $\|g(\cdot; \phi) - g(\cdot; \phi_0)\|_{\tilde{\mathcal{I}}} = \|\tilde{g}_\phi - \tilde{g}_0\|_{\tilde{\mathcal{I}}}$, where $\tilde{\mathcal{I}}$ is determined by the NTK $k_{g,ntk}(z, z') := \langle \frac{\partial g(z)}{\partial \phi}|_{\phi_0}, \frac{\partial g(z')}{\partial \phi}|_{\phi_0} \rangle_2$. Similar arguments can be made for $\tilde{f}_\theta$ and $\tilde{f}_0$. Consequently, (16) is equivalent to an instance of (15) with $\tilde{\mathcal{H}}, \tilde{\mathcal{I}}$ defined by the NTKs.

Implementation details for the algorithm, including hyperparameter selection, are discussed in Appendix D.

**Convergence analysis** We now complete the analysis of the inference algorithm, by showing that for any fixed set of test points $x_*$, SGDA can approximate the marginal distribution $\Pi(f(x_*) \mid \mathcal{D}^{(n)})$ arbitrarily well given a sufficient computational budget. This implies that the approximate posterior is good enough for prediction purposes.

We place several mild assumptions on the random feature model, listed in Appendix C.2; they are satisfied by common approximations such as the random Fourier features [48]. The SGDA algorithm is described in detail in Appendix C.4. Under this setup, we have

**Proposition 4.2** (Proof in Appendix C.5). *Fix $\mathcal{D}^{(n)}$ and $\lambda, \nu > 0$. Then there exist a sequence of choices of $m$ and SGDA step-size schemes, such that for any $l \in \mathbb{N}$, we have*

$$\sup_{x^* \in \mathcal{X}^l} \max(\|\hat{\mu}_m - \mu_m\|_2, \|\hat{S}_m - S_m\|_F) \xrightarrow{p} 0.$$

*In the above, $\hat{\mu}_m, \hat{S}_m$ denote the mean and covariance of the approximate marginal posterior for $f(x_*)$, $\mu, S$ correspond to the true posterior, $\|\cdot\|_F$ denotes the Frobenius norm, and the convergence in probability is defined with respect to the sampling of random feature basis.*

## 5   Related Work

Quasi-Bayesian analysis for GMM estimation problems was first developed in [49, 50, 16], which provided theoretical analysis for parametric models. The use of the quasi-posterior is motivated from the maximum entropy principle, based on which similar ideas have been developed in the machine learning literature [51–53]. Alternatively, it can also be obtained as a non-informative limit of certain Bayesian posteriors [54]. [17, 18] first analyzed the use of nonparametric models for quasi-Bayesian IV; no numerical study was presented. Closer to our work is [19] which constructed a quasi-posterior using Nadaraya-Watson smoothing as $\hat{E}_n$. While similar in form to our method, the smoothing approach relies on stronger assumptions, especially for high-dimensional data; see [7, Appendix A.2.1] which compared the corresponding point estimators. Their approach is also harder to scale to large datasets.

Our quasi-Bayesian procedure builds upon the kernelized IV models [7, 20] and the dual formulation of IV regression [6, 20, 21, 26]. The formulation (3)-(4) is from [20, 26]. [7] proposes a similar method. Although it was motivated differently as a kernelized two-stage least squares (2SLS) estimator, its objective is asymptotically equivalent to (3); however, the slight difference in regularization prevents the use of estimation procedures similar to (4), as we discuss in Appendix C.1.4. An alternative kernel-based estimator is proposed in [55, 56]; in particular, [55] also includes a Gaussian process construction, but for the different purpose of computing a leave-one-out validation statistics. Other recent work on ML models for IV include [4, 57, 58]. It remains interesting future work to develop scalable quasi-Bayesian procedures for these methods, although the mean estimator derived from our quasi-posterior implementation has competitive performance.

[59] studies semi-parametric IV estimation using an exponentially tilted empirical likelihood (ETEL) estimator. ETEL also connect to Bayesian methods. However, similar to the quasi-Bayesian

---

[5]The same linearization assumption has been employed in [40, 45]. We refer readers to [46, 21] for analysis of the linearization error in various settings similar to ours.

approach, the need to estimate the conditional moment restrictions complicates the understanding of its frequentist behavior, as well as the design of scalable inference algorithms; in their case, it is especially unclear if the empirical likelihood can be computed in sublinear time. Other approaches for uncertainty quantification include Bayesian inference and bootstrap. We have discussed previous works on Bayesian IV and their limitations in Section 1. Bootstrap is typically justified in the asymptotic regime, which does not cover many scenarios where uncertainty is most needed; this is different from the quasi-Bayesian approach which can always be justified through (7). Moreover, standard bootstrap inference on 2SLS is known to be unreliable when the instrument strength is weak [36–38]; while remedies exist (e.g., [38]), they heavily rely on the linearity and additive noise assumptions, and are thus difficult to generalize to the nonlinear setting we are interested in. As the kernelized IV methods generalize 2SLS, we expect similar issues to exist in our setting.

## 6    Experiments

Code to reproduce the experiments is available at `https://github.com/meta-inf/qbdiv`.

### 6.1    1D Simulation

We first experiment on a variety of 1D synthetic datasets, constructed by modifying the setup in [5] to incorporate a nonlinear first stage, in a way similar to [7, 60]:

$$z := \text{sigmoid}(w), \quad x := \text{sigmoid}\left(\frac{\alpha w + (1-\alpha)u'}{\sqrt{\alpha^2 + (1-\alpha)^2}}\right), \quad y_i \sim \mathcal{N}(f_0(2x-1) + 2u, 0.1),$$

where $(u, u')$ are normal random variables with unit variance and a correlation of 0.5, $w \sim \mathcal{N}(0, 1)$ is independent of $(u, u')$, $\alpha$ is a parameter controlling the instrument strength, and $f_0$ is constructed from the `sine`, `step`, `abs` or a `linear` function. We choose $N \in \{200, 1000\}$ and $\alpha \in \{0.05, 0.5\}$.

Our baselines include BayesIV [15], a state-of-the-art Bayesian model based on B-splines and Dirichlet process mixture; we also include bootstrap on 2SLS with ridge regularization, either applied directly to the input features (Linear), on their polynomial expansion (Poly), or on the same kernelized models (KIV) as ours.[6] Hyperparameter for the kernelized IV methods are selected by cross validation based on the observable first-stage and second-stage losses as in previous work [7, 20]; see Appendix D.1. For kernels we choose the RBF and Matérn kernels, although results for Matérn kernels are deferred to appendix for brevity. See Appendix D.3 for the detailed setup.

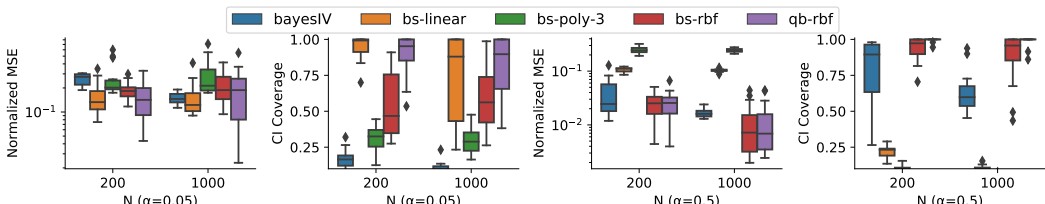

Figure 1: Test MSE and CI coverage on the `sine` dataset. The left two plots correspond to $\alpha = 0.05$, while the right two correspond to $\alpha = 0.5$. `bs` denotes bootstrap, `qb` denotes quasi-Bayesian.

Normalized MSE and coverage rate of $95\%$ credible intervals (CI) on the `sine` datasets are plotted in Figure 1. We report results on 20 independently generated datasets. As we can see, quasi-Bayesian inference provides the most reliable uncertainty estimates, and is especially advantageous in the weak IV setting ($\alpha = 0.05$). While its CI can be conservative, we note that it is still informative, and properly reflects the sample size and instrument strength; see Appendix D.4 for visualizations. These results connect to previous work showing (in a different setting) that the radius of credible set produced by a GP posterior can have the correct order of magnitude [61].

We report the average run time in Table 1. We can see that in addition to the improved flexibility, our method is also more scalable compared with the standard Bayesian baseline, as it avoids the costly modeling of the joint residual distribution.

---

[6]We do not compare with [19] since their source code is unavailable. Note that their smoothing-based method does not scale to large datasets, and there is no numerical study on the credible interval in [19].

Table 1: Average run-time (in seconds) in the 1D simulation, for a single set of hyperparameters. N/A: does not converge after 20 minutes. For the approximate inference algorithm, we report the average time cost for parallel runs on a single accelerator; see Appendix D.3 for details.

| $N$ | 1000 | 5000 | 20000 |
|---|---|---|---|
| BayesIV (CPU) | 655 | N/A | N/A |
| QB (closed-form, CPU) | 2.37 | 49.1 | 1150 |
| QB (approx. inf., GPU) | $\sim 10$ | $\sim 40$ | $\sim 140$ |

Full results on all datasets, visualizations and additional experiments are deferred to Appendix D.4. As a summary, (i) on the `abs` and `linear` datasets where all modeling assumptions hold, the results are qualitatively similar to the sin dataset. Moreover, the over-smoothed RBF kernel appears to have similar coverage comparing with the optimal kernel, and follow a similar contraction rate, as the previous work on GP regression [62] suggests. (ii) On the `step` dataset which violates Assumption 3.2, the quasi-posterior still provides more coverage than the baselines. (iii) Uncertainty estimates produced by the approximate inference algorithm are similar to that from the exact quasi-posterior.

## 6.2 Airline Demand

We now turn to the more challenging demand simulation first proposed by [4]. The dataset simulates a scenario where we need to predict the demand of airline tickets $y$, as a function of the price $x$, and two observed confounders: customer type $s$, and time of year $t$. The data generating process is

$$x := (z + 3)\psi(t) + 25 + u', \quad y := f_0(x, t, s) + u, \quad f_0(x, t, s) := 100 + (10 + x) \cdot s \cdot \psi(t) - 2x$$

where $(u, u')$ are standard normal variables with correlation $\rho$, $z \sim \mathcal{N}(0, 1)$ is independent of $(u, u')$, and $\psi$ is a nonlinear function whose shape is given in Figure 2. The variable $s$ either varies across $\{0, \ldots, 6\}$ (the lower-dimensional setting), or is observed as an MNIST image representing the corresponding digit; the latter case represents the real-world scenario where only high-dimensional surrogates of the true confounder is observed. We only report results for $\rho = 0.5$, noting that results using other choices of $\rho$ have been similar. We use $n \in \{1000, 10000\}$ for the lower-dimensional setting, and use $n = 50000$ for the image-based setting.

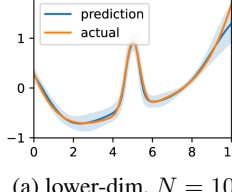 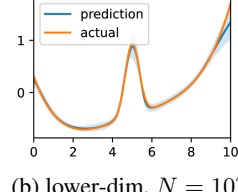 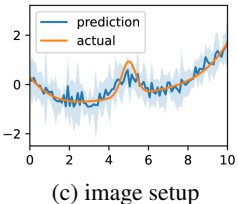

(a) lower-dim, $N = 10^3$     (b) lower-dim, $N = 10^4$     (c) image setup

Figure 2: Approximate quasi-posteriors in the demand simulation. We plot a cross-section by fixing $s, x$ to their mean values and varying $t$.

We compare our method with bootstrap on the same model, BayesIV, and bootstrap on linear or polynomial models. Performance of other point estimators on this dataset has been reported in [7, 20, 58], compared with which our method is generally competitive. We implement the dual IV model using both an RBF kernel and DNN models. See Appendix D.5 for details.

We report the test MSE and CI coverage for $N = 1000$ in Table 2, and visualize all approximate quasi-posteriors for the NN models in Figure 2. As we can see, when implemented with DNNs, our method produces uncertainty estimates with excellent coverage, which also correctly reflects the information available in the dataset: the CI is wider when $N$ is smaller, or in the high-dimensional experiment where estimation is harder. Bootstrap has a noticeably worse performance when $N = 1000$. Still, it performs well in the (arguably less interesting) large-sample setting, with a CI coverage similar to our method; see Appendix D.6. This is because on this dataset, the total instrument strength is stronger due to the presence of observed confounders, and the NN model is a good fit. Consequently, the asymptotic behavior of bootstrap can be observed when $N$ is large.

Both methods have poorer performances when we switch to the RBF kernel, although the quasi-posterior is still more reliable. We conjecture that both $\mathcal{H}$ and $\mathcal{I}$ are misspecified in this case, and

thus the credible intervals can only reflect the uncertainty within the prior model. These results show that NN models can be advantageous, which our inference algorithm supports.

The other baselines perform poorly due to their inflexibility; in particular, note that BayesIV uses additive models for both stages (e.g., $f(x, t, s) = f_1(x) + f_2(t) + f_3(s)$) which do not approximate this data generating process well. Full results and visualizations are deferred to Appendix D.6.

Table 2: Test normalized MSE, average CI coverage and CI width on the demand dataset with $N = 1000$. Results averaged over 20 trials.

| Method | BS-Linear | BS-Poly | BayesIV | BS-RBF | QB-RBF | BS-NN | QB-NN |
|---|---|---|---|---|---|---|---|
| NMSE | $.37 \pm .01$ | $.31 \pm .06$ | $.28 \pm .04$ | $.17 \pm .01$ | $.17 \pm .01$ | $.06 \pm .03$ | $.04 \pm .00$ |
| CI Cvg. | $.09 \pm .01$ | $.15 \pm .03$ | $.27 \pm .06$ | $.45 \pm .02$ | $.77 \pm .02$ | $.86 \pm .02$ | $.94 \pm .01$ |
| CI Wid. | $.09 \pm .01$ | $.16 \pm .04$ | $.08 \pm .06$ | $.18 \pm .02$ | $.37 \pm .01$ | $.14 \pm .01$ | $.26 \pm .04$ |

## 7 Conclusion

In this work we propose a scalable quasi-Bayesian procedure for IV regression. We analyze the theoretical properties of the proposed quasi-posterior, and derive a scalable algorithm for approximate inference. Empirical evaluations show that the proposed method scales to large and high-dimensional datasets, and can be particularly advantageous when the instrument strength is weak.

Beyond IV regression, formulations like (1) also appear in various other problems in causal inference and statistics, as discussed in, e.g., [21]; our method can be readily applied to these problems. Future work includes extension to more general conditional moment restriction problems with nonlinear constraints [63, 17].

## Acknowledgement

We thank the anonymous reviewers for their helpful feedback and references. Z.W., Y.Z. and J.Z. were supported by NSFC Projects (Nos. 61620106010, 62061136001, 61621136008, 62076147, U19B2034, U19A2081, U1811461), Beijing NSF Project (No. JQ19016), Beijing Academy of Artificial Intelligence (BAAI), Tsinghua-Huawei Joint Research Program, a grant from Tsinghua Institute for Guo Qiang, Tiangong Institute for Intelligent Computing, and the NVIDIA NVAIL Program with GPU/DGX Acceleration.

## Broader Impact

IV analysis is widely used in social science and clinical research, where it can be highly undesirable to have a systematic error or increased prediction variance on subpopulations which are underrepresented in the training dataset, or on which the instrument is locally weak. Uncertainty quantification is an important first step in detecting such issues, and the quasi-Bayesian approach we take can be preferable since it has a clear interpretation in a non-asymptotic setting, and when the instrument strength is weak. However, epistemic uncertainty estimates are only as good as the model allows, and it should not create a false sense of complete security. Caution must be taken when employing complex models in scenarios with potential fairness implications.

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
