# A Background on Gaussian Process Regression

We review some standard results on Gaussian process regression. They will be needed in our proof in the following, and provide more context to the results in the main text. For a thorough overview of this subject, see, for example, [34].

**Notations** The appendix uses the following additional notations: $\lesssim, \gtrsim$ represent inequality up to a universal constant, $\asymp$ denotes equivalent up to constants. $\|\cdot\|_{\mathrm{HS}}$ denotes the Hilbert-Schmidt norm.

For infinite-dimensional Gaussian process models, the prior draws almost surely fall out of the corresponding RKHS. Therefore, our posterior analysis will rely on the following result, showing that the GP prior support can be approximated with increasing accuracy using balls in the RKHS with increasing norm, in terms of a weaker norm that can be defined on the entire prior support (e.g., the sup norm).

**Theorem A.1** ([30], Theorem 2.1). *Let $W$ be a Borel measurable, zero-mean Gaussian random element in a separable Banach space $(\mathbb{B}, \|\cdot\|)$ with RKHS $(\mathcal{H}, \|\cdot\|_{\mathcal{H}})$, and let $w_0$ be contained in the closure of $\mathcal{H}$ in $\mathbb{B}$. Let $\tau_n^2 > 0$ be a number such that*

$$\phi_{w_0}(\tau_n) \leq n\tau_n^2, \tag{17}$$

*where*

$$\phi_{w_0}(\tau) = \inf_{h \in \mathcal{H}: \|h - w_0\| < \tau} \|h\|_{\mathcal{H}}^2 - \log P(\|W\| < \tau). \tag{18}$$

*Then, for any $C_\Theta > 1$ with $e^{-C_\Theta n\tau_n^2} < 1/2$, the set*

$$\Theta_n = \tau_n \mathbb{B}_1 + \underline{J}_n \mathcal{H}_1 \tag{19}$$

*is measurable and satisfy*

$$\log N(3\tau_n, \Theta_n, \|\cdot\|) \leq 6C_\Theta n\tau_n^2, \tag{20}$$

$$\mathbb{P}(W \notin \Theta_n) \leq e^{-C_\Theta n\tau_n^2}, \tag{21}$$

$$\mathbb{P}(\|W - w_0\| < 2\tau_n) \geq e^{-n\tau_n^2}. \tag{22}$$

*In the above $\mathbb{B}_1, \mathcal{H}_1$ are the unit norm balls in the corresponding spaces, and $\underline{J}_n = -2\Phi^{-1}(e^{-C_\Theta n\tau_n^2})$ where $\Phi^{-1}$ is the inverse CDF of the standard normal distribution.*

Our analysis will make use of the following:

**Corollary A.1.** *Fix any $w_0 \in \mathbb{B}$. Then for any $n \in \mathbb{N}$,*

(i). $\underline{J}_n \leq 2\sqrt{2C_\Theta n\tau_n^2} =: J_n$.

(ii). *there exists $w_n^\dagger \in \mathcal{H}$ such that $\|w_n^\dagger - w_0\| \leq \tau_n$, and*

$$\mathbb{P}(\|W - w_n^\dagger\| \leq 2\tau_n) \geq e^{-3n\tau_n^2}. \tag{23}$$

*Proof.* (i) holds because $\Phi(t) \geq 1 - e^{-t^2/2}$. for (ii), from (17)-(18) we can see that such $w_n^\dagger \in \mathcal{H}$ exists, and we can find $w_n^\dagger$ so that

$$\|w_n^\dagger\|_{\mathcal{H}} \leq 2\phi_{w_0}(\tau_n) \leq 2n\tau_n^2.$$

(23) follows from thet inequality

$$-\log P(\|W - w_n^\dagger\| \leq 2\tau_n) \overset{(a)}{\leq} \phi_{w_n^\dagger}(\tau_n) \leq \|w_n^\dagger\|_{\mathcal{H}}^2 - \log P(\|W\| < \tau_n) \overset{(b)}{\leq} 3n\tau_n^2.$$

where (a) can be found in Lemma I.28, [34]; and (b) from (17). $\square$

*Remark* A.1. For Gaussian processes the space $\mathbb{B}$ is a function space. In the analysis of our algorithms, we require that the norm $\|\cdot\|$ in the space $\mathbb{B}$ is at least equivalent to the sup norm $\|f\|_\infty := \sup_x |f(x)|$, i.e., $\|f\|_\infty \lesssim \|f\|$. This requirement is natural for most examples. For example, the space $\mathbb{B}$ is generally chosen to be the continuous function space equipped with the sup norm.

**Choices of $\tau_n$**    Choices of $\tau_n$ will affect the posterior contraction rate. In general, $\tau_n$ is determined by the ability of RKHS $\mathcal{H}$ to approximate the target function $w_0$, and the *small-ball probability* $\log P(\|W\| \leq \tau)$ which is usually determined by the metric entropy $\log N(\tau, \mathcal{H}_1, \|\cdot\|)$ [34, Lemma I.30]. For the standard Matérn and RBF kernels and the sup norm as $\|\cdot\|$, valid choices for $\tau_n$ are provided in [31], which we review below.

**Lemma A.1** (Matérn kernel. Lemma 3-4, [31])**.** *If $\mathcal{H}$ is the RKHS corresponding to the Matérn-$\alpha$ kernel in $[0,1]^d$, $w_0 \in C^\beta([0,1]^d) \cap H^\beta([0,1]^d)$,[7] the condition* (17) *will be satisfied with*

$$\tau_n^2 \asymp n^{-\frac{2\min(\alpha,\beta)}{2\alpha+d}},$$

*where the constant hidden in $\asymp$ may depend on $w_0$.*

*Remark* A.2. $\tau_n$ above usually determines the posterior contraction rate of GP regression using a normal likelihood with fixed variance [31]. For any fixed $\beta > 0$, it is minimized when we set $\alpha = \beta$. When $\alpha > d/2$, samples from the the corresponding GP belong to the space $C^{\underline{\alpha}}[0,1]^d \cap H^{\underline{\alpha}}[0,1]^d$ with probability 1, for all $\underline{\alpha} < \alpha$: see [31, pp. 2104], and [28, pp. 37-38]. Therefore, when $\alpha = \beta > \frac{d}{2}$, the above lemma applies to $w_0$ in a space that is very slightly smaller than the "sample space" of the prior. And in this case, $\tau_n$ matches the minimax rate for regression in $H^\beta([0,1]^d)$.

The practice of choosing kernel so that the GP sample space (approximately) matches the regularity of the target function is different from in kernel ridge regression, where the kernel is chosen so that the corresponding RKHS, a much smaller space than the GP sample space, matches the target regularity. Still, in all cases we can always invoke the above lemma when $w_0$ has less regularity. Although the resulted $\tau_n^2$ may be worse, it is known that using an "over-smoothed" prior does not lead to worse rates if we allow the noise variance parameter to vary with $n$ [62].

*Remark* A.3. When $w_0 = f^\dagger \in C^\beta[0,1]^d \cap H^\beta[0,1]^d$ with $\beta = \frac{d+1}{2}$, we can invoke the above theorem with $\alpha = \frac{d+1}{2}$ and obtain $\tau_n^2 \asymp n^{-1/2}$. The RKHS of the Matérn-$1/2$ kernel is norm equivalent to $H^\beta$ [28], and $C^\beta$ is often referred to as qualitatively having the same degree of regularity as $H^\beta$ (see, e.g., [31]). This is a very basic assumption for regularity, since the eigendecay of the Matérn-$1/2$ kernel is $\lambda_j \asymp j^{-\frac{d+1}{d}}$; if we further slow down the decaying rate below $j^{-1}$, $\mathcal{C}_x$ will no longer be trace-class; equivalently, $k_x$ will no longer be bounded, contradicting our Assumption 3.3.

The following lemma applies when RKHS $\mathcal{H}$ corresponding to the standard RBF kernel $k(x,x') := \exp(-\|x-x'\|^2/2)$, and $f \in A^{\gamma,r}$ which is a function space requiring exponential decrease of the Fourier transform.[8]

**Lemma A.2** (RBF kernel. Lemma 6, 9, [31])**.** *Let $f_0$ be the restriction to $[0,1]^d$ an element of $A^{\gamma,r}(\mathbb{R}^d)$. Then*

*(i). For $r > 2$ or $r = 2, \gamma \geq 4$, $f_0$ is in $\mathcal{H}$.*

*(ii). For $r \in (0,2)$, we have*

$$\inf_{h \in \mathcal{H}: \|h-f_0\| < \tau} \|h\|_{\mathcal{H}}^2 - \log P(\|W\| \leq \tau) \leq C_1 \exp\left(\frac{\left(\log \tau^{-1}\right)^{2/r}}{4\gamma^{2/r}}\right) + C_2\left(\log \tau^{-1}\right)^{1+d}.$$

*where $C_1, C_2$ only depends on $d$ and $f_0$. Consequently, for any $r \geq 1$ and $w_0 \in A^{\gamma,r}(\mathbb{R}^d)$, we have*

$$\tau_n^2 \asymp \frac{(\log n)^{2/r}}{n}.$$

*Remark* A.4. The Gaussian process using RBF kernel takes value in the space of real analytic functions, which corresponds to $A^{\gamma,r}$ with $r = 1$ [31]. Therefore, the above lemma applies to all functions in the "sample space" of the GP prior.

*Remark* A.5. Finally, note that in the sequel we will always assume that

$$n\tau_n^2 \to \infty.$$

As $\tau_n$ upper bounds the posterior contraction rate in Gaussian process regression, the above will always holds for infinite-dimensional models of interest; in general, as $\liminf n\tau_n^2 > 0$ must hold by (17), we can increase $\tau_n$ by, e.g., a logarithm factor, although for finite-dimensional models the analysis can be simplified considerably.

---

[7]$C^\beta$ denotes the Hölder space of order $\beta$, and $H^\beta$ denotes the Sobolev space of order $\beta$.

[8]The specific form is irrelevant for our purposes; see van der Vaart and van Zanten [31].

# B   Analysis of the Quasi-Posterior

In this section we prove Theorem 3.1 and Proposition 3.1. Our proof is similar to the adaptation of the posterior contraction framework [64] in [18], and involves bounding the log quasi-likelihood on certain events. However, it is different since in our case, $\hat{E}_n$ and the GP prior are not constructed with orthonormal basis in $L_2(p_{data})$. Moreover, we directly provide guarantee on the true GMM conditions (1), and do not make additional assumption on the data distribution; whereas [18] analyzed the estimated GMM conditions constructed from $\hat{E}_n$ and the empirical data distribution, and then moved to analyze $f$ under various assumptions about the joint distribution $p_{data}(dz \times dx)$, including identifiability.

We introduce the following event

$$B_n(r, L) := A_n(r) \cap \left\{ \|\hat{C}_{zx} - C_{zx}\| \leq \frac{L}{\sqrt{n}} \right\} \cap \left\{ \left\| \frac{S_z^*}{n}(f^\dagger(X) - Y) \right\|_{\mathcal{I}} \leq \frac{L}{\sqrt{n}} \right\}, \qquad (24)$$

where the event $A_n(r) := \left\{ \|C_{zz,\bar{\nu}}^{-1/2}(\hat{C}_{zz} - C_{zz})C_{zz,\bar{\nu}}^{-1/2}\| \leq r \right\}$. We will then bound the (scaled) log quasi-likelihood

$$\ell_n(f) := -\frac{2\lambda}{n} \log \frac{d\Pi(\cdot \mid \mathcal{D}^{(n)})}{d\Pi} = \left\| \hat{C}_{zz,\bar{\nu}}^{-1/2} \frac{S_z^*(f(X) - Y)}{n} \right\|_{\mathcal{I}}^2 \qquad (25)$$

in both directions on the event $B_n(r, L)$.

## B.1   Bounds on the Quasi-likelihood

**Lemma B.1.** *Conditioned on the event $B_n(r, L)$ for $r \in (0, 1/2)$, we have for all $f \in \mathcal{H}$*

$$-r_n^{(0)} + \sqrt{\frac{1-2r}{1-r}} \, \|C_{zz}^{-\frac{1}{2}} C_{zx} f\|_{\mathcal{I}} \leq \|\hat{C}_{zz,\bar{\nu}}^{-\frac{1}{2}} \hat{C}_{zx} f\|_{\mathcal{I}} \leq r_n^{(0)} + \sqrt{\frac{1}{1-r}} \, \|C_{zz}^{-\frac{1}{2}} C_{zx} f\|_{\mathcal{I}}, \qquad (26)$$

*where*

$$r_n^{(0)} \lesssim \left( \frac{L}{\sqrt{\bar{\nu} n}} + \sqrt{\bar{\nu}} \right) \|f\|_{\mathcal{H}}. \qquad (27)$$

*Proof.* On the event $A_n(r)$, we have

$$
\begin{aligned}
\|\|\hat{C}_{zz,\bar{\nu}}^{-1/2} C_{zx} f\|_{\mathcal{I}}^2 - \|C_{zz,\bar{\nu}}^{-1/2} C_{zx} f\|_{\mathcal{I}}^2| &= |\langle C_{zx} f, (C_{zz,\bar{\nu}}^{-1} - \hat{C}_{zz,\bar{\nu}}^{-1}) C_{zx} f \rangle_{\mathcal{I}}| \\
&\leq \|C_{zz,\bar{\nu}}^{1/2}(C_{zz,\bar{\nu}}^{-1} - \hat{C}_{zz,\bar{\nu}}^{-1})C_{zz,\bar{\nu}}^{1/2}\| \cdot \|C_{zz,\bar{\nu}}^{-1/2} C_{zx} f\|_{\mathcal{I}}^2 \\
&= \|I - C_{zz,\bar{\nu}}^{1/2} \hat{C}_{zz,\bar{\nu}}^{-1} C_{zz,\bar{\nu}}^{1/2}\| \cdot \|C_{zz,\bar{\nu}}^{-1/2} C_{zx} f\|_{\mathcal{I}}^2 \\
&\leq \frac{r}{1-r} \|C_{zz,\bar{\nu}}^{-1/2} C_{zx} f\|_{\mathcal{I}}^2,
\end{aligned}
$$

where the last inequality above uses (51) in Lemma B.6. Thus

$$\sqrt{\frac{1-2r}{1-r}} \|C_{zz,\bar{\nu}}^{-1/2} C_{zx} f\|_{\mathcal{I}} \leq \|\hat{C}_{zz,\bar{\nu}}^{-1/2} C_{zx} f\|_{\mathcal{I}} \leq \sqrt{\frac{1}{1-r}} \|C_{zz,\bar{\nu}}^{-1/2} C_{zx} f\|_{\mathcal{I}}.$$

Since $\|C_{zz,\bar{\nu}}^{-1/2} C_{zz}^{1/2}\| \leq 1$, the right hand side above is $\leq \sqrt{1/(1-r)}\|C_{zz}^{-1/2} C_{zx} f\|_{\mathcal{I}}$; for the left hand side, observe that

$$\|C_{zz}^{-1/2} C_{zx} f\|_{\mathcal{I}} - \|C_{zz,\bar{\nu}}^{-1/2} C_{zx} f\|_{\mathcal{I}} \leq \|C_{zz}^{1/2} - C_{zz,\bar{\nu}}^{-1/2} C_{zz}\| \|Ef\|_{\mathcal{I}}$$

where we recall $E = C_{zz}^{-1} C_{zx}$ is bounded by Assumption 3.1. To bound $\|C_{zz}^{1/2} - C_{zz,\bar{\nu}}^{-1/2} C_{zz}\|$, denote by $\{\lambda_i\}$ the eigenvalues of $C_{zz}$, then the $i$-th eigenvalue of $C_{zz}^{1/2} - C_{zz,\bar{\nu}}^{-1/2} C_{zz}$ is

$$\lambda_i^{1/2} - \frac{\lambda_i}{\sqrt{\lambda_i + \bar{\nu}}} = \frac{\sqrt{\lambda_i^2 + \lambda_i \bar{\nu}} - \sqrt{\lambda_i^2}}{\sqrt{\lambda_i + \bar{\nu}}} \overset{(a)}{\leq} \frac{\bar{\nu}/2}{\sqrt{\lambda_i + \bar{\nu}}} \leq \sqrt{\bar{\nu}}/2,$$

where (a) follows from the concavity of the square root function. Thus

$$\|C_{zz}^{-1/2}C_{zx}f\|_{\mathcal{I}} - \|C_{zz,\bar{\nu}}^{-1/2}C_{zx}f\|_{\mathcal{I}} \le \sqrt{\bar{\nu}}\|Ef\|_{\mathcal{I}}/2 \le \sqrt{\bar{\nu}}\|E\|\|f\|_{\mathcal{H}}/2,$$

and

$$\sqrt{\frac{1-2r}{1-r}}\left(\|C_{zz}^{-1/2}C_{zx}f\|_{\mathcal{I}} - \frac{\sqrt{\bar{\nu}}}{2}\|E\|\|f\|_{\mathcal{H}}\right) \le \|\hat{C}_{zz,\bar{\nu}}^{-1/2}C_{zx}f\|_{\mathcal{I}} \le \sqrt{\frac{1}{1-r}}\|C_{zz}^{-1/2}C_{zx}f\|_{\mathcal{I}}. \tag{28}$$

Note that on the event $B_n(r, L)$, we have

$$\|\|\hat{C}_{zz,\bar{\nu}}^{-1/2}\hat{C}_{zx}f\|_{\mathcal{I}} - \|\hat{C}_{zz,\bar{\nu}}^{-1/2}C_{zx}f\|_{\mathcal{I}}| \le \|\hat{C}_{zz,\bar{\nu}}^{-1/2}\|\|\hat{C}_{zx} - C_{zx}\|\|f\|_{\mathcal{H}} \le \frac{L}{\sqrt{n\bar{\nu}}}\|f\|_{\mathcal{H}}. \tag{29}$$

Combining (28) and (29) completes the proof. $\qquad\square$

**Lemma B.2.** *Conditioned on the event $B_n(r, L)$ for $r \in (0, 1/2)$, we have for all $f \in \mathbb{B}$ that can be written as $f = f_h + f_e$ where $f_h \in \mathcal{H}$, $\|f_e\| \le 2\tau_m$ and for arbitrary $m \in \mathbb{N}$,*

$$-r_{n,m}^{(1)} + \sqrt{\frac{1-2r}{1-r}}\|\mathbb{E}(f - \mathbf{y} \mid \mathbf{z})\|_p \le \sqrt{\ell_n(f)} \le r_{n,m}^{(1)} + \sqrt{\frac{1}{1-r}}\|\mathbb{E}(f - \mathbf{y} \mid \mathbf{z})\|_p, \tag{30}$$

*where $\ell_n(f)$ is defined in (25), and $f_m^\dagger \in \mathcal{H}$ is an approximation of $f^\dagger$ in $\mathcal{H}$ such that $\|f^\dagger - f_m^\dagger\| \le \tau_m$, and*

$$r_{n,m}^{(1)} \lesssim \left(\frac{L}{\sqrt{\bar{\nu}n}} + \sqrt{\bar{\nu}}\right)(\|f_h - f_m^\dagger\|_{\mathcal{H}} + 1) + \tau_m. \tag{31}$$

*Proof.* Define the random vectors

$$R := Y - f^\dagger(X), \quad E := f^\dagger(X) - f_m^\dagger(X) - f_e(X),$$

so that $\mathbb{E}(R \mid \mathbf{Z}) = 0$, $\|E\|_\infty \le 2\tau_m$. Consider the decomposition

$$\sqrt{\ell_n(f)} = \left\|\hat{C}_{zz,\bar{\nu}}^{-1/2}\left(-\frac{S_z^*(R + E)}{n} + \hat{C}_{zx}(f_h - f_m^\dagger)\right)\right\|_{\mathcal{I}}$$

$$\le \left\|\hat{C}_{zz,\bar{\nu}}^{-1/2}\frac{S_z^*(R + E)}{n}\right\|_{\mathcal{I}} + \left\|\hat{C}_{zz,\bar{\nu}}^{-1/2}\hat{C}_{zx}(f_h - f_m^\dagger)\right\|_{\mathcal{I}}, \tag{32}$$

$$\sqrt{\ell_n(f)} \ge -\left\|\hat{C}_{zz,\bar{\nu}}^{-1/2}\frac{S_z^*(R + E)}{n}\right\|_{\mathcal{I}} + \left\|\hat{C}_{zz,\bar{\nu}}^{-1/2}\hat{C}_{zx}(f_h - f_m^\dagger)\right\|_{\mathcal{I}}. \tag{33}$$

On the event $B_n(r, L)$, we have

$$\left\|\hat{C}_{zz,\bar{\nu}}^{-1/2}\frac{S_z^*R}{n}\right\|_{\mathcal{I}} \le \bar{\nu}^{-1/2}\left\|\frac{S_z^*R}{n}\right\|_{\mathcal{I}} \le L(n\bar{\nu})^{-1/2}.$$

And since

$$\left\|\hat{C}_{zz,\bar{\nu}}^{-1/2}\frac{S_z^*E}{n}\right\|_{\mathcal{I}}^2 = \frac{1}{n^2}\left\langle S_z C_{zz,\bar{\nu}}^{-1}S_z^*E, E\right\rangle \le \frac{1}{n^2}\|S_z C_{zz,\bar{\nu}}^{-1}S_z^*\|\|E\|_2^2$$

$$= \underbrace{\|K_{zz}(K_{zz} + \bar{\nu}nI)^{-1}\|}_{\le 1} \cdot \frac{1}{n}\|E\|_2^2 \le 9\tau_m^2,$$

where the last inequality follows from the fact that $\|E\|_\infty \le \|E\| \le 3\tau_m$, we have

$$\left\|\hat{C}_{zz,\bar{\nu}}^{-1/2}\frac{S_z^*(R + E)}{n}\right\|_{\mathcal{I}} \le L(n\bar{\nu})^{-1/2} + 3\tau_m. \tag{34}$$

For the second term in (32) and (33), recall that by Lemma B.1 we have

$$-r_n^{(0)} + \sqrt{\frac{1-2r}{1-r}}\|C_{zz}^{-1/2}C_{zx}(f_h - f_m^\dagger)\|_{\mathcal{I}} \le \|\hat{C}_{zz,\bar{\nu}}^{-1/2}\hat{C}_{zx}(f_h - f_m^\dagger)\|_{\mathcal{I}}$$

$$\le r_n^{(0)} + \sqrt{\frac{1}{1-r}}\|C_{zz}^{-1/2}C_{zx}(f_h - f_m^\dagger)\|_{\mathcal{I}},$$

where

$$r_n^{(0)} \lesssim \left( L(\bar{\nu}n)^{-1/2} + \bar{\nu}^{1/2} \right) \|f_h - f_m^\dagger\|_{\mathcal{H}}.$$

From the triangle inequality $||a| - |b|| \le |a - b|$, we have

$$\begin{aligned}
&|\|\mathbb{E}(f - \mathbf{y} \mid \mathbf{z})\|_p - \|C_{zz}^{-1/2} C_{zx}(f_h - f_m^\dagger)\|_{\mathcal{I}}| \\
&= |\|\mathbb{E}(f - f^\dagger \mid \mathbf{z})\|_p - \|\mathbb{E}(f_h - f_m^\dagger \mid \mathbf{z})\|_p| \\
&\le \|\mathbb{E}(f_e + f_m^\dagger - f^\dagger \mid \mathbf{z})\|_p \\
&\le \|f_e + f_m^\dagger - f^\dagger\|_\infty \le 3\tau_m.
\end{aligned}$$

Since $r \in (0, 1/2)$, we know $\sqrt{\frac{1}{1-r}} \le 2$ and $\sqrt{\frac{1-2r}{1-r}} \le 1$. Thus, we have

$$\sqrt{\frac{1-2r}{1-r}} \|\mathbb{E}(f - \mathbf{y} \mid \mathbf{z})\|_p - r_n^{(0)} - 4\tau_m \le \|\hat{C}_{zz,\bar{\nu}}^{-1/2} \hat{C}_{zx}(f_h - f_m^\dagger)\|_{\mathcal{I}} \tag{35}$$

$$\le \sqrt{\frac{1}{1-r}} \|\mathbb{E}(f - \mathbf{y} \mid \mathbf{z})\|_p + r_n^{(0)} + 6\tau_m. \tag{36}$$

Plugging (34), (35) and (36) to (32) and (33) completes the proof. $\qquad\square$

## B.2 Proof of Theorem 3.1

Let $\{m_n : n \in \mathbb{N}\}$ be an increasing sequence to be determined later. We drop the subscript $n$ below for brevity. Let $\{\Theta_m : m \in \mathbb{N}\}$ be defined as in Theorem A.1 with $w_0 = f^\dagger$; recall that $C_\Theta$ can be set arbitrarily large. In the event $B_n(r, L)$ we fix $r = 1/3$ and determine $L$ later; both parameters $r, L$ will be dropped for brevity.

We define the unnormalized quasi-posterior measure as follows:

$$\tilde{\Pi}(A \mid \mathcal{D}^{(n)}) := \int_A \exp\left( -\frac{n}{2\lambda} \ell_n(f) \right) \Pi(df). \tag{37}$$

Consider the decomposition

$$\begin{aligned}
\mathbb{E}(\Pi(\mathrm{err}_{n,f} \mid \mathcal{D}^{(n)})) &\le \mathbb{E}(\Pi(\mathrm{err}_{n,f} \mid \mathcal{D}^{(n)}) \mid B_n) + (1 - \mathbb{P}(B_n)) \\
&\le \mathbb{E}\left( \frac{\tilde{\Pi}(\Theta_m^c \mid \mathcal{D}^{(n)}) + \tilde{\Pi}(\mathrm{err}_{n,f} \cap \Theta_m \mid \mathcal{D}^{(n)})}{\tilde{\Pi}(\Theta \mid \mathcal{D}^{(n)})} \,\Big|\, B_n \right) + (1 - \mathbb{P}(B_n)) \\
&\le \mathbb{E}\left( \frac{\Pi(\Theta_m^c) + \tilde{\Pi}(\mathrm{err}_{n,f} \cap \Theta_m \mid \mathcal{D}^{(n)})}{\tilde{\Pi}(\Theta \mid \mathcal{D}^{(n)})} \,\Big|\, B_n \right) + (1 - \mathbb{P}(B_n)) \\
&=: (\mathrm{I}) + (\mathrm{II}),
\end{aligned}$$

where the last inequality follows from $-\frac{n}{2\lambda} \ell_n(f) \le 0$ and the definition of $\tilde{\Pi}$ in (37).

By Assumption 3.2, we can find $f_m^\dagger \in \mathcal{H}$ such that $\|f^\dagger - f_m^\dagger\| \le \tau_m$ and

$$\|f_m^\dagger\|_{\mathcal{H}}^2 \le \inf_{h \in \mathcal{H}: \|h - f^\dagger\| \le \tau_m} \|h\|_{\mathcal{H}}^2 + 1 \le m\tau_m^2 + 1. \tag{38}$$

We first consider the denominator $\tilde{\Pi}(\Theta \mid \mathcal{D}^{(n)})$ in (I). For any $f \in \mathbb{B}$ with $\|f - f_m^\dagger\| \le 2\tau_m$, using Lemma B.2 with $f_h = f_m^\dagger$ and $f_e = f - f_m^\dagger$, we have the following on the event $B_n$:

$$\sqrt{\ell_n(f)} \le r_{n,m}^{(1)} + \sqrt{\frac{3}{2}} \|\mathbb{E}(f - \mathbf{y} \mid \mathbf{z})\|_p \le r_{n,m}^{(1)} + 4\tau_m,$$

where $r_{n,m}^{(1)}$ is defined in (31) and the last inequality follows from that

$$|\mathbb{E}(f - \mathbf{y} \mid \mathbf{z})| \le \mathbb{E}(|f - f_m^\dagger| \mid \mathbf{z}) + \mathbb{E}(|f_m^\dagger - f^\dagger| \mid \mathbf{z}) \le \|f - f_m^\dagger\| + \|f_m^\dagger - f^\dagger\| \le 3\tau_m.$$

Plugging $f_h - f_m^\dagger = 0$ into the definition of $r_{n,m}^{(1)}$ yields

$$\ell_n(f) \lesssim \frac{L^2}{\bar{\nu}n} + \bar{\nu} + \tau_m^2, \quad \text{if } B_n \text{ and } \|f - f_m^\dagger\| \le 2\tau_m \text{ hold.} \tag{39}$$

Thus, on the event $B_n$, we have for some fixed constant $C_1 > 0$,

$$
\begin{aligned}
\tilde{\Pi}(\Theta \mid \mathcal{D}^{(n)}) &\ge \int_{\{f \in \Theta : \|f - f_m^\dagger\| \le 2\tau_m\}} \exp\left(-\frac{n}{2\lambda}\ell_n(f)\right)\Pi(df) \\
&\ge \Pi(\{\|f - f_m^\dagger\| \le 2\tau_m\}) \cdot \exp\left(-\frac{C_1 n}{\lambda}\left(\frac{L^2}{\bar{\nu}n} + \bar{\nu} + \tau_m^2\right)\right) \\
&\overset{(23)}{\ge} \exp\left(-3m\tau_m^2 - \frac{C_1 n}{\lambda}\left(\frac{L^2}{\bar{\nu}n} + \bar{\nu} + \tau_m^2\right)\right).
\end{aligned} \tag{40}
$$

Now we consider the numerators in (I). First by Theorem A.1 we have $\Pi(\Theta_m^c) \le \exp(-C_\Theta m\tau_m^2)$, where $C_\Theta$ is any constant such that $e^{-C_\Theta m\tau_m^2} \le 1/2$, to be determined later. Thus,

$$\frac{\Pi(\Theta_m^c)}{\tilde{\Pi}_n(\Theta \mid \mathcal{D}^{(n)})} \le \exp\left(-\left(C_\Theta - 3 - \frac{C_1 n}{m\lambda}\right)m\tau_m^2 + \frac{C_1 n}{2\lambda}\left(\frac{L^2}{\bar{\nu}n} + \bar{\nu}\right)\right). \tag{41}$$

We now turn to the $\tilde{\Pi}(\mathrm{err}_{n,f} \cap \Theta_n)$ term in the numerators of (I). Noting that for non-negative numbers, $a \ge b - c$ implies $2a^2 \ge b^2 - 2c^2$, and by Lemma B.2, on the event $B_n$, for any $f \in \mathrm{err}_{n,f} \cap \Theta_m$ we have

$$\ell_n(f) \ge \frac{1}{4}\|\mathbb{E}(f - \mathbf{y} \mid \mathbf{z})\|_p^2 - \left(r_{n,m}^{(1)}\right)^2 \ge \frac{M\epsilon_n^2}{4} - \left(r_{n,m}^{(1)}\right)^2. \tag{42}$$

Recalling that when $f \in \Theta_m$, we can write $f = f_h + f_e$, where $f_h \in \underline{J}_m \mathcal{H}_1$ and $\|f_e\| \le \tau_n$. In view of (17), (18) and (38), we find $\|f_m^\dagger\|_{\mathcal{H}}^2 \le m\tau_m^2 + 1$. From Corollary A.1 and (31), we know

$$
\begin{aligned}
\left(r_{n,m}^{(1)}\right)^2 &\lesssim \left(\frac{L^2}{\bar{\nu}n} + \bar{\nu}\right) \cdot \left(\|f_h\|_{\mathcal{H}}^2 + \|f_m^\dagger\|_{\mathcal{H}}^2 + 1\right) + \tau_m^2 \\
&\lesssim \left(\frac{L^2}{\bar{\nu}n} + \bar{\nu}\right) \cdot \left((C_\Theta + 1)m\tau_m^2 + 1\right) + \tau_m^2.
\end{aligned} \tag{43}
$$

Combining (40), (42) and (43), we know there is a fixed constant $C_2 > C_1$ such that the following holds on the event $B_n$,

$$\frac{\tilde{\Pi}(\Theta_m \cap \mathrm{err}_{n,f} \mid \mathcal{D}^{(n)})}{\tilde{\Pi}(\Theta \mid \mathcal{D}^{(n)})} \le \exp\left(-\frac{Mn\epsilon_n^2}{8\lambda} + \Gamma_1 m\tau_m^2 + \Gamma_2\left(\frac{L^2}{\bar{\nu}n} + \bar{\nu}\right)\right), \tag{44}$$

where

$$\Gamma_1 := 3 + \frac{C_2 n}{m\lambda},$$

$$\Gamma_2 := 1 + (C_\Theta + 1)m\tau_m^2 + \frac{C_2 n}{2\lambda}.$$

Setting $C_\Theta = 4 + 2C_1$, $\epsilon_n = \tau_m$, $m = \lambda = \sqrt{n}$, $\bar{\nu} = L/\sqrt{n}$, $L = \min\{m\tau_m^2, \gamma_n\}$ where $\gamma_n \to \infty$ is a sequence with arbitrarily slow growth, we can verify that there exists an $M > 0$ such that both (41) and (44) converges to zero by noting that as $n \to \infty$, $m\tau_m^2 \to \infty$, $\tau_m^2 \to 0$. Hence, the term (I) converges to zero.

Next, we shall show that (II) tends to zero as $n \to \infty$. This is equivalent to verify that the right hand sides of (47), (48) and (49) tend to zero. Since $L \to \infty$, we know (48) and (49) will vanish. The following inequality shows that (47) also vanishes.

$$\bar{\nu}n - \log N(\bar{\nu}) = L\sqrt{n} - \log N(\bar{\nu}) \ge \sqrt{n} - \log O\left(\frac{\sqrt{n}}{L}\right) \to \infty,$$

where the inequality follows from the fact $N(\bar{\nu}) = O(\bar{\nu}^{-1})$ (see [65, Proposition 3]).

## B.3 Proof of Proposition 3.1

We follow the choice of parameters (except for $r$, which will be set to $1/\max\{3, L\}$) as in Theorem 3.1 to show that

$$\Pi\left(\left\{f : \lim_{n\to\infty}\mathbb{P}_{\mathcal{D}^{(n)}}\left(\left|\sqrt{\ell_n(f)} - \|\mathbb{E}(f - \mathbf{y}\mid\mathbf{z})\|_p\right| > \delta\right) = 0, \forall\delta > 0\right\}\right) = 1.$$

From (17) and (18) we can see that for any $\tau_m$ that satisfies the condition of Theorem A.1, $\tilde{\tau}_m \geq \tau_m$ will also satisfy it. Thus we choose $\tilde{\tau}_m := \max\{\tau_m, \sqrt{2(C_\Theta m)^{-1}\log m}\}$ and define $\tilde{\Theta}_m$ accordingly. Then by Theorem 3.1 and the Borel-Cantelli Lemma, the set

$$S := \{f \in \mathbb{B} : \text{there exists } M_f > 0 \text{ such that } f \in \tilde{\Theta}_m \text{ for every } m > M_f\} \tag{45}$$

has prior probability 1 since $\sum_{m\geq 1} e^{-C_\Theta m \tilde{\tau}_m^2} \leq \sum_{m\geq 1} m^{-2} < \infty$.

For $f \in S$ and $m > M_f$, by Lemma B.2, we know the following holds on the event $B_n(r, L)$:

$$-r_{n,m}^{(1)} + \sqrt{\frac{1 - 2r}{1 - r}}\|\mathbb{E}(f - \mathbf{y}\mid\mathbf{z})\|_p \leq \sqrt{\ell_n(f)} \leq r_{n,m}^{(1)} + \sqrt{\frac{1}{1 - r}}\|\mathbb{E}(f - \mathbf{y}\mid\mathbf{z})\|_p,$$

with $m = \sqrt{n}, \bar{\nu} = L/\sqrt{n}$ as in Theorem A.1, the above becomes

$$r_{n,m}^{(1)} \lesssim \left(\frac{L}{\sqrt{\bar{\nu}n}} + \sqrt{\bar{\nu}}\right)(\|f_h - f_m^\dagger\|_{\mathcal{H}} + 1) + \tilde{\tau}_m \lesssim \sqrt{\frac{L}{n}}(\sqrt{m\tilde{\tau}_m^2} + 1) + \tilde{\tau}_m \lesssim \sqrt{L}\tilde{\tau}_m.$$

Since the growth of $L$ can be arbitrarily slow, and $\tilde{\tau}_m \to 0$, we have $\lim_{n\to\infty} r_{n,m}^{(1)} = 0$. Note that $r := 1/\max\{3, L\} \to 0$, from (47), (48) and (49), it can be verified that $\lim_{n\to\infty}\mathbb{P}(B_n(r, L)) = 1$. Combining with the above inequality, we know that

$$\lim_{n\to\infty}\mathbb{P}_{\mathcal{D}^{(n)}}\left(\left|\sqrt{\ell_n(f)} - \|\mathbb{E}(f - \mathbf{y}\mid\mathbf{z})\|_p\right| > \delta\right) = 0, \quad \forall f \in S, \delta > 0.$$

## B.4 Auxiliary Results

In this section, we collect several auxiliary results used in our proofs.

**Lemma B.3.** *For $r \in (0, 1)$, define*

$$A_n(r) := \{\|C_{zz,\bar{\nu}}^{-1/2}(\hat{C}_{zz} - C_{zz})C_{zz,\bar{\nu}}^{-1/2}\| \leq r\}. \tag{46}$$

*Then when $\bar{\nu} \leq \sup_z k(z, z) =: \kappa^2$, and $r \geq \sqrt{\kappa^2/(\bar{\nu}n)} + \kappa^2/(3\bar{\nu}n)$, we have*

$$1 - \mathbb{P}(A_n(r)) \leq 4N(\bar{\nu})\exp\left(-\frac{\bar{\nu}nr^2}{2\kappa^2(1 + r/3)}\right), \tag{47}$$

*where $N(\bar{\nu}) := \mathrm{Tr}(C_{zz}C_{zz,\bar{\nu}}^{-1})$ is the effective dimension of $C_{zz}$.*

*Proof.* This is Lemma 1 in [33], with $\delta = 0$ (in their notation). $\qquad\square$

The following lemma is a standard concentration result on the operator $C_{zx}$. See, e.g., Caponnetto and De Vito [65], Fukumizu [66], De Vito et al. [67]. We will give its proof for completeness.

**Lemma B.4.** *If $\sup_x k(x, x) \leq \kappa^2$ and $\sup_z k(z, z) \leq \kappa^2$, then for any $\delta \in (0, 1)$, we have for any constant $C > 0$,*

$$1 - \mathbb{P}\left(\|\hat{C}_{zx} - C_{zx}\| \leq \frac{C}{\sqrt{n}}\right) \leq 2\exp\left(-\frac{C}{4\kappa^2}\right). \tag{48}$$

*Proof.* Define the random variable $\xi := k(x, \cdot) \otimes k(z, \cdot)$. It is easy to verify that $\xi$ is a Hilbert-Schmidt operator from $\mathcal{H}$ to $\mathcal{I}$, and $\mathbb{E}_{x,z}\xi = C_{zx}$. Note that $\|\xi\|_{\mathrm{HS}} = \sqrt{k(x,x)k(z,z)} \leq \kappa^2$ and $\mathbb{E}\|\xi\|_{\mathrm{HS}}^2 \leq \kappa^4$. From Proposition 2 in [65], we conclude that for any $\delta \in (0, 1)$,

$$\mathbb{P}\left(\|\hat{C}_{zx} - C_{zx}\|_{\mathrm{HS}} \leq \frac{4\kappa^2}{\sqrt{n}}\log\frac{2}{\delta}\right) \geq 1 - \delta.$$

Finally, this lemma can be proved by a simple algebra and the fact that $\|\cdot\| \leq \|\cdot\|_{\mathrm{HS}}$. $\qquad\square$

**Lemma B.5.** *Assume that $f^\dagger(x) - y$ is a $\Lambda$-subexponential random variable and $\sup_z k(z, z) \leq \kappa^2$, then there exists a universal constant $c_1$ such that for all $C > 0$,*

$$1 - \mathbb{P}\left(\left\|\frac{S_z^*}{n}(f^\dagger(X) - Y)\right\|_{\mathcal{I}} \leq \frac{C}{\sqrt{n}}\right) \leq 2\exp\left(-\frac{C}{c_1 \Lambda \kappa}\right). \tag{49}$$

*Proof.* Define the random variable $\xi := k(z, \cdot)(f^\dagger(x) - y)$. Since $f^\dagger(x) - y$ is $\Lambda$-subexponential, we know $\left(\mathbb{E}|f^\dagger(x) - y|^p\right)^{1/p} \leq c_0 \Lambda p$ for all $p \geq 1$ for some universal constant $c_0$ (See, e.g., Proposition 2.7.1 in Vershynin [68]). Recall that the Stirling's formula $\sqrt{2\pi} n^{n+\frac{1}{2}} e^{-n} \leq n!$, we know $\mathbb{E}\|\xi\|_{\mathcal{I}}^n = \mathbb{E}k(z, z)^{\frac{n}{2}}|f^\dagger(x) - y|^n \leq cn!(c\Lambda\kappa)^n$ for some universal constant $c$. Thus, from the fact that $\mathbb{E}\xi = \mathbb{E}(k(z, \cdot)\mathbb{E}(f^\dagger(x) - y \mid z)) = 0$ and Proposition 2 in [65], it has

$$\mathbb{P}\left(\left\|\frac{S_z^*}{n}(f^\dagger(X) - Y)\right\|_{\mathcal{I}} \leq \frac{4c\kappa\Lambda}{\sqrt{n}}\log\frac{2}{\delta}\right) \geq 1 - \delta.$$

The final conclusion follows by a simple algebra. $\qquad\square$

**Lemma B.6.** *On the event $A_n(r)$, we have*

$$\|C_{zz}^{1/2}\hat{C}_{zz,\bar\nu}^{-1}C_{zz}^{1/2}\| \leq \|C_{zz,\bar\nu}^{1/2}\hat{C}_{zz,\bar\nu}^{-1}C_{zz,\bar\nu}^{1/2}\| \leq \frac{1}{1-r}, \tag{50}$$

$$\|C_{zz,\bar\nu}^{1/2}(C_{zz,\bar\nu}^{-1} - \hat{C}_{zz,\bar\nu}^{-1})C_{zz,\bar\nu}^{1/2}\| \leq \frac{r}{1-r}. \tag{51}$$

*Proof.* (50) is Eq. (19) in [33], with (in their notation) $z = 1$. For (51), note that

$$\|C_{zz,\bar\nu}^{1/2}(C_{zz,\bar\nu}^{-1} - \hat{C}_{zz,\bar\nu}^{-1})C_{zz,\bar\nu}^{1/2}\| = \|I - C_{zz,\bar\nu}^{1/2}(C_{zz,\bar\nu} - (C_{zz} - \hat{C}_{zz}))^{-1}C_{zz,\bar\nu}^{1/2}\|$$

$$= \|I - (I - C_{zz,\bar\nu}^{-1/2}(C_{zz} - \hat{C}_{zz})C_{zz,\bar\nu}^{-1/2})^{-1}\|.$$

Define $D := C_{zz,\bar\nu}^{-1/2}(C_{zz} - \hat{C}_{zz})C_{zz,\bar\nu}^{-1/2}$. Then on the event $A_n(r)$, the right hand side above is

$$\|(I - D)^{-1}\cdot(-D)\| \leq \|(I - D)^{-1}\|\|D\| \leq \frac{1}{1-r}\cdot r,$$

where the last inequality uses the fact that $\|D\| \leq r$ on $A(r)$, and that $\|(I - D)^{-1}\| \leq (1 - \|D\|)^{-1}$. $\qquad\square$

# C  Analysis of the Approximate Inference Algorithm

## C.1  Proof of the Double Randomized Prior Trick

### C.1.1  A Function-Space Equivalent to Proposition 4.1

We first claim that Proposition 4.1 is equivalent to the following function-space version, the proof of which is deferred to Section C.1.3:

**Proposition C.1.** *Let* $\tilde{\mathcal{H}}, \tilde{\mathcal{I}}$ *be finite-dimensional RKHSes with kernels* $k_x, k_z$, *respectively,*

$$g_0 \sim \mathcal{GP}(0, \lambda\nu^{-1}\tilde{k}_z), \; f_0 \sim \mathcal{GP}(0, \tilde{k}_x), \; \tilde{y}_i \sim \mathcal{N}(y_i, \lambda).$$

*Then the optima* $f^*$ *of*

$$\min_{f \in \tilde{\mathcal{H}}} \max_{g \in \tilde{\mathcal{I}}} \mathcal{L}(f, g) := \sum_{i=1}^{n} \left( (f(x_i) - \tilde{y}_i)g(z_i) - \frac{g(z_i)^2}{2} \right) - \frac{\nu}{2}\|g - g_0\|_{\tilde{\mathcal{I}}}^2 + \frac{\lambda}{2}\|f - f_0\|_{\tilde{\mathcal{H}}}^2 \quad (52)$$

*follows the posterior distribution* (6), *with the kernels* $\tilde{k}_x, \tilde{k}_z$.

*Proof of the equivalence.* Observe that (52) is exactly the same as (14) when the random feature parameterization $\phi \mapsto g(z; \phi)$ is injective,[9] in which case we have $\|\phi\|_2 = \|g(\cdot; \phi)\|_{\tilde{\mathcal{I}}}$. Otherwise, observe that on the subspace

$$\Phi_s := \text{span}\{\phi_{z,m}(z') : z' \in \mathcal{Z}\},$$

$\|\phi\|_2 = \|g(\cdot; \phi)\|_{\tilde{\mathcal{I}}}$ always holds: this follows by definition of $\tilde{k}_z$ when $\phi$ is a finite linear combination of the $\phi$'s, and the general case follows by continuity (note that $\tilde{\mathcal{I}}$ is already defined by $\tilde{k}_z$). Clearly any $g - g_0 \in \tilde{\mathcal{I}}$ can be parameterized with some $\phi$ in this subspace, so the optima of (52) is a valid candidate solution for (14). On the other hand, for any $\phi - \phi_0$ outside the aforementioned subspace, we have $\|\phi - \phi_0\|_2 > \|g(\cdot; \phi) - g(\cdot; \phi_0)\|_{\tilde{\mathcal{I}}}$. Therefore, the optimal $\phi$ of (14) must satisfy $\|\phi - \phi_0\|_2 = \|g(\cdot; \phi) - g(\cdot; \phi_0)\|_{\tilde{\mathcal{I}}}$, and thus solves (52). As a similar result also holds for $f$, we conclude that the two objectives are equivalent. $\square$

*Remark* C.1. The non-injective setting above justifies the formal analysis of (16) in the main text. We also remark that any parameter $\theta, \phi$ visited by the SGDA algorithm on (14) or (16) (starting from $\theta_0, \phi_0$) satisfies

$$\theta - \theta_0 \in \Theta_s, \; \phi - \phi_0 \in \Phi_s.$$

Thus $\|\phi - \phi_0\|_2 = \|g(\cdot; \phi) - g(\cdot; \phi_0)\|_{\tilde{\mathcal{I}}}$ (and similarly for $\theta$), and from the perspective of the SGDA algorithm, the objectives (52) and (14) are *always* the same. This can be proved by induction. Take $\phi$ for example; clearly $\phi = \phi_0$ satisfies the above. For $\phi_\ell$ obtained at the $\ell$-th step of SGDA, we have

$$\phi_\ell - \phi_0 = (1 - \nu)(\phi_{\ell-1} - \phi_0) + V_\ell^\top \phi_{z,m}(Z),$$

where $V_\ell \in \mathbb{R}^n$ is independent of $\phi_\ell$. Thus $\phi_\ell - \phi_0 \in \Phi_s$ by definition of $\Phi_s$ and the inductive hypothesis.

### C.1.2  Matrix Identities

We list two identities here that will be used in the derivations.

**Lemma C.1.** *Let* $U, C, V, S$ *be operators between appropriate Banach spaces,* $\lambda \in \mathbb{R} \setminus \{0\}$, *then*

$$(\lambda I + UCV)^{-1} = \lambda^{-1}(I - U(\lambda C^{-1} + VU)^{-1}V), \quad (53)$$

$$S(S^*S + \lambda I)^{-1} = (SS^* + \lambda I)^{-1}S. \quad (54)$$

*Proof.* Recall the Woodbury identity:

$$(A + UCV)^{-1} = A^{-1} - A^{-1}U(C^{-1} + VA^{-1}U)^{-1}VA^{-1}.$$

---

[9]Most random feature models, such as the random Fourier feature model, satisfies this property almost surely.

Then, we have
$$(\lambda I + UCV)^{-1} = \lambda^{-1}I - \lambda^{-2}U(C^{-1} + \lambda^{-1}VU)^{-1}V$$
$$= \lambda^{-1}(I - U(\lambda C^{-1} + VU)^{-1}V).$$
And,
$$S(S^*S + \lambda I)^{-1} = S(\lambda^{-1}I - \lambda^{-2}S^*(\lambda^{-1}SS^* + I)^{-1}S)$$
$$= \lambda^{-1}(S - SS^*(SS^* + \lambda I)^{-1}S)$$
$$= (SS^* + \lambda I)^{-1}S.$$

$\square$

### C.1.3 Proof of Proposition C.1

Define $Y = (y_1, \ldots, y_n), \tilde{Y} = (\tilde{y}_1, \ldots, \tilde{y}_n)$. We rewrite the objective as

$$\mathcal{L}(f,g) = \left( \langle n\hat{C}_{zx}f - S_z^*\tilde{Y}, g\rangle_{\tilde{\mathcal{I}}} - \frac{1}{2}\langle n\hat{C}_{zz}g, g\rangle_{\tilde{\mathcal{I}}} - \frac{\nu}{2}\|g - g_0\|_{\tilde{\mathcal{I}}}^2 \right) + \frac{\lambda}{2}\|f - f_0\|_{\mathcal{H}}^2$$

$$= n\left( \langle \hat{C}_{zx}f - n^{-1}S_z^*\tilde{Y}, g\rangle_{\tilde{\mathcal{I}}} - \frac{1}{2}\langle \hat{C}_{zz,\bar{\nu}}\, g, g\rangle_{\tilde{\mathcal{I}}} + \bar{\nu}\langle g, g_0\rangle_{\tilde{\mathcal{I}}} - \frac{\bar{\nu}}{2}\|g_0\|_{\tilde{\mathcal{I}}}^2 \right) + \frac{\lambda}{2}\|f - f_0\|_{\mathcal{H}}^2,$$

where $S_z, \hat{C}_{zx}, \hat{C}_{zz}$ are now defined w.r.t. the approximate kernels. The optimal $g^*$ for fixed $f$ is

$$g^*(f) = \hat{C}_{zz,\bar{\nu}}^{-1}(\hat{C}_{zx}f - n^{-1}S_z^*\tilde{Y} + \bar{\nu}g_0). \tag{55}$$

Plugging $g^*$ back to the objective, we have

$$\mathcal{L}(f, g^*(f)) = \frac{n}{2}\langle g^*, \hat{C}_{zz,\bar{\nu}}g^*\rangle_{\tilde{\mathcal{I}}} + \frac{\lambda}{2}\|f - f_0\|_{\mathcal{H}}^2 - \frac{n\bar{\nu}}{2}\|g_0\|_{\tilde{\mathcal{I}}}^2,$$

$$\partial_f\mathcal{L} = n\hat{C}_{xz}\hat{C}_{zz,\bar{\nu}}^{-1}\hat{C}_{zz,\bar{\nu}}g^* + \lambda(f - f_0)$$

$$= n\hat{C}_{xz}\hat{C}_{zz,\bar{\nu}}^{-1}(\hat{C}_{zx}f - n^{-1}S_z^*\tilde{Y} + \bar{\nu}g_0) + \lambda(f - f_0).$$

Setting $\partial_f\mathcal{L}$ to zero, we obtain

$$f^* = (n\hat{C}_{xz}\hat{C}_{zz,\bar{\nu}}^{-1}\hat{C}_{zx} + \lambda I)^{-1}(n\hat{C}_{xz}\hat{C}_{zz,\bar{\nu}}^{-1}(n^{-1}S_z^*\tilde{Y} - \bar{\nu}g_0) + \lambda f_0). \tag{56}$$

Since

$$(n\hat{C}_{xz}\hat{C}_{zz,\bar{\nu}}^{-1}\hat{C}_{zx} + \lambda I)^{-1} = (n^{-1}S_x^*S_z\hat{C}_{zz,\bar{\nu}}^{-1}S_z^*S_x + \lambda I)^{-1}$$

$$= (S_x^*LS_x + \lambda I)^{-1}$$

$$\overset{(53)}{=} \lambda^{-1}\underbrace{(I - S_x^*(\lambda L^{-1} + S_xS_x^*)^{-1}S_x)}_{\text{defined as } \mathcal{C}}, \tag{57}$$

we can rewrite $f^*$ as

$$f^* = \lambda^{-1}\mathcal{C}(\hat{C}_{xz}\hat{C}_{zz,\bar{\nu}}^{-1}(S_z^*\tilde{Y} - \nu g_0) + \lambda f_0).$$

Clearly, $f^*$ is a Gaussian process. Suppose $f^*(x_*) \sim \mathcal{N}(S_*\mu', S_*\mathcal{C}'S_*^*)$, then

$$\mu' = \lambda^{-1}\mathcal{C}\, n\hat{C}_{xz}\hat{C}_{zz,\bar{\nu}}^{-1}(n^{-1}S_z^*Y) = \lambda^{-1}(I - S_x^*(\lambda L^{-1} + S_xS_x^*)^{-1}S_x)S_x^*LY$$

$$= \lambda^{-1}S_x^*(I - (\lambda L^{-1} + S_xS_x^*)^{-1}S_xS_x^*)LY$$

$$= S_x^*(\lambda L^{-1} + S_xS_x^*)^{-1}Y.$$

The RHS above matches the posterior mean (9) (with $k_x, k_z$ replaced by their random feature approximations) since $S_xS_x^* = K_{xx}$ and

$$S_*\mu' = S_*S_x^*(\lambda L^{-1} + S_xS_x^*)^{-1}Y = K_{*x}(\lambda L^{-1} + K_{xx})^{-1}Y = K_{*x}(\lambda + LK_{xx})^{-1}LY.$$

As $\tilde{Y} - Y, g_0$ and $f_0$ are independent, the covariance operator of $f^*$ is

$$\mathcal{C}' = \lambda^{-1}\mathcal{C}(\hat{C}_{xz}\hat{C}_{zz,\bar{\nu}}^{-1}(n\lambda\hat{C}_{zz} + \lambda\nu I)\hat{C}_{zz,\bar{\nu}}^{-1}\hat{C}_{zx} + \lambda^2 I)\lambda^{-1}\mathcal{C}$$

$$= \lambda^{-1}\mathcal{C}(\lambda n\hat{C}_{xz}\hat{C}_{zz,\bar{\nu}}^{-1}\hat{C}_{zx} + \lambda^2 I)\lambda^{-1}\mathcal{C} \overset{(57)}{=} \mathcal{C}.$$

In view of (57), we know

$$S_*\mathcal{C}'S_*^* = S_*S_*^* - S_*S_x^*(\lambda L^{-1} + S_xS_x^*)^{-1}S_xS_*^*$$

$$= K_{**} - K_{*x}(\lambda L^{-1} + K_{xx})^{-1}K_{x*},$$

which matches the posterior covariance matrix (10) with replaced kernels.

### C.1.4 Discussion of KernelIV [7]

The KernelIV estimator [7] is motivated as a kernelized generalization for 2SLS. Its *first stage* consists of estimating the conditional expectation operator $E$, restricted on $\mathcal{H}$; we can see from Theorem 1 therein that their estimator $E_\lambda^n$ coincides with our choice of $\hat{E}_n = \hat{C}_{zz,\bar{\nu}}^{-1}\hat{C}_{zx}$. Thus when the domain of the response variable $\mathcal{Y} = \mathbb{R}$, their second-stage objective reduces to

$$
\hat{\mathcal{E}}_n(f) := \frac{1}{n}\sum_{i=1}^{n}(\tilde{y}_i - \langle f, \hat{E}_n^* k(\tilde{z}_i,\cdot)\rangle_{\mathcal{H}})^2 + \bar{\lambda}\|f\|_{\mathcal{H}}^2
$$

$$
\equiv \langle f, (\hat{C}_{xz}\hat{C}_{zz,\bar{\nu}}^{-1}\hat{C}_{zz}\hat{C}_{zz,\bar{\nu}}^{-1}\hat{C}_{zx} + \bar{\lambda}I)f\rangle_{\mathcal{H}} + \left\langle f, \hat{C}_{xz}\hat{C}_{zz,\bar{\nu}}^{-1}\frac{S_{\tilde{z}}^*\tilde{Y}}{n}\right\rangle_{\mathcal{H}} + \bar{\lambda}\|f\|_{\mathcal{H}}^2 \tag{58}
$$

where in the last equality we have dropped the quadratic term about $\tilde{Y}$ as it is independent of $f$. Comparing with the kernelized DualIV objective (3), (58) is only different in their use of separate samples $(\tilde{z}_i, \tilde{y}_i)$,[10] and the replacement of $\hat{C}_{zz,\bar{\nu}}^{-1}$ in (3) with the asymptotically equivalent $\hat{C}_{zz,\bar{\nu}}^{-1}\hat{C}_{zz}\hat{C}_{zz,\bar{\nu}}^{-1}$. The similarity between the two objectives is also supported by previous report that empirically, the resulted estimators perform similarly [20].

(58) has an optimization-based equivalent form, similar to (4) to (3). Indeed, using a similar argument to Appendix C.1.3, we can see that

$$
\langle f, \hat{C}_{xz}\hat{C}_{zz,\bar{\nu}}^{-1}\hat{C}_{zz}\hat{C}_{zz,\bar{\nu}}^{-1}\hat{C}_{zx}f\rangle_{\mathcal{H}} = \frac{1}{n}\sum_{i=1}^{n}(2g(\tilde{z}_i)f(\tilde{x}_i) - g(\tilde{z}_i)^2) - 2\bar{\nu}\|g\|_{\mathcal{I}}^2,
$$

where $g = \hat{C}_{zz,\bar{\nu}}^{-1}\hat{C}_{zx}f$ solves

$$
\max_{g\in\mathcal{I}}\frac{1}{n}\sum_{i=1}^{n}(2g(\tilde{z}_i)f(\tilde{x}_i) - g(\tilde{z}_i)^2) - \bar{\nu}\|g\|_{\mathcal{I}}^2 \tag{59}
$$

which is equivalent to the KRR objective. Following this we can see that

$$
\hat{\mathcal{E}}_n(f) \equiv \frac{1}{n}\sum_{i=1}^{n}(2g(\tilde{z}_i)f(\tilde{x}_i) - g(\tilde{z}_i)^2 + f(\tilde{x}_i)h(\tilde{z}_i)) - 2\bar{\nu}\|g\|_{\mathcal{I}}^2 + \bar{\lambda}\|f\|_{\mathcal{H}}^2, \tag{60}
$$

where $h = \hat{C}_{zz,\bar{\nu}}^{-1}\frac{S_{\tilde{z}}^*\tilde{Y}}{n}$ represents $\hat{b}_n$ in (2). However, note the different regularizers on $g$ in (60) and (59) above, which is due to the replacement of $\hat{C}_{zz,\bar{\nu}}^{-1}$ with $\hat{C}_{zz,\bar{\nu}}^{-1}\hat{C}_{zz}\hat{C}_{zz,\bar{\nu}}^{-1}$ in (58); consequently, the objective $\hat{\mathcal{E}}_n$ no longer has a minimax formulation, and it is less clear whether a GDA-like algorithm will converge to the expected optima.

Finally, we note that Mastouri et al. [56] provides additional discussions on the difference between the kernelIV estimator and the kernelized dualIV estimator.

### C.2 Assumptions used in Proposition 4.2

The analysis in the subsequent subsections relies on the following assumptions on the random feature expansion. We only state them for $x$ for conciseness; the requirements for $z$ are similar.

The following assumption holds for, e.g., random Fourier features [48].

**Assumption C.1.**

$$
\sup_{x,x'\in\mathcal{X}}\left|k_x(x,x') - \tilde{k}_{x,m}(x,x')\right| \xrightarrow{p} 0, \quad as\ m\to\infty, \tag{61}
$$

The following assumption may be relaxed to require $\sup_x \tilde{k}_{x,m}(x,x)$ to have finite higher-order moments; we use this for simplicity.

**Assumption C.2.** *There exists a constant $\tilde{\kappa} > 0$ such that $\max_{m\in\mathbb{N}}\sup_{x\in\mathcal{X}}\tilde{k}_{x,m}(x,x) \leq \tilde{\kappa}$.*

---

[10]Note that $\tilde{y}_i$ here refers to the separate batch of unperturbed samples (see [7]), as opposed to the perturbed samples in the main text; we also assume that the two set of samples have the same sample size for simplicity.

## C.3 Analysis of Random Feature Approximation

We recall the following facts: for $A, B \in \mathbb{R}^{n \times n}$,
$$\|A\| \le \|A\|_F \le \sqrt{n}\|A\|, \quad A^{-1} - B^{-1} = A^{-1}(B - A)B^{-1}.$$

**Lemma C.2.** *For all $m \in \mathbb{N}$, let $k_{x,m}$ be a random feature approximation to $k_x$ such that* (61) *holds, and let $\tilde{k}_{z,m}$ be an approximation to $k_z$ satisfying a similar requirement as above. Then the random feature-approximated posterior $\Pi_m(f(x_*) \mid \mathcal{D}^{(n)}) = \mathcal{N}(\tilde{\mu}, \tilde{S})$ satisfies*
$$\lim_{m \to \infty} \sup_{x^* \in \mathcal{X}^l} \|\mu - \tilde{\mu}\|_2 = 0, \quad \lim_{m \to \infty} \sup_{x^* \in \mathcal{X}^l} \|\tilde{S} - S\|_F = 0,$$

*for any fixed training data $(X, Y, Z)$, $l \in \mathbb{N}$, and $\lambda, \nu > 0$. In the above, $\tilde{\mu}$ and $\tilde{S}$ are defined as*
$$\tilde{\mu} = \tilde{K}_{*x}(\lambda I + \tilde{L}\tilde{K}_{xx})^{-1}\tilde{L}Y,$$
$$\tilde{S} = \tilde{K}_{**} - \tilde{K}_{*x}\tilde{L}(\lambda I + \tilde{K}_{xx}\tilde{L})^{-1}\tilde{K}_{x*},$$
$$\tilde{L} = \tilde{K}_{zz}(\tilde{K}_{zz} + \nu I)^{-1},$$

*and the Gram matrices are defined using $\tilde{k}_{x,m}$ and $\tilde{k}_{z,m}$.*

*Proof.* Define
$$\epsilon_m = \max\left( \sup_{x,x' \in \mathcal{X}} \left| k(x,x') - \tilde{k}_{x,m}(x,x') \right|, \sup_{z,z' \in \mathcal{Z}} \left| k(z,z') - \tilde{k}_{z,m}(z,z') \right| \right).$$

By assumption $\epsilon_m \xrightarrow{p} 0$. For $\tilde{S}$ we consider the decomposition
$$\|\tilde{S} - S\| \le \|\tilde{K}_{**} - K_{**}\|$$
$$+ \|\tilde{K}_{*x} - K_{*x}\|\|\tilde{L}\|\|(\lambda I + \tilde{K}_{xx}\tilde{L})^{-1}\tilde{K}_{x*}\|$$
$$+ \|K_{*x}\|\|\tilde{L} - L\|\|(\lambda I + \tilde{K}_{xx}\tilde{L})^{-1}\tilde{K}_{x*}\|$$
$$+ \|K_{*x}L\|\|(\lambda I + \tilde{K}_{xx}\tilde{L})^{-1} - (\lambda I + K_{xx}L)^{-1}\|\|\tilde{K}_{x*}\|$$
$$+ \|K_{*x}L(\lambda I + K_{xx}L)^{-1}\|\|\tilde{K}_{x*} - K_{x*}\|$$
$$=: \text{(I)} + \text{(II)} + \text{(III)} + \text{(IV)} + \text{(V)}.$$

In the following, we use $O(\cdot)$ and $O_p(\cdot)$ to represent the asymptotic behaviour when $m \to \infty$. Since $n$ and $l$ are fixed, the operator norms of the matrices $K_{*x}, L, K_{xx}$ are $O(1)$. Observe that $\|K_{zz} - \tilde{K}_{zz}\| \le \sqrt{n}\epsilon_m$. By the triangle inequality, the inequality $\|\cdot\| \le \|\cdot\|_F$ and the boundedness of $\tilde{k}_{x,m}$ and $\tilde{k}_{z,m}$, we have $\|\tilde{K}_{*x}\| = O(1)$. Both $O(\cdot)$ terms above are independent of $x^*$. Finally, recall that $\|L\| = \|K_{zz}(K_{zz} + \nu I)^{-1}\| \le 1$ and similarly $\|\tilde{L}\| \le 1$. Using these facts, we have
$$\text{(I)} \le \|\tilde{K}_{**} - K_{**}\|_F \le l\epsilon_m \to 0.$$
$$\text{(II)} \le \sqrt{ln}\epsilon_m \cdot 1 \cdot \lambda^{-1} \cdot O(1) \to 0.$$
$$\|\tilde{L} - L\| = \|K_{zz}(K_{zz} + \nu I)^{-1} - \tilde{K}_{zz}(\tilde{K}_{zz} + \nu I)^{-1}\| \to 0.$$
$$\le \|K_{zz} - \tilde{K}_{zz}\| \cdot \nu^{-1} + \|\tilde{K}_{zz}(\tilde{K}_{zz} + \nu I)^{-1}\|\|(K_{zz} - \tilde{K}_{zz})(K_{zz} + \nu I)^{-1}\|$$
$$\le 2\sqrt{n}\epsilon_m \cdot \nu^{-1} \to 0.$$
$$\text{(III)} \le O(1) \cdot \|\tilde{L} - L\| \cdot \lambda^{-1}O(1) \to 0$$
$$\text{(IV)} = O(1) \cdot \|(\lambda I + \tilde{K}_{xx}\tilde{L})^{-1}\|\|\tilde{K}_{xx}\tilde{L} - K_{xx}L\|\|(\lambda I + K_{xx}L)^{-1}\|$$
$$\le O(1) \cdot \lambda^{-2} \cdot (\|\tilde{K}_{xx} - K_{xx}\|\|\tilde{L}\| + \|K_{xx}\|\|\tilde{L} - L\|) \to 0.$$
$$\text{(V)} = O(1) \cdot \sqrt{ln}\epsilon_m \to 0.$$

Moreover, the converges above are all independent of the choice of $x^*$. Thus we have
$$\sup_{x^* \in \mathcal{X}^l} \|\tilde{S} - S\|_F \le l \sup_{x^* \in \mathcal{X}^l} \|\tilde{S} - S\| \to 0.$$

Using a similar argument we have
$$\sup_{x^* \in \mathcal{X}^l} \|\tilde{\mu} - \mu\|_2 \to 0.$$

$\square$

## C.4 Analysis of the Optimization Algorithm

---

**Algorithm 1:** Modified randomized prior algorithm for approximate inference.

---
**Input:** Hyperparameters $\nu, \lambda \in \mathbb{R}$. Random feature models $\theta \mapsto f(\cdot; \theta), \varphi \mapsto g(\cdot; \varphi)$.
**Result:** A single sample from the approximate posterior
Initialize: draw $\theta_0 \sim \mathcal{N}(0, I), \varphi_0 \sim \mathcal{N}(0, \lambda \nu^{-1} I), \tilde{Y} \sim \mathcal{N}(Y, \lambda I)$;
**for** $\ell \leftarrow 1, \ldots, L-1$ **do**
$\quad \hat{\theta}_\ell \leftarrow \theta_{\ell-1} - \eta_\ell \hat{\nabla}_\theta \mathcal{L}_{\mathrm{rf}}(\theta_{\ell-1}, \varphi_{\ell-1}, \theta_0, \varphi_0)$;
$\quad \hat{\varphi}_\ell \leftarrow \varphi_{\ell-1} + \eta_\ell \hat{\nabla}_\varphi \mathcal{L}_{\mathrm{rf}}(\theta_{\ell-1}, \varphi_{\ell-1}, \theta_0, \varphi_0)$;
$\quad \theta_{\ell+1} \leftarrow \mathrm{Proj}_{B_f}(\hat{\theta}_\ell)$;
$\quad \varphi_{\ell+1} \leftarrow \mathrm{Proj}_{B_g}(\hat{\varphi}_\ell)$;
**end**
**return** $f(\cdot; \theta_L)$

---

For the purpose of the analysis we consider the standard SGDA algorithm as outlined in Algorithm 1. In the algorithm $\mathcal{L}_{\mathrm{rf}}$ denotes the objective in (14), and $\mathrm{Proj}_B$ denotes the projection into the $\ell_2$-norm ball with radius $B$, and $\hat{\nabla}\mathcal{L}_{\mathrm{rf}}$ represents a stochastic (unbiased) approximation of the gradient $\nabla \mathcal{L}_{\mathrm{rf}}$. In the following, we will suppress the dependency of $\mathcal{L}_{\mathrm{rf}}$ on $\theta_0, \varphi_0$ for simplicity.

Concretely, we introduce the notations

$$\Phi_f := \frac{1}{\sqrt{m}} \begin{bmatrix} \phi_{x,m}(x_1)^\top \\ \vdots \\ \phi_{x,m}(x_n)^\top \end{bmatrix} \in \mathbb{R}^{n \times m}, \quad \Phi_g := \frac{1}{\sqrt{m}} \begin{bmatrix} \phi_{z,m}(z_1)^\top \\ \vdots \\ \phi_{z,m}(z_n)^\top \end{bmatrix} \in \mathbb{R}^{n \times m},$$

where we recall $X := (x_1, \ldots, x_n)$ and $Z := (z_1, \ldots, z_n)$ are the training data.

Observe that $\Phi_f \theta = f(X; \theta), \Phi_g \varphi = g(Z; \varphi)$, we can rewrite the objective (14) as

$$\mathcal{L}_{\mathrm{rf}}(\theta, \varphi) = \theta^\top \Phi_f^\top \Phi_g \varphi - \tilde{Y}^\top \Phi_g \varphi - \frac{1}{2} \varphi^\top \Phi_g^\top \Phi_g \varphi - \frac{\nu}{2} \|\varphi - \varphi_0\|_2^2 + \frac{\lambda}{2} \|\theta - \theta_0\|_2^2. \tag{62}$$

We additionally define

$$\mathcal{L}_i(\theta, \varphi) = n \left( \theta^\top \Phi_f^\top E_i \Phi_g \varphi - \tilde{Y}^\top E_i \Phi_g \varphi - \frac{1}{2} \varphi^\top \Phi_g^\top E_i \Phi_g \varphi \right) - \frac{\nu}{2} \|\varphi - \varphi_0\|_2^2 + \frac{\lambda}{2} \|\theta - \theta_0\|_2^2,$$

where $E_i := e_i e_i^\top$ and $\{e_i\}_{i \in [n]}$ is the standard orthogonal basis of $\mathbb{R}^n$. We can see that

$$\mathcal{L}_{\mathrm{rf}}(\theta, \varphi) = \frac{1}{n} \sum_{i \in [n]} \mathcal{L}_i(\theta, \varphi).$$

Therefore, the stochastic gradient in Algorithm 1 can be defined as

$$\hat{\nabla}\mathcal{L}_{\mathrm{rf}}(\theta, \varphi) := \nabla \mathcal{L}_\mathcal{I}(\theta, \varphi) = \sum_{i \in [n]} \nabla \mathcal{L}_i(\theta, \varphi) \mathbf{1}_{i = \mathcal{I}}, \tag{63}$$

where $\mathcal{I}$ is a random variable sampled from the uniform distribution of the set $[n]$.

In practice we run the algorithm concurrently on $J$ sets of parameters, starting from independent draws of initial conditions $\{\theta_0^{(j)}, \phi_0^{(j)}\}$; moreover, the projection is not implemented, and there are various other modifications to further improve stability, as described in Appendix D.2.

The following lemma is a convergence theorem of Algorithm 1 under the choice of stochastic gradient defined in (63).

**Lemma C.3.** *Fix an* $m \in \mathbb{N}$. *Denote by* $\theta^*$ *the optima of* (14) *and take* $\eta_\ell := \frac{1}{\mu(\ell+1)}$ *with* $\mu = \min\{\lambda, \nu\}$. *Then for any* $\epsilon, B_1, B_2, B_3 > 0$, *there exist* $B_f, B_g > 0$ *such that when* $L = \Omega(\delta^{-1}\epsilon^{-2})$, *the approximate optima* $\theta_L$ *returned by Algorithm 1 satisfies*

$$\mathbb{P}\left(\{\|\theta_L - \theta^*\|_2 > \epsilon\} \cap E_n\right) \leq \delta,$$

*where*

$$E_n := \left\{ \|\theta_0\|_2 + \|\varphi_0\|_2 \le B_1, \|\tilde{Y}\|_2 \le B_2, \sup_{z \in \mathcal{Z}} \tilde{k}_{z,m}(z,z) + \sup_{x \in \mathcal{X}} \tilde{k}_{x,m}(x,x) \le B_3 \right\},$$

*and $\tilde{k}_{\cdot,m}$ denotes the random feature-approximated kernel. The randomness in the statement above is from the sampling of the initial values $\theta_0, \varphi_0$, the gradient noise.*

*Proof.* Recall from (56) that $\theta^*$ is a sum of bounded linear transforms of $\theta_0, \varphi_0$ and $\tilde{Y}_0$. Thus on the event $E_n$ the norm of the optima $\|\theta^*\|_2$ is bounded. Similarly, $\|\varphi^*\|_2$ is also bounded on $E_n$ by (55). We choose $B_f$ and $B_g$ to be their maximum values on the event $E_n$.

Notice that $\mathcal{L}_{\mathrm{rf}}$ is strongly-convex in $\theta$, and strongly-concave in $\varphi$, so it has the unique stationary point $(\theta^*, \varphi^*)$. We will then bound $\|\theta_\ell - \theta^*\|_2^2 + \|\varphi_\ell - \varphi^*\|_2^2$. Let $\sigma_f, \sigma_g$ be the minimal constants such that $\|\nabla_\theta \mathcal{L}_i(\theta,\varphi)\|_2^2 \le \sigma_f^2$, $\|\nabla_\varphi \mathcal{L}_i(\theta,\varphi)\|_2^2 \le \sigma_g^2$ for all $i \in [n], \|\theta\|_2 \le B_f$ and $\|\varphi\|_2 \le B_g$. Introducing the notation $B := \max\{B_f, B_g\}$, so we have $\|\theta\|_2, \|\varphi\|_2 \le B$. Define

$$r_\ell = \mathbb{E}\left[ \|\theta_\ell - \theta^*\|_2^2 + \|\varphi_\ell - \varphi^*\|_2^2 \right].$$

We want to know how $r_\ell$ contracts. We first make a stochastic gradient step on $\theta_\ell$ with step size $\eta_\ell$, i.e., $\hat{\theta}_{\ell+1} := \theta_\ell - \eta_\ell \hat{\nabla}_\theta \mathcal{L}_{\mathrm{rf}}(\theta_\ell, \varphi_\ell)$ with $\hat{\nabla}\mathcal{L}_{\mathrm{rf}}$ defined in (63). Then,

$$\mathbb{E}[\|\hat{\theta}_{\ell+1} - \theta^*\|_2^2 \mid \theta_\ell, \varphi_\ell] \le \|\theta_\ell - \theta^*\|_2^2 - 2\eta_\ell \langle \theta_\ell - \theta^*, \nabla_\theta \mathcal{L}(\theta_\ell, \varphi_\ell)\rangle + \eta_\ell^2 \sigma_f^2,$$

where the expectation is taken with respect to the randomness of the gradient. For the above inner product term, we have that

$$\langle \theta_\ell - \theta^*, \nabla_\theta \mathcal{L}_{\mathrm{rf}}(\theta_\ell, \varphi_\ell)\rangle = \langle \theta_\ell - \theta^*, \nabla_\theta \mathcal{L}_{\mathrm{rf}}(\theta_\ell, \varphi_\ell) - \nabla_\theta \mathcal{L}_{\mathrm{rf}}(\theta^*, \varphi^*)\rangle$$
$$= \lambda \|\theta_\ell - \theta^*\|_2^2 + \langle \theta_\ell - \theta^*, \Phi_f^\top \Phi_g (\varphi_\ell - \varphi^*)\rangle.$$

Next, we consider the gradient step on $\varphi_\ell$ with step size $\eta_\ell$, i.e., $\hat{\varphi}_{\ell+1} := \varphi_\ell + \eta_\ell \hat{\nabla}_\varphi \mathcal{L}_{\mathrm{rf}}(\theta_\ell, \varphi_\ell)$. Then, we have that

$$\mathbb{E}[\|\hat{\varphi}_{\ell+1} - \varphi^*\|_2^2 \mid \theta_\ell, \varphi_\ell] \le \|\varphi_\ell - \varphi^*\|_2^2 + 2\eta_\ell \langle \varphi_\ell - \varphi^*, \nabla_\varphi \mathcal{L}_{\mathrm{rf}}(\theta_\ell, \varphi_\ell)\rangle + \eta_\ell^2 \sigma_g^2.$$

We similarly deal with the inner product term:

$$\langle \varphi_\ell - \varphi^*, \nabla_\varphi \mathcal{L}_{\mathrm{rf}}(\theta_\ell, \varphi_\ell)\rangle = \langle \varphi_\ell - \varphi^*, \nabla_\varphi \mathcal{L}_{\mathrm{rf}}(\theta_\ell, \varphi_\ell) - \nabla_\varphi \mathcal{L}_{\mathrm{rf}}(\theta^*, \varphi^*)\rangle$$
$$= -\langle \varphi_\ell - \varphi^*, (\Phi_g^\top \Phi_g + \nu I)(\varphi_\ell - \varphi^*)\rangle + \langle \varphi_\ell - \varphi^*, \Phi_g^\top \Phi_f (\theta_\ell - \theta^*)\rangle$$
$$\le -\nu \|\varphi_\ell - \varphi^*\|_2^2 + \langle \varphi_\ell - \varphi^*, \Phi_g^\top \Phi_f (\theta_\ell - \theta^*)\rangle,$$

Combining the above results, we have

$$r_{\ell+1} \le \mathbb{E}[\|\hat{\theta}_{\ell+1} - \theta^*\|_2^2 + \|\hat{\varphi}_{\ell+1} - \varphi^*\|_2^2 \mid \theta_\ell, \varphi_\ell] \le (1 - 2\mu\eta_\ell)r_\ell + \eta_\ell^2(\sigma_f^2 + \sigma_g^2),$$

where we have set $\mu := \min\{\nu, \lambda\}$, and the first inequality follows from the fact that the projection onto a convex set is a contraction map, i.e., $\|\mathrm{Proj}_B(x) - \mathrm{Proj}_B(y)\| \le \|x - y\|$.

Let $\sigma^2 = \sigma_f^2 + \sigma_g^2$ and $\eta_\ell = \frac{\xi}{\ell+1}$ for some $\xi > \frac{1}{2\mu}$, by induction we have

$$r_\ell \le \frac{c_\xi}{\ell+1}, \quad \text{where } c_\xi = \max\left\{ r_0, \frac{2\xi^2 \sigma^2}{2\mu\xi - 1}\right\}.$$

Specifically, taking $\xi = \mu^{-1}$, we have

$$r_\ell \le \frac{1}{\ell+1} \max\left\{ r_0, \frac{2\sigma^2}{\mu^2}\right\}. \tag{64}$$

We now track the constants we have used in (64). Note that on the event $E_n$,

$$r_0 \le 2\left( \|\theta_0\|_2^2 + \|\theta^*\|_2^2 + \|\varphi_0\|_2^2 + \|\varphi^*\|_2^2 \right) \le 4(B_1^2 + B^2).$$

Recall that the definition of $\sigma^2$ is

$$\sigma^2 = \max_{i \in [n], \|\theta\|_2, \|\varphi\|_2 \le B} \|\nabla_\theta \mathcal{L}_i(\theta,\varphi)\|_2^2 + \max_{i \in [n], \|\theta\|_2, \|\varphi\|_2 \le B} \|\nabla_\varphi \mathcal{L}_i(\theta,\varphi)\|_2^2 =: \text{(I)} + \text{(II)}.$$

For the first term, we have

$$
\begin{aligned}
(\mathrm{I}) &= \max_{i \in [n], \|\theta\|_2, \|\varphi\|_2 \leq B} \|\lambda(\theta - \theta_0) + n\Phi_f^\top E_i \Phi_g \varphi\|_2^2 \\
&\leq \max_{i \in [n], \|\theta\|_2, \|\varphi\|_2 \leq B} \left(2\lambda^2 \|\theta - \theta_0\|_2^2 + 2n^2 \|\Phi_f^\top E_i \Phi_g \varphi\|_2^2\right) \\
&\leq 4\lambda^2 (B^2 + B_1^2) + 2n^2 B_3^2 B^2.
\end{aligned}
$$

Similarly, for the second term, we have

$$
\begin{aligned}
(\mathrm{II}) &= \max_{i \in [n], \|\theta\|_2, \|\varphi\|_2 \leq B} \|n(\theta^\top \Phi_f^\top E_i \Phi_g - \tilde{Y}^\top E_i \Phi_g - \Phi_g^\top E_i \Phi_g \varphi) - \nu(\varphi - \varphi_0)\|_2^2 \\
&\leq 4n^2 B_3^2 B^2 + 2n^2 B_2^2 B^2 + 4\nu^2 (B^2 + B_1^2).
\end{aligned}
$$

Thus, we know that

$$
\sigma^2 \leq 8(\lambda^2 + \nu^2)(B^2 + B_1^2) + 6n^2 B_3^2 B^2 + 2n^2 B_2^2 B^2 =: \tilde{C}.
$$

Taking $L_\delta = \delta^{-1} \epsilon^{-2} \max\{4B_1^2 + 4B^2, \tilde{C}\mu^{-1}\}$ and $\eta_\ell = \frac{1}{\mu(\ell+1)}$, by (64), we know that

$$
\mathbb{P}(\|\theta_L - \theta^*\|_2 > \epsilon) \leq \epsilon^{-2} \mathbb{E}\|\theta_L - \theta^*\|_2^2 \leq \epsilon^{-2} r_\ell \leq \delta.
$$

$\square$

## C.5  Proof of Proposition 4.2

By Lemma C.2, for any $\epsilon_1 > 0$ we have

$$
\lim_{m \to \infty} \mathbb{P}\left(\left\{\sup_{x^* \in \mathcal{X}^l} \|\tilde{\mu} - \mu\|_2 > \epsilon_1\right\} \cup \left\{\sup_{x^* \in \mathcal{X}^l} \|\tilde{S} - S\|_F > \epsilon_1\right\}\right) = 0, \tag{65}
$$

where the randomness is from the sampling of random feature bases.

Fix an arbitrary set of $\epsilon_1 > 0, \delta_0 > 0$. Then we can find $m \in \mathbb{N}$ such that the event in (65) has probability smaller than $\delta_0$. Combining Assumption C.2 with the fact that $\theta_0, \phi_0, \tilde{Y}_0$ are now Gaussian random variables with fixed dimensionality, for any $\delta_1 > 0$, we can choose $B_1, B_2, B_3$ such that the event $E_n$ defined in Lemma C.3 has probability $1 - \delta_1$. Thus for any $\epsilon_2 > 0$, when the number of iteration steps exceeds $\Omega(\delta_1^{-1}\epsilon_2^{-2})$, we have

$$
\mathbb{P}(\|\hat{\theta}_m - \theta_m^*\|_2 > \epsilon_2) \leq \mathbb{P}(\{\|\hat{\theta}_m - \theta_m^*\|_2 > \epsilon_2\} \cap E_n) + \mathbb{P}(E_n^c) \leq 2\delta_1, \tag{66}
$$

where $\hat{\theta}_m$ denotes the approximate optima returned by Algorithm 1 after $\Omega(\delta_1^{-1}\epsilon_2^{-2})$ iterations, $\theta_m^*$ denotes the exact optima of the minimax objective, and the randomness is from the gradient noise as well as the perturbations $f_0, g_0, \tilde{Y}$. Thus we have

$$
\mathbb{E}\|\hat{\theta}_m - \theta_m^*\|_2 \leq \epsilon_2 + 2\delta_1(\mathbb{E}\|\hat{\theta}_m\|_2 + \mathbb{E}\|\theta_m^*\|_2) \leq \epsilon_2 + 4\delta_1 B.
$$

From the choice of $B$ in Lemma C.3, we can see that $\delta_1 B \leq \mathbb{E}(\|\theta_m^*\| \cdot (1 - \mathbf{1}_{E_n}))$, and thus converges to 0 as $\delta_1 \to 0$. Therefore, $\mathbb{E}\|\hat{\theta}_m - \theta_m^*\|_2$ converges to 0, and for any $x^* \in \mathcal{X}^l$,

$$
\begin{aligned}
\mathbb{E}\sup_{x^* \in \mathcal{X}^l} \|f(x^*; \hat{\theta}_m) - f(x^*; \theta_m^*)\|_2 &= \mathbb{E}\sup_{x^* \in \mathcal{X}^l} \|\phi_{x,m}(x^*)^\top (\hat{\theta}_m - \theta_m^*)\|_2 \\
&\leq l\sqrt{\tilde{\kappa}} \cdot \mathbb{E}\|\hat{\theta}_m - \theta_m^*\|_2 \to 0,
\end{aligned}
$$

where the expectation is taken with respect to the gradient noise, perturbations, and random feature draws. Hence, the mean and covariance of $f(x^*; \hat{\theta}_m)$ converges to that of $f(x^*; \theta_m^*)$ as intended, and we know that the following holds with probability at least $1 - \delta_0$

$$
\sup_{x^* \in \mathcal{X}^l} \max\left\{\|\mathbb{E}(f(x^*; \hat{\theta}_m)) - \mathbb{E}(f(x^*; \theta_m))\|_2, \|\mathrm{Cov}(f(x^*; \hat{\theta}_m)) - \mathrm{Cov}(f(x^*; \theta_m))\|_F\right\} \leq \epsilon_1
$$

Combining this with (65) completes the proof.

# D    Implementation Details, Experiment Setup and Additional Results

## D.1    Hyperparameter Selection

We follow the strategy in previous work [e.g., 7, 20] and select hyperparameters by minimizing the *observable* first or second stage loss, depending on which part they directly correspond to.

For the first stage, the loss is

$$\mathcal{L}_{v1} = \text{Tr}(K_{xx} - 2K_{x\tilde{x}}L + K_{\tilde{x}\tilde{x}}L^\top L) = \mathbb{E}_{f \sim \mathcal{GP}(0,k)} \| f(X) - Lf(\tilde{X}) \|_2^2$$

where $L := K_{z\tilde{z}}(K_{\tilde{z}\tilde{z}} + \nu I)^{-1}$, and tilde indicates the held-out data. From the above equality we can see that a Monte-Carlo estimator for $L_1$ can be constructed with the following procedure:

   (i). Draw $f \sim \mathcal{GP}(0, k_x)$.
   (ii). Perform kernel ridge regression on the dataset $\{(\tilde{z}_i, f(\tilde{x}_i))\}$.
   (iii). Return the mean squared error on the dataset $(X, Z)$.

This procedure can also be implemented for the NN-based models.

For the second stage, the loss $\sum_{i=1}^n \hat{d}_n(\hat{E}_n f, \hat{b})$ can be computed directly, for both the closed-form quasi-posterior and the random feature approximation. For the approximate inference algorithm, as we can see from (15) that the dual functions $\{g(\cdot; \varphi^{(k)})\}$ are samples from Gaussian process posteriors centered at the needed point estimates $\hat{E}_n f(\cdot; \theta^{(k)})$, instead of the point estimates themselves, we train separate validator models to approximate the latter. The validator models have the architecture to the dual functions used for training, and follow the same learning rate schedule. The validator models are trained before estimating the validation statistics, and we run SGD until convergence to ensure an accurate estimate.

## D.2    Details in the Approximate Inference Algorithm

To draw multiple samples from the quasi-posterior efficiently, our algorithm runs $J$ SGDA chains in parallel, with different perturbations $\{(\tilde{Y}^{(j)}, f_0^{(j)}, g_0^{(j)}) : j \in [J]\}$. While the convergence analysis works with the extremely simple Algorithm 1, in practice we extend it to improve stability and accelerate convergence:

   (i). we employ early stopping based on the validation statistics;
   (ii). before the main optimization loop we initialize the dual parameters at the approximate optima $\arg\min_\varphi L_{\text{rf}}(f^{(j)}, g(\cdot; \varphi))$, by running SGD until convergence;
   (iii). in each SGDA iteration, we use $K_1 > 1$ GD steps on $g$ and one GA step for $f$;
   (iv). after every $K_2$ epochs, we fix $\theta^{(j)}$ and train the dual parameters $\varphi^{(j)}$ for one epoch.

All the above choices are shown to improve the observable validation statistics. We fix $K_1 = 3, K_2 = 2$ which are determined on the 1D datasets using the validation statistics.

## D.3    1D Simulation: Experiment Setup Details

In constructing the datasets, let $\tilde{f}_0$ denote the sine, step, abs or linear ($\tilde{f}_0(x) = x$) function; we then set $f_0 = \tilde{f}_0(4 \cdot (2x - 1))$ if $\tilde{f}_0$ is sine, abs or linear, $\mathbf{1}_{\{2x-1<0\}} + 2.5 \cdot \mathbf{1}_{\{2x-1\geq0\}}$ otherwise. These choices are made to maintain similarity with previous work [5, 6], which used the same transformed step function and defined $\mathbf{x}$ so that it has a range of approximately $[-4, 4]$.

For 2SLS and the kernelized IV methods, we determine $\lambda$ and $\nu$ following D.1. To improve stability, we repeat the procedures on 50 random partitions of the combined training and validation set, and choose the hyperparameters that minimize the average loss. The hyperparameters are chosen from a log-linearly scaled grid consisting of 10 values in the range of $[0.1, 30]$. We note that the occasional instability of hyperparameter selection is also reported in [20]. For BayesIV, we run the MCMC sampler for 25000 iterations, discard the first 5000 iterations for burn in, and take one sample out of every 80 consecutive iterations to construct the approximate posterior. For bootstrap we use 20

samples. In both cases we verify that further increasing the computational budget does not improve the final performance.

We normalize the dataset to have zero mean and unit variance. For all kernel methods we set the kernel bandwidth using the median trick.

Finally, we provide details in the run time measurement in Table 1: BayesIV is evaluated on a machine with an i9-9900k processor (8 cores, 16 threads; 5.0 GHz) and 64GB RAM; the closed-form quasi-posterior is evaluated on a machine with two Xeon E5-2620v4 processors (total 16 cores, 32 threads; 3.0 GHz) and 220GB RAM; the approximate inference method is evaluated on the same machine, with a GeForce GTX 1080Ti GPU. The use of different machines is because BayesIV requires a Windows environment; it should not put BayesIV into disadvantage, as it only makes efficient use of 4 CPU cores. For all methods, the reported runtime excludes non-computational tasks such as data preparation and JIT compilation. For the approximate inference algorithm, we report the runtime for the optimal hyperparameters; runtime for suboptimal hyperparameters is typically lower due to early stopping. As a single run of the algorithm does not fully utilize the GPU, we run 6 experiments in parallel and report the elapsed time divided by 6. This is a realistic evaluation setting, since in practice all methods require multiple runs for hyperparameter selection, and will benefit from parallelization whenever possible.

### D.4   1D Simulation: Full Results and Visualizations

Full results are reported in Table 3-4. As we can see, the gap in CI coverage between bootstrap and the quasi-posterior consistently appears across all datasets, and is most evident in the small-sample setting or when Matérn kernels are used instead of the RBF kernel.

We provide the following visualizations:

(i). We visualize the quasi-posterior and the bootstrap predictive distribution on all datasets, using the nonparametric kernel that best matches the smoothness of the target function. This amounts to Matérn-$3/2$ for abs and step, and RBF for sin and linear.[11] Results for $\alpha = 0.5$ are plotted in Figure 3, and $\alpha = 0.05$ in Figure 4. We can see that

  - The credible intervals produced by our method shrink when $N$ or $\alpha$ increases, correctly reflecting the increased amount of available information in training data. Their width also has the same order of magnitude as bootstrap, when $\alpha = 0.5$ (i.e., when bootstrap is more reliable).
  - When the instrument strength is weak ($\alpha = 0.05$), our method is significantly more robust than bootstrap, especially when the sample size is smaller.
  - On the step dataset where Assumption 3.2 is violated, our method still provides good coverage.

(ii). We plot the quasi-posterior using over-smoothed kernels on the abs dataset, which include the RBF kernel and the Matérn-$5/2$ kernel, in Figure 5 (b-c).

  - We can see that both kernels produce CIs with good coverage, and the CIs have similar (albeit slightly smaller) width comparing with the Matérn-$3/2$ kernel. This is consistent with previous results on GP regression using oversmoothed priors [62]; the slight shrink in CI width could be attributed to the fact that the abs function is smoother than $C^0$ in most regions.

(iii). We plot the approximate quasi-posterior using the approximate inference algorithm in Figure 5 (d).[12]  Comparing Figure 5 (c) and (d), we can see that the approximate and exact quasi-posterior are visually similar.

---

[11]None of the kernels match the discontinuous step function, so we use the least smooth one; for the linear function, we skip the linear kernel, since numerical study of quasi-posteriors using low-dimensional parametric models exists in literature [16].

[12]We use 400 random Fourier feature basis to approximate the RBF kernel. Regularization hyperparameters are determined using the closed-form validation statistics, and optimization hyperparamaters are determined by grid search following the setting of the lower-dimensional demand experiment below.

| Method | bayesIV | bs-lin | qb-lin | bs-poly | qb-poly | bs-ma3 | qb-ma3 | bs-ma5 | qb-ma5 | bs-rbf | qb-rbf |
|---|---|---|---|---|---|---|---|---|---|---|---|
| $f_0$ = sin, $N$ = 200, $\alpha$ = 0.5 | | | | | | | | | | | |
| MSE | .024 (.038) | .111 (.011) | .109 (.010) | .243 (.037) | .243 (.034) | .023 (.010) | .025 (.014) | .022 (.011) | .021 (.015) | .025 (.013) | .026 (.015) |
| CI Cvg. | .895 (.252) | .232 (.039) | .110 (.019) | .077 (.032) | .045 (.017) | .965 (.065) | 1.00 (.000) | .972 (.065) | 1.00 (.000) | .972 (.079) | 1.00 (.013) |
| CI Wid. | .188 (.035) | .143 (.028) | .072 (.004) | .078 (.025) | .041 (.006) | .283 (.031) | .661 (.066) | .288 (.032) | .569 (.067) | .293 (.040) | .408 (.063) |
| $f_0$ = sin, $N$ = 1000, $\alpha$ = 0.5 | | | | | | | | | | | |
| MSE | .016 (.003) | .103 (.006) | .103 (.006) | .237 (.020) | .239 (.020) | .009 (.011) | .008 (.014) | .008 (.012) | .007 (.015) | .007 (.012) | .007 (.013) |
| CI Cvg. | .598 (.155) | .097 (.020) | .049 (.006) | .036 (.012) | .017 (.004) | .962 (.113) | 1.00 (.000) | .954 (.111) | 1.00 (.000) | .957 (.168) | 1.00 (.036) |
| CI Wid. | .085 (.006) | .061 (.011) | .032 (.001) | .038 (.009) | .019 (.002) | .186 (.032) | .602 (.037) | .173 (.029) | .509 (.037) | .164 (.029) | .326 (.041) |
| $f_0$ = abs, $N$ = 200, $\alpha$ = 0.5 | | | | | | | | | | | |
| MSE | .042 (.038) | .454 (.052) | .456 (.053) | .478 (.085) | .477 (.089) | .033 (.026) | .035 (.025) | .032 (.024) | .031 (.025) | .031 (.019) | .031 (.021) |
| CI Cvg. | .863 (.184) | .190 (.039) | .085 (.198) | .055 (.027) | .020 (.072) | .945 (.110) | 1.00 (.004) | .942 (.118) | 1.00 (.004) | .920 (.125) | 1.00 (.030) |
| CI Wid. | .207 (.027) | .214 (.035) | .077 (.220) | .110 (.038) | .043 (.091) | .316 (.037) | .676 (.072) | .317 (.034) | .599 (.079) | .277 (.037) | .462 (.082) |
| $f_0$ = abs, $N$ = 1000, $\alpha$ = 0.5 | | | | | | | | | | | |
| MSE | .024 (.011) | .448 (.016) | .449 (.016) | .468 (.025) | .469 (.026) | .020 (.006) | .019 (.005) | .019 (.006) | .018 (.006) | .017 (.007) | .016 (.006) |
| CI Cvg. | .507 (.196) | .083 (.018) | .111 (.092) | .028 (.007) | .009 (.003) | .857 (.075) | 1.00 (.000) | .823 (.079) | 1.00 (.000) | .829 (.102) | 1.00 (.016) |
| CI Wid. | .092 (.005) | .098 (.019) | .127 (.103) | .061 (.016) | .019 (.002) | .181 (.024) | .646 (.050) | .174 (.026) | .530 (.056) | .168 (.024) | .383 (.061) |
| $f_0$ = step, $N$ = 200, $\alpha$ = 0.5 | | | | | | | | | | | |
| MSE | .045 (.026) | .075 (.010) | .075 (.010) | .179 (.025) | .180 (.023) | .041 (.013) | .047 (.017) | .043 (.012) | .046 (.015) | .046 (.011) | .048 (.013) |
| CI Cvg. | .845 (.176) | .347 (.069) | .220 (.045) | .110 (.048) | .067 (.022) | .797 (.101) | 1.00 (.022) | .787 (.085) | .998 (.038) | .710 (.085) | .917 (.051) |
| CI Wid. | .194 (.018) | .139 (.023) | .072 (.004) | .068 (.022) | .041 (.006) | .300 (.047) | .665 (.083) | .285 (.041) | .593 (.082) | .252 (.035) | .453 (.061) |
| $f_0$ = step, $N$ = 1000, $\alpha$ = 0.5 | | | | | | | | | | | |
| MSE | .023 (.009) | .070 (.004) | .069 (.004) | .178 (.012) | .176 (.011) | .035 (.012) | .038 (.015) | .038 (.011) | .040 (.014) | .039 (.012) | .040 (.014) |
| CI Cvg. | .616 (.116) | .185 (.042) | .098 (.014) | .053 (.021) | .030 (.005) | .784 (.153) | 1.00 (.016) | .739 (.164) | .976 (.028) | .661 (.134) | .839 (.077) |
| CI Wid. | .086 (.004) | .060 (.011) | .032 (.001) | .032 (.010) | .019 (.002) | .206 (.062) | .565 (.053) | .196 (.073) | .483 (.065) | .168 (.031) | .312 (.054) |
| $f_0$ = linear, $N$ = 200, $\alpha$ = 0.5 | | | | | | | | | | | |
| MSE | .009 (.011) | .002 (.002) | .001 (.002) | .128 (.019) | .128 (.017) | .012 (.008) | .017 (.013) | .011 (.009) | .014 (.014) | .011 (.011) | .013 (.016) |
| CI Cvg. | .948 (.091) | 1.00 (.153) | 1.00 (.130) | .087 (.026) | .060 (.018) | .995 (.044) | 1.00 (.001) | .995 (.050) | 1.00 (.003) | .990 (.060) | 1.00 (.043) |
| CI Wid. | .129 (.025) | .088 (.012) | .072 (.004) | .055 (.017) | .041 (.006) | .269 (.031) | .520 (.049) | .261 (.031) | .438 (.058) | .239 (.034) | .298 (.041) |
| $f_0$ = linear, $N$ = 1000, $\alpha$ = 0.5 | | | | | | | | | | | |
| MSE | .005 (.002) | .000 (.001) | .000 (.001) | .121 (.012) | .121 (.012) | .006 (.004) | .007 (.005) | .006 (.003) | .007 (.005) | .006 (.003) | .005 (.004) |
| CI Cvg. | .626 (.130) | 1.00 (.174) | 1.00 (.230) | .034 (.008) | .026 (.004) | .992 (.094) | 1.00 (.000) | .990 (.095) | 1.00 (.000) | .975 (.103) | 1.00 (.000) |
| CI Wid. | .051 (.002) | .039 (.006) | .032 (.001) | .026 (.006) | .019 (.002) | .171 (.031) | .508 (.043) | .156 (.032) | .418 (.044) | .135 (.024) | .242 (.043) |

Table 3: Full results in the 1D simulation, for $\alpha = 0.5$

| Method | bayesIV | bs-lin | qb-lin | bs-poly | qb-poly | bs-ma3 | qb-ma3 | bs-ma5 | qb-ma5 | bs-rbf | qb-rbf |
|---|---|---|---|---|---|---|---|---|---|---|---|
| $f_0$ = sin, $N$ = 200, $\alpha$ = 0.05 | | | | | | | | | | | |
| MSE | .275 (.045) | .133 (.070) | .193 (.125) | .202 (.121) | .215 (.161) | .231 (.037) | .190 (.068) | .209 (.037) | .163 (.073) | .183 (.047) | .142 (.081) |
| CI Cvg. | .165 (.077) | .992 (.080) | 1.00 (.134) | .325 (.092) | .425 (.080) | .270 (.082) | .960 (.084) | .332 (.121) | .955 (.086) | .468 (.219) | .952 (.134) |
| CI Wid. | .192 (.030) | .589 (.166) | .971 (.481) | .394 (.171) | .771 (.265) | .192 (.036) | .712 (.045) | .233 (.054) | .694 (.065) | .297 (.085) | .638 (.105) |
| $f_0$ = sin, $N$ = 1000, $\alpha$ = 0.05 | | | | | | | | | | | |
| MSE | .146 (.025) | .123 (.077) | .169 (.315) | .213 (.145) | .228 (.192) | .246 (.071) | .216 (.113) | .238 (.085) | .214 (.131) | .188 (.096) | .188 (.131) |
| CI Cvg. | .082 (.060) | .880 (.296) | .888 (.346) | .289 (.086) | .344 (.135) | .373 (.111) | .819 (.131) | .436 (.145) | .840 (.185) | .562 (.218) | .897 (.212) |
| CI Wid. | .095 (.018) | .552 (.367) | .852 (.568) | .405 (.294) | .588 (.400) | .254 (.036) | .605 (.049) | .298 (.042) | .568 (.050) | .373 (.086) | .536 (.055) |
| $f_0$ = abs, $N$ = 200, $\alpha$ = 0.05 | | | | | | | | | | | |
| MSE | .806 (.478) | .487 (.259) | .500 (.505) | .489 (.233) | .472 (.453) | .392 (.064) | .349 (.156) | .368 (.074) | .350 (.180) | .336 (.094) | .393 (.221) |
| CI Cvg. | .122 (.197) | .435 (.167) | .775 (.201) | .247 (.119) | .545 (.122) | .217 (.102) | .795 (.137) | .287 (.130) | .832 (.163) | .352 (.243) | .805 (.250) |
| CI Wid. | .239 (.049) | .526 (.311) | 1.30 (.473) | .303 (.152) | .895 (.279) | .294 (.047) | .712 (.045) | .351 (.065) | .694 (.065) | .424 (.103) | .638 (.105) |
| $f_0$ = abs, $N$ = 1000, $\alpha$ = 0.05 | | | | | | | | | | | |
| MSE | 1.45 (.250) | .479 (.105) | .500 (.083) | .472 (.159) | .472 (.111) | .376 (.090) | .390 (.144) | .374 (.109) | .367 (.181) | .304 (.134) | .265 (.214) |
| CI Cvg. | .019 (.014) | .464 (.179) | .742 (.249) | .332 (.165) | .562 (.191) | .306 (.109) | .665 (.207) | .374 (.165) | .667 (.248) | .505 (.269) | .625 (.308) |
| CI Wid. | .139 (.017) | .460 (.328) | 1.24 (.576) | .340 (.238) | .923 (.384) | .339 (.044) | .605 (.049) | .367 (.056) | .561 (.049) | .504 (.114) | .536 (.061) |
| $f_0$ = step, $N$ = 200, $\alpha$ = 0.05 | | | | | | | | | | | |
| MSE | .226 (.127) | .105 (.069) | .148 (.199) | .157 (.092) | .192 (.148) | .214 (.036) | .183 (.070) | .194 (.038) | .176 (.077) | .160 (.056) | .147 (.090) |
| CI Cvg. | .193 (.090) | .777 (.158) | .787 (.197) | .382 (.079) | .438 (.066) | .262 (.103) | .952 (.068) | .300 (.158) | .920 (.089) | .432 (.282) | .890 (.149) |
| CI Wid. | .184 (.032) | .621 (.297) | .767 (.452) | .456 (.179) | .652 (.292) | .262 (.051) | .712 (.045) | .310 (.070) | .694 (.065) | .365 (.103) | .638 (.112) |
| $f_0$ = step, $N$ = 1000, $\alpha$ = 0.05 | | | | | | | | | | | |
| MSE | .079 (.044) | .083 (.044) | .131 (.207) | .152 (.105) | .236 (.120) | .231 (.059) | .214 (.107) | .224 (.070) | .208 (.127) | .192 (.082) | .170 (.130) |
| CI Cvg. | .149 (.231) | .715 (.171) | .696 (.201) | .366 (.087) | .390 (.120) | .290 (.075) | .841 (.165) | .353 (.144) | .853 (.212) | .515 (.229) | .800 (.189) |
| CI Wid. | .125 (.021) | .544 (.295) | .815 (.564) | .408 (.187) | .539 (.423) | .298 (.037) | .605 (.049) | .330 (.052) | .561 (.049) | .420 (.095) | .543 (.065) |
| $f_0$ = linear, $N$ = 200, $\alpha$ = 0.05 | | | | | | | | | | | |
| MSE | .083 (.023) | .013 (.032) | .041 (.202) | .103 (.065) | .151 (.149) | .052 (.015) | .046 (.024) | .046 (.016) | .045 (.023) | .046 (.020) | .053 (.023) |
| CI Cvg. | .238 (.133) | 1.00 (.284) | 1.00 (.172) | .372 (.099) | .412 (.109) | .490 (.194) | 1.00 (.000) | .750 (.219) | 1.00 (.000) | .895 (.188) | 1.00 (.026) |
| CI Wid. | .121 (.030) | .559 (.213) | .595 (.406) | .495 (.181) | .543 (.327) | .220 (.028) | .712 (.045) | .270 (.041) | .694 (.065) | .323 (.074) | .638 (.105) |
| $f_0$ = linear, $N$ = 1000, $\alpha$ = 0.05 | | | | | | | | | | | |
| MSE | .051 (.007) | .018 (.034) | .022 (.165) | .125 (.092) | .188 (.411) | .070 (.023) | .069 (.041) | .070 (.028) | .070 (.047) | .057 (.031) | .066 (.043) |
| CI Cvg. | .094 (.041) | 1.00 (.215) | 1.00 (.200) | .338 (.113) | .298 (.122) | .455 (.107) | 1.00 (.028) | .528 (.192) | 1.00 (.066) | .684 (.237) | 1.00 (.096) |
| CI Wid. | .058 (.007) | .535 (.289) | .423 (.470) | .435 (.309) | .406 (.365) | .205 (.020) | .605 (.049) | .230 (.033) | .561 (.049) | .297 (.089) | .530 (.061) |

Table 4: Full results in the 1D simulation, for $\alpha = 0.05$

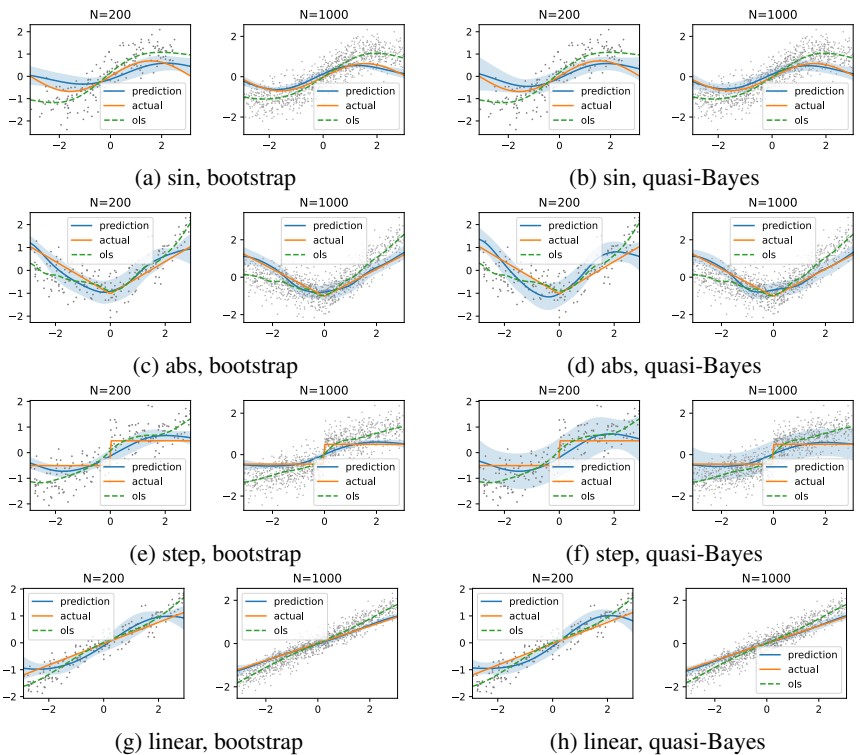

Figure 3: 1D datasets: visualizations of predictive distribution with $\alpha = 0.5$. Dot indicates the training data, and "ols" indicates biased regression predictions using KRR. Shade indicates $95\%$ CI.

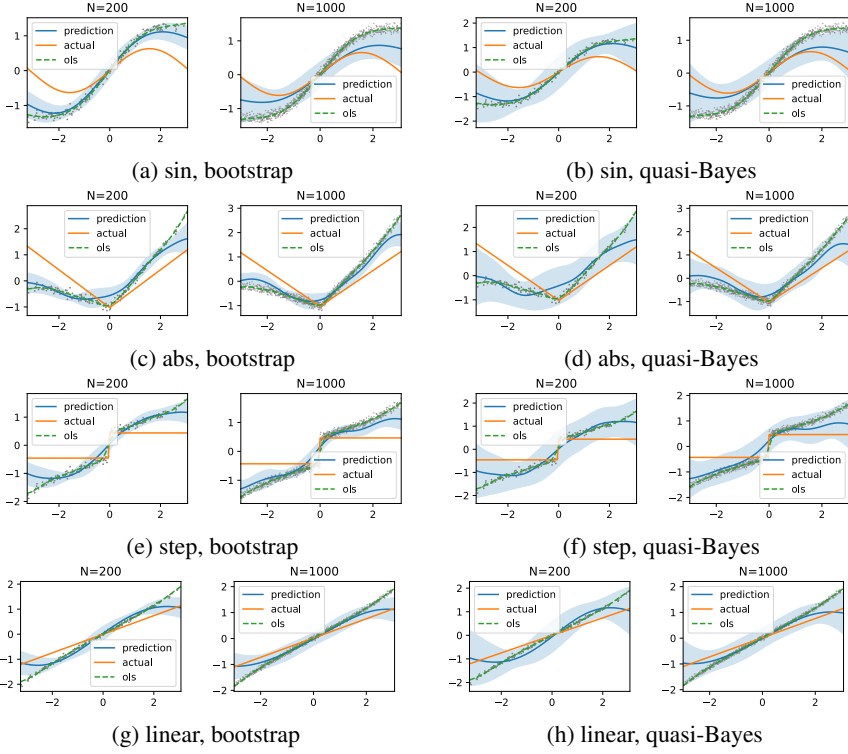

Figure 4: 1D datasets: visualizations of predictive distribution with $\alpha = 0.05$. Best viewed when zoomed. Due to the hyperparameter selection procedure, the CIs do not always shrink as $N$ increases.

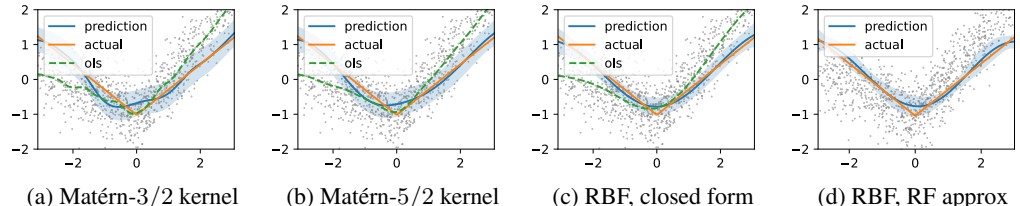

| (a) Matérn-3/2 kernel | (b) Matérn-5/2 kernel | (c) RBF, closed form | (d) RBF, RF approx |

Figure 5: 1D datasets: visualization of the quasi-posterior on the abs dataset using various models. We fix $N = 1000, \alpha = 0.5$.

## D.5 Demand Simulation: Experiment Setup Details

All variables in the dataset are normalized to have zero mean and unit variance. For BayesIV, we run the MCMC sampler for 50000 iterations, discard the first 10000 samples as burn in, and take every 80th sample for inference. For the kernelized methods, hyperparamater selection follows the 1D experiments. For the NN-based methods, implementation details are discussed in Appendix D.2; for both our method and bootstrap, we draw $J = 10$ samples from the predictive distribution.

We select hyperparameters by applying the procedure in Appendix D.1 to a fixed train / validation split, since on this dataset we observe little variation in its results. Hyperparameters include $\lambda, \nu$, and the learning rate schedule (initial learning rate $\eta_0$ and period of learning rate decay $\tau$). The learning rate is adjusted by multiplying it by a factor of $0.8$ every $\tau$ iterations. We fix the optimizer to Adam, and train until validation statistics no longer improves.

For the lower-dimensional setup, we select $\lambda$ and $\nu$ from a log-linearly scaled grid of 10 values, with the range of $[5 \times 10^{-3}, 5]$ and $[0.05, 1]$, respectively. The ranges are chosen based on preliminary experiments using the range of $[0.1, 30]$. We determine $\eta$ from $\{5 \times 10^{-4}, 10^{-3}, 5 \times 10^{-3}, 1 \times 10^{-2}, 5 \times 10^{-2}\}$, and $\tau$ from $\{80, 160, 320, 640\}$. We fix the batch size at $256$. The NN architecture consists of two fully-connected layers, with 50 hidden units and the tanh activation. We also experimented with NNs with 3 hidden layers or with ReLU activation, and made this choice based on the validation statistics.

For the image-based setup, the range of $\lambda$ and $\eta$ follows the above. For $\nu, \tau$ we consider $\nu \in [1, 100], \tau \in \{640, 1280, 2560, 5120\}$, based on preliminary experiments. We fix the batch size at 80. The network architecture is adapted from [4], and consists of two $3 \times 3$ convolutional layers with $64$ filters, followed by max pooling, dropout, and three fully-connected layers with $64, 32$ and $1$ units.

Following the setup in all previous work, we use a uniform grid on $[5, 30] \times [0, 10] \times \{0, \dots, 6\}$ as the test set.

**Computational cost** We report the typical training time for a single set of hyperparameters, excluding JIT compilation time, on a GeForce GTX 1080Ti GPU. In the lower-dimensional experiments, training takes around 25 minutes for a single set of hyperparameters when $N = 10^3$, or around 30 minutes when $N = 10^4$; in both cases 6 experiments can be carried out in parallel on a single GPU. In the image experiment, training takes around 7.5 hours.

The time cost reported above is for the optimal hyperparameter configuration; experiments using suboptimal hyperparameters usually take a shorter period of time due to early stopping. It can also be improved by switching to low-precision numerical operations, or with various heuristics in the hyperparameter search (e.g., using a smaller $J$ in an initial search).

## D.6 Demand Simulation: Full Results and Visualizations

Results in the large-sample settings are presented in Table 5. We only include 2SLS for comparison, since the time complexity of the other baselines is too high. The results are consistent with the discussion in the main text.

We plot the predictive distributions for all methods in Figure 7, on the same cross-section as in the main text, for $N = 1000$. (We omit the plot for $N = 10^4$ and the image experiment, since in

those settings bootstrap and the quasi-posterior have similar behaviors.) As we can see, all non-NN baselines except BayesIV produce overly smooth predictions, presumably due to the lack of flexibility in these models. Note that the visualizations only correspond to an intersection of the true function $f(x_0, t_0, s)$, with $x_0, t_0$ fixed; the complete function has the form of $x \cdot s \cdot \psi(t)$, ignoring the less significant terms, and thus may incur a large norm penalty in the less flexible RKHSes. The issue is further exacerbated by the discrepancy between the training and test distributions: the former is non-uniform due to confounding. As we can see from Figure 6, in the region where $t$ is close to 5, the data is scarce for most values of $x$, which may explain the reason that the RBF-based methods fail to provide good coverage around $t = 5$ (and $s = 3, x = 17.5$, as used in the visualizations), and the reason that both NN-based methods assign higher uncertainty around this location.

BayesIV has a different failure mode: as it employs additive regression models for both stages $p(\mathbf{x} \mid \mathbf{z}), p(\mathbf{y} \mid \mathbf{x})$, it approximates this cross-section relatively well. However, as the true structural function does not have an additive decomposition, its prediction in other regions can be grossly inaccurate; we plot one such cross-section in Figure 8(a).

When implemented with the NN model, bootstrap CIs are more optimistic in regions with more training data, although the difference is often insignificant. The difference in out-of-distribution regions is more significant, where bootstrap is often less robust. An example is provided in Figure 8.

Table 5: Deferred results on the demand design. Results are averaged over 20 trials for the low-dimensional experiment, and 10 trials for the image experiment.

| Setting | Low-dimensional, $N = 10^4$ | | | Image, $N = 5 \times 10^4$ | | |
|---|---|---|---|---|---|---|
| Method | BS-2SLS | BS-NN | QB-NN | BS-2SLS | BS-NN | QB-NN |
| NMSE | $.371 \pm .003$ | $.014 \pm .003$ | $.020 \pm .002$ | $.559 \pm .008$ | $.168 \pm .027$ | $.138 \pm .037$ |
| CI Cvg. | $.024 \pm .005$ | $.944 \pm .009$ | $.957 \pm .008$ | $.112 \pm .005$ | $.892 \pm .022$ | $.909 \pm .017$ |
| CI Wid. | $.014 \pm .002$ | $.136 \pm .015$ | $.203 \pm .013$ | $.132 \pm .039$ | $.636 \pm .027$ | $.597 \pm .024$ |

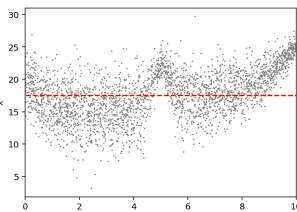

Figure 6: Demand experiment: scatter plot of $10^4$ samples from the training data distribution $p(x, t \mid s = 4)$. The dashed line indicates the cross-section used in Figure 2.

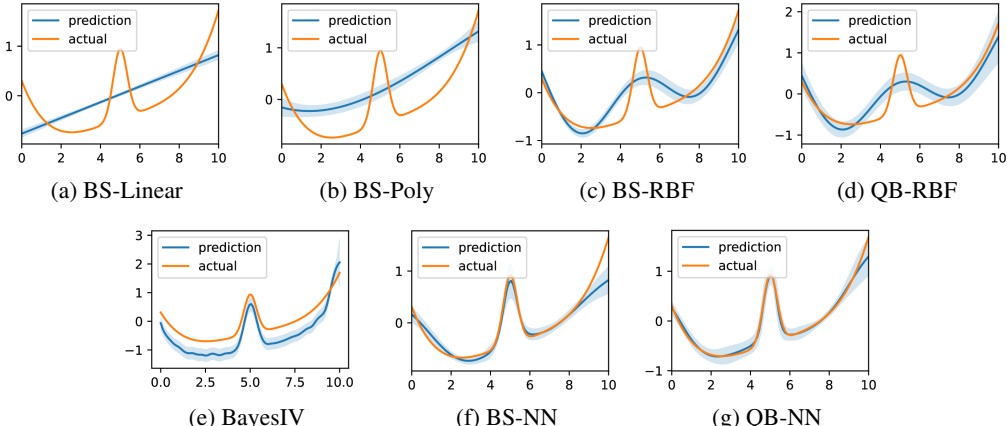

Figure 7: Demand experiment: visualizations of the predictive distributions for $N = 1000$, on the same cross-section as in Figure 2.

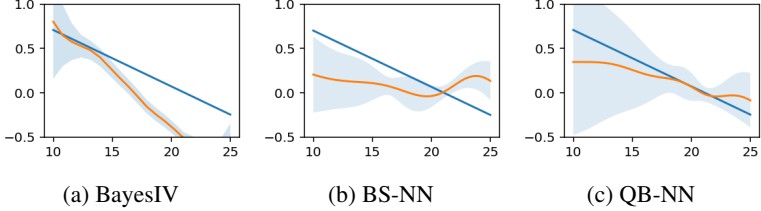

Figure 8: Demand experiment: visualizations of the predictive distributions for $N = 1000$ on a out-of-distribution cross-section, obtained by fixing $t = 9, s = 6$ and varying $x$.