# OpenReview forum: "Scalable Quasi-Bayesian Inference for Instrumental Variable Regression"
_NeurIPS.cc/2021/Conference — NeurIPS 2021 Poster_

### Official Review · Reviewer_6kaq · 2021-07-08

**Rating:** 7
**Confidence:** 4

**Summary:**

The paper proposes a scalable quasi-Bayesian inference for instrumental variable regression. Specifically, the paper builds a quasi-posterior on the recent quadratic kernel IV loss function. For scalability, the paper extends the randomized prior trick based on the recent dual formulation of the quadratic loss. The approach has a theoretical guarantee that incorrect solutions are excluded while valid kept. Moreover, the convergence analysis for the randomized prior trick with the stochastic gradient descent-ascent is present. Experiments validate the approach.

**Limitations And Societal Impact:**

Yes.

**Main Review:**

1. Originality

The quasi-posterior Bayesian approach employs the recent development of non-linear instrumental variable regression and the paper extends the randomized prior trick for scalability of the model. However, a recent method [Zhang, 2020] similarly connects the quadratic kernel loss with the Gaussian process and this work is expected to clarify the relation to it.

2. Quality

The approach is technically sound. The paper presents assumptions for theoretical analyses and detailed development for the approach. The theoretical analyses are sound and provides performance guarantees. The experiments assess the scalability and the modeling ability of the approach.

3. Clarity

Yes, the paper is clearly written.

4. Significance

The paper presents important results and will be interesting for researchers in the Bayesian community as well as the causality inference community.

Zhang, R., Imaizumi, M., Schölkopf, B., & Muandet, K. (2020). Maximum moment restriction for instrumental variable regression. arXiv preprint arXiv:2010.07684.

**Time Spent Reviewing:**

8

---

> ### Author Response · Authors · 2021-08-09
> **Response**
>
>
> Thank you for your positive and constructive feedback. Below we address your question on the relation of our work to the GP discussion in Zhang et al (2020). A discussion will also be included in the revision.
>
> Zhang et al (2020) used a GP formulation to provide an elegant derivation for the leave-$m$-out cross validation error.  Like our method, their GP formulation is also not strictly Bayesian in nature [\^1], and thus frequentist properties of the GP (quasi-)posterior need to be separately established before using it for prediction or inference.  However, as the goal of Zhang et al (2020) is to develop a novel point estimator, they did not establish such results, nor did they address the problem of scalable approximate inference.  Our work address these two problems, starting from the different point estimator in [7, 21].  It remains interesting future work to address these problems for the quasi-posterior formulated in Zhang et al (2020).
>
> [\^1]: To see why the data generating process is not truly Bayesian, recall that in IV we assume the data points $\{(x_i,y_i,z_i): i=1,\ldots,n\}$ are i.i.d. samples frm the joint distribution $p_{data}(dx\times dy\times dz)$.  Consequently, for two data ponts $(x_1,y_1,z_1),(x_2,y_2,z_2)$, it must hold that $p(y_1,y_2\mid x_1,x_2,z_1,z_2,f)=p(y_1\mid x_1,z_1,f)p(y_2\mid x_2,z_2,f)$.  However, one could verify that under their (fictitious) data generating process this factorization does not hold.

---

> > ### Comment · Reviewer_6kaq · 2021-08-11
> > **Response to rebuttal**
> >
> > I read through other reviewers' comments and I don't have their concerns. The quasi-Bayesian ideas had been explored while the connection to the recent development of instrumental variable regression is new. The memory and time complexities are always intrinsic problems with Gaussian processes and solving them is difficult. The theorems are helpful and necessary in the setting of instrumental variable regression. The contents are clear and well-organized.
> >
> > I also agree with the authors that the GP formulation in Zhang et al. (2020) employs quasi-likelihood.
> >
> > **I would also add some new comments to help to improve the future version but won't change my decision.** It would be great if the authors could summarize recent related quasi-Bayesian and Bayesian works in their new version, **which is a major weakness in my opinion but isn't hard to fix**. There is clearly a gap in related works to the Bayesian estimation in this submission and this won't help followers to get familiar with this direction. Some related works:
> >
> > Siddhartha Chib, Minchul Shin, Anna Simoni (2019). Bayesian Estimation and Comparison of Conditional Moment Models. (https://apps.olin.wustl.edu/faculty/chib/css2.pdf)
> >
> > Florens, J.-P. and Simoni, A. (2019), ‘Gaussian processes and Bayesian moment estimation’, Journal of
> > Business & Economic Statistics. published online October 25, 2019.

---

> > > ### Author Response · Authors · 2021-08-11
> > > **Thanks for the comments and references**
> > >
> > > Thank you for your valuable suggestion and references.  Our related work section had focused on (quasi-)Bayes methods with application to nonparametric IV, a conditional moment model where both the dimensionality of the parameter of interest ($f$) and the number of moment constraints are infinite.  We agree that adding discussion about Bayesian and Bayes-like methods for a broader range of conditional moment models will help to put our work in perspective.
> > >
> > > Regarding the references you provided, in both papers the parameter of interest is finite-dimensional.  Chib et al (2019) works with the different framework of exponentially-tilted empirical likelihood; scalable approximate inference appears nontrivial due to the need to estimate a parameter-dependent tilting vector.  Florens and Simoni (2019) studies unconditional moment models which only include IV regression when the conditional expectation operator $E$ is finite-dimensional; extension appears nontrivial as their data-dependent prior would require fine knowledge about $E$ (Section 2.1.2).  Interestingly, Florens and Simoni (2019) noted that the quasi-posterior in our and previous work connects to a certain non-informative limit of Bayesian posteriors, although conceptual subtleties exists (Remark 2.1; also there are differences in regularization).  A detailed discussion about these and related papers will be included in the revision.

---

### Official Review · Reviewer_rhpF · 2021-07-13

**Rating:** 7
**Confidence:** 2

**Summary:**

The authors introduce a scalable quasi-Bayesian method for nonparametric instrumental variable (IV) regression that provides uncertainty quantification for the IV models. They base their approach on kernelized IV models and define a quasi-posterior as the Radon-Nikodym derivative of a generalized method of moments (GMM) objective function with respect to the standard Gaussian process prior. They go on to show that the quasi-posterior exhibits satisfies certain posterior contraction criteria that lead to asymptotically sound results, and prove a conservative posterior contraction rate under a set of typical assumptions. They then show how to achieve scalable inference for the quasi-posterior using a previously proposed “randomized prior trick” for Gaussian process regression, and provide a proof for the convergence of the inference algorithm. Finally, the authors showcase the performance of their method in a series of simulated experiments of varying complexity, by comparing it against other approaches that provide uncertainty quantification (Bayesian, bootstrap).

**Limitations And Societal Impact:**

The authors have discussed the potential impact of their work in a separate section, where they explain how uncertainty quantification can help in assessing the degree of error in the estimate, but warn the practitioner that its validity ultimately hinges on the chosen model.

**Main Review:**

*Originality*: The present work is, to the best of my knowledge, novel. To arrive at this contribution, the authors have combined and built upon previously published results on kernelized IV regression and quasi-Bayesian analysis. The authors carefully put their method in the context of related work and adequately describe how the contribution differs.

*Quality*: The submission seems to be technically sound, and the theoretical claims are reasonably supported in the main paper, with the majority of proofs deferred to the supplement. The numerical experiments are also appropriate, with the authors providing at least one challenging example for the task under hand. However, one aspect that is missing in the numerical experiments’ evaluation is the computational cost. The authors do discuss the time and computational complexity of their method in more detail in the supplement, but since scalability is one of the main selling-points of the method, I would have liked to have seen a comparison in terms of computational cost and/or run time.

*Clarity*: The paper is well-organized for the most part and is relatively error-free. Many technical details are left for the supplement, but the key ideas all seem to be present.

*Significance*: Judging by the numerical results, the authors’ submission successfully addresses a very challenging task and equals, or improves upon, previous results in terms of estimation accuracy and uncertainty quantification. However, it seems to me that the major advantage of this quasi-Bayesian approach for nonparametric IV regression could lie in its scalability, as the use of such flexible methods is often computationally prohibitive. I think the authors could have stressed this point more to underline the significance of their work.

*Minor comments*:
-	Line 13: “causal effect” -> “causal effects”
-	Line 38: It seems that a verbal phrase like “are” or “need be” is missing before “present”.
-	Line 90: $\bar{\nu}$ is first used here in defining $ \hat{C}_{zz, \bar{\nu}}^{-1}$, but I think both have yet to be defined.
-	Equation (11): $\nu$ can be inferred to be equal to $n \bar{\nu}$, but this does not seem to be explicitly mentioned in the paper.
-	Proposition 3.1: It is not mentioned how the log quasi-likelihood is “scaled”. Is it the same scaling as for the evidence in line 115?
-	Figure 1: The right-most plot (CI coverage for $\alpha = 0.5$) is hard to read. The results for some algorithms are barely visible or not visible at all. Perhaps leaving more margin below 0.00 and above 1.00 on the y-axis could help.
-	Lines 275-277: I’m not sure I agree with the conclusion that “quasi-Bayesian inference provides the most reliable uncertainty estimates” by looking at Figure 1. For $\alpha=0.05$, the results look promising, but for $\alpha = 0.5$, the CI coverage appears to be 100%, which I’d think is a rather poor uncertainty estimate.
-	Line 288: The first paragraph in Section 6.2 does not have numbered lines. This does not affect the review, but perhaps there is a formatting issue that the authors need to be made aware of.
-	Lines 406-409: Single-author references [30] and [31] have erroneous “et al.”.


**Time Spent Reviewing:**

five

---

> ### Author Response · Authors · 2021-08-09
> **Response**
>
> Thank you for your positive and constructive feedback, and we will incorporate them in the revision. Below we address the individual questions.
>
> ## Evaluation / discussion of computational cost
>
> Thanks for the suggestion and we agree that there should be more discussion in the main text.  Regarding a comparison of run time,
> 1. Our focus is in (i) developing a principled quasi-Bayesian approach to IV regression, based on modern ML models, and (ii) an approximate inference algorithm that can also be applied to NN models.  As discussed in the introduction, the quasi-Bayesian approach is inherently advantageous in terms of scalability, as it avoids the need to conduct Bayesian inference over generative model parameters, which, for DNN models, is both computationally prohibitive, and brings challenges to the existing approximate inference methods. These scalability issues make Bayesian nonlinear IV a relatively unexplored area. The fact that in the demand experiment, our approximate inference algorithm has the same run time as ensembled point estimation, whereas Bayesian baselines with similar flexibility (e.g. based on NN models) cannot be found in literature at all, is the best demonstration of the scalability of our method.
>
> 2. In the 1D simulation, the following table provides a rough idea on the runtime. We can see that the quasi-Bayesian methods are significantly faster than the Bayesian baseline, even though the latter is less flexible, and the runtime of the approximate inference algorithm grows more slowly as N increases.
>
> |     run time (s) \ N     |   1000    |   5000   |   20000    |
> |:------------------------:|:---------:|:--------:|:----------:|
> |     BayesIV (CPU,8C)     |    655    | N/A [\^1] |  N/A [\^1]  |
> |   QB closed-form (GPU)   |   0.08    |   2.37   | N/A [\^2] |
> | QB closed-form (CPU,16C) |    2.47    |   49.1   |    1150    |
> |   QB approx inf (GPU)    | ~10 | ~40         |    ~140  |
>
> [\^1]: does not complete after 20min
> [\^2]: out of memory
>
> **Evaluation setup.** In the above, (CPU,8C) indicates a machine with i9-9900k (8 cores, 16 threads) and 64GB RAM, and (CPU,16C) indicates another machine with two Xeon E5-2620v4 CPUs (16 cores, 32 threads) and 220GB RAM. The environment difference is because the BayesIV package only works on Windows and our code only works on Linux. Since we find the BayesIV package cannot make use of more than 4 CPU cores, the difference should not put it into disadvantage. The GPU experiments are carried out on the second machine, on a GTX 1080Ti GPU with 11GB VRAM. For all methods, the reported runtime excludes non-computational tasks such as data preparation and JIT compilation.
> For the approximate inference algorithm we we report the runtime for the best hyperparameters; runtime for other hyperparameters is typically lower due to early stopping. When applied to random feature models, a single run of the algorithm does not fully utilize the GPU, so we run 6 experiments in parallel and report the elapsed time divided by 6. We believe this is a realistic evaluation setting, since in practice all methods will require multiple runs for hyperparameter selection, and thus will benefit from parallelization. For the closed-form quasi-posterior, a single run always makes full use of the GPU, and thus we do not adopt this strategy.
>
> ## Minor comments
>
> Thanks for the comments. We will incorporate them in the revision and revise the paper for readability. Below we clarify on a few issues:
>
> ### Q2. $\bar\nu$ and $\nu$ yet to be defined
> We apologize for the confusion.  $\bar\nu$ is a regularization hyperparameter such that $n^{-1}\ll \bar\nu\ll 1$, and $\nu:=n\bar\nu$. Given this,  $\hat{{C}}\_{zz,\bar\nu} := \hat{{C}}\_{zz}+\bar\nu I$ (L74), and $\hat{C}\_{zz}$ is defined in L90.
>
> ### Q3. How quasi-likelihood is scaled
>
> The scaled log quasi-likelihood should refer to $d_n^2(\hat{E}_n f,\hat{b})$; please see our response to Reviewer hAiR, Q3.(a-d).  We apologize for the confusion.
>
> ### Q4. For $\alpha=0.5$ CI coverage appear to be 100\%
>
> The revision will make it more clear that the uncertainty estimates can be conservative. Still, please note that our credible interval always reflects the information available in data, and appears to have the right order-of-magnitude as visualized in Appendix. Also note that bootstrap coverage is unstable in this setting, so there is no perfect choice.
>
> The reason is that IV is a nonparametric inverse problem, for which uncertainty quantification is generally difficult, and it is often more desirable to err on the conservative side and obtain credible intervals with a correct order of magnitude (Knapik et al, 2011; Ghosal, 2015). Also note that, as past results in other problems (Kanagawa et al, 2018, Section 3.4; Knapik et al, 2011) suggest, GP posterior covariance usually reflects the worst-case prediction error, which will be conservative.
>
> ## References
>
> - M. Kanagawa, P. Hennig, D. Sejdinovic, and B. K. Sriperumbudur, “Gaussian processes and kernel methods: A review on connections and equivalences,” arXiv preprint arXiv:1807.02582, 2018.
> - B. T. Knapik, A. W. van der Vaart, and J. H. van Zanten, “Bayesian inverse problems with Gaussian priors.” The Annals of Statistics, vol. 39, no. 5, pp. 2626–2657, Oct. 2011.
> - S. Ghosal. Discussion:“Coverage of Bayesian credible sets”. Annals of Statistics, 85, 2015.

---

> > ### Comment · Reviewer_rhpF · 2021-08-17
> > **Acknowledgment of response**
> >
> > Thank you very much for addressing my minor concerns through this detailed response. Perhaps it could still be a good idea to briefly mention the performance improvement for the 1D simulation in the main paper, especially since the results are quite impressive.

---

> > > ### Author Response · Authors · 2021-08-19
> > > **Thanks for the suggestion**
> > >
> > > We agree it's a good idea, and will update the text to include a discussion.

---

### Official Review · Reviewer_km2d · 2021-07-14

**Rating:** 6
**Confidence:** 2

**Summary:**

The paper concerns the uncertain quantification for instrumental variable (IV) regression that it is an approach for estimating causal effect from confounded observational data and, proposes a scalable quasi-Bayesian procedure. The procedure was built upon the kernelized IV models (Muandet et al (2020)). The contribution is forming a quasi-posterior by employing a Gaussian process prior and constructing a kernel conditional expectation estimator using a randomized prior trick. Theoretical properties and simulation studies are provided.


**Ethical Concerns:**

I didn't see any point that it will raise ethical issues.

**Limitations And Societal Impact:**

Authors showed some sensitivities associated with the kernel choice and this will be one limitation.
I didn't see when their method will be poor in the text.

**Main Review:**

(a) This work is stretched out my area. With my limited search, I haven’t seen IV regressions using a kernelized IV model employing a Gaussian process prior. I feel the use of “quasi-Bayesian” is slightly abusing as only a prediction is inferenced according to the Bayes rule.

(b) I found the paper was little difficult to follow. Some notations like C^{-1}_{zz,\nu}  or S^*_z in page 3 were not defined although they were key terms in their expressions. Although authors described steps of their approach, an algorithm (or a list of steps) will be more effective, and it is easier to see what estimates implied/steps were involved.

(c) There is some unclearness about the approach.

(c-i) It is not clear how the objective function, L(f) (Eqn 4) fits in the posterior (8) and how K-zz and other parameters (kernel specific,
scale parameters) are obtained.

(c-ii) A randomize prior trick approximation will be sensitive to m and, how would you choose m and what do we expect to see when m is too small/large? How it is sensitive to z-distribution?

(d) In the simulation study, authors compared their method with existing estimators and, showed the new method was competitive. Authors also honestly showed some kernel choice sensitivity in Appendix. In the main text, this sensitivity was not shown, and I thought including this in the main paper would be very valuable to provide correct strength and weakness of their method. Moreover, some discussion on the kernel choice (if there is any existing guidelines) would be good.

(e) The paper proposed a quasi-Bayesian procedure for IV regression and showed that it can be an efficient estimator when a kernel is carefully chosen. I didn’t find any false in their approach and empirical evidence of their claim is given. Weakness will be clarification and brief description of a randomized prior trick without investigation of its properties and impacts on the resulting inference.


**Time Spent Reviewing:**

5

---

> ### Author Response · Authors · 2021-08-09
> **Response**
>
> Thank you for your constructive feedback, and we will incorporate them in the final version. We will also revise the paper to improve clarity. Below we address your individual questions.
>
> ## Q1. Use of “quasi-Bayesian” slightly abusing
>
> Please note that while similar in form, our prediction is not based on the Bayes rule: application of the Bayes rule requires sufficient knowledge of the data generating process, so that the likelihood can be specified, whereas in our setting we do not have such knowledge.
>
> To better understand the difference, we compare the data generating process assumed in standard (OLS) regression with additive error and that in IV.  For standard regression we have [\^1]
> $p(y_i,x_i,f)=p(x_i)p(f)p(y_i\mid x_i,f)=p(x_i)p(f)p_{err}(y_i-f(x_i))$, where $p_{err}$ is the distribution of the unconfounded residual (e.g. normal); as $p(x_i)$ is independent of $f$, we can use the structural knowledge of $p_{err}(y_i-f(x_i))$ to define the likelihood and conduct Bayesian inference.  For IV, however, we only have
> $p(y_i,x_i,z_i,f) = p(f)p(z_i,x_i)p(u_i=y_i-f(x_i)\mid x_i,z_i)$. Due to confounding, we cannot drop the conditioning of $u_i$ on $x_i$, and thus we cannot compute the likelihood.  To apply the Bayes rule, we would have to introduce a *Bayesian* conditional generative model for the distribution $p(u_i\mid x_i,z_i)$.  This would require further structural knowledge of the distribution (e.g. additivity, parameterizable by neural nets), and, at the same time, make approximate inference prohibitively expensive.  The quasi-Bayesian approach does not have these drawbacks; instead, it defines a quasi-likelihood with generalized moment conditions.
>
> (The term “quasi-Bayesian” is introduced by previous work, e.g., [17-19].)
>
> [\^1]: for notational simplicity we assume a single data point, and assume the existence of some “prior density” for $f$.  The formulation can be easily made rigorous.
>
> ## Q2. Clarity of writing
>
> We apologize for the confusion, and will revise the paper to fix undefined notations and improve general readability.  For the specific issues you mentioned:
>
>
> 1. $\hat{{C}}\_{zz,\bar\nu} := \hat{{C}}\_{zz}+\bar\nu I$ (L74), and $\hat{C}\_{zz}$ is defined in L90.  $\bar\nu\in\mathbb{R}$ was undefined; it is a regularization hyperparameter.
> 2. $S_z$ is defined in L93 and $S_z^*$ refer to its adjoint (L73).
> 3. The inference algorithm is described in Algorithm 1, Appendix C.4.  We will add a link in the main text.
>
> ## Q3-i. How $\mathcal{L}(f)$ as in Eq.(4) fits in Eq.(8), and how $K_{zz}$ and other scale parameters are obtained
>
> 1. The MAP objective $\mathcal{L}(f)$ is defined by Eq.(3).  The equivalence between Eq.(3) and (4) can be found in [21, 25]; alternatively, we can repeat the proof of Proposition C.1 up to L737, with $f_0,g_0$ replaced by 0.
> 2. Eq.(3) is transformed into the equivalent form Eq.(5) using the expressions for the empirical covariance operators in L103-104 (also see below for possible clarifications).  As discussed in Section 3, Eq.(5) motivates us to define the quasi-posterior as Eq.(6), and Eq.(6) coincides with the true posterior of the *fictitious* data generating process immediately below Eq.(7).  Applying the Gaussian conditioning formula (Appendix A.2, Rasmussen and Williams, 2006) to Eq.(7), we arrive at Eq.(8-10).
> 3. $\lambda$ appears in the definition of the quasi-posterior as it also appears in Eq.(3-5); it directly controls the regularization strength on $f$.  $\nu=n\bar\nu$ enters the quasi-posterior through the matrix $L$ (L105); it regularizes the conditional expectation estimation.  The use of two regularization hyperparameters also appear in past work on kernelized IV estimators [7, 21].
>    $K_{zz}$ appears in the quasi-posterior (8-11) when we apply Woodbury identity to the definition of matrix $L$, which led to Eq.(11) (see below for details).
>
>
> We further clarify on two points in the derivation which may have caused confusion:
>
>
> 1. The identities $S_z S_z^* = K_{zz}, S_z^* S_z = n\hat{C}\_{zz}$:
>     Recall that the *sampling operator* $S_z: \mathcal I \to \mathbb R^n$ is defined as $S_z g := (g(z_1), g(z_2), \cdots, g(z_n))$. Its adjoint $S\_z^\*: \mathbb R^n \to \mathcal I$ fulfills the condition:
> $\langle S\_z f, c\rangle_{\mathbb R^n} = \langle f, S^\*\_z c\rangle\_{\mathcal I}$ for all $f \in \mathcal I, c \in \mathbb R^n$.
> The LHS can be rewritten as $\sum_{i=1}^n f(z_i)c_i = \sum_{i=1}^n \langle f, k(z_i, \cdot)\rangle_{\mathcal I} c_i = \left  \langle f, \sum_{i=1}^nk(z_i, \cdot) c_i\right\rangle_{\mathcal I}$. By comparing with the above condition, we can find $S_z^* c = \sum_{i=1}^nk(z_i, \cdot) c_i$.
>
>     Observe that $S_zS_z^*$ is a map from $\mathbb R^n$ to $\mathbb R^n$, which can be identified as a matrix $\mathbb R^{n \times n}$. Also observe that
> $S\_z S\_z^* c = \sum_{i=1}^n S_zk(z_i, \cdot) c_i$ and
> $S_z k(z_i, \cdot) = (k(z_i, z_1), \cdots, k(z_i, z_n))$ is the $i$-th column of the Gram matrix $K_{zz}$. Thus, $S_z S_z^* c$ can be regarded as $K_{zz}c$ and hence $S_z S_z^* = K_{zz}$.
>
>     Similarly, $S\_z^*S\_z$ is a map from $\mathcal I$ to itself, and can be written as
> $S^\*\_zS\_z f = S^\*\_z (f(z_1), f(z_2), \cdots, f(z_n)) = \sum_{i=1}^n k(z_i, \cdot) f(z_i) = n\hat C_{zz} f$.
>
> 2. how Eq.(11) is obtained:
>     $L=\frac{1}{n}S_z \hat{C}\_{zz,\bar\nu}^{-1}S_z^*=S_z(S_z^* S_z+n\bar\nu I)^{-1}S_z^*=(S_z S_z^*+n\bar\nu I)^{-1}S_z S_z^*=(K_{zz}+n\bar\nu I)^{-1}K_{zz}$. In the above, the first equality is by the definition of $L$ on L105, the second by $\hat{C}\_{zz}=n^{-1}S\_z^\*S\_z$, the third by an application of the Woodbury identity (Eq.(52)), and the last equality is by $S\_zS\_z^\*=K\_{zz}$.
>
> ## Q3-ii. Sensitivity and choice of $m$
>
> ### (pt 1). what if m is too large / small
>
>
> Under a fixed computational budget, the convergence analysis suggests a trade-off between approximation error and optimization error: the former dominates when $m$ is too small, and the latter dominates otherwise.  In practice we find a wide range of $m$ to be applicable; see (pt 3) below.
>
> ### (pt 2). how to choose m
>
> In practice we view $m$ as a hyperparameter and employ the hyperparameter selection procedure described in Appendix D.1.
> Past work on random feature ridge regression (Rudi and Rosasco, 2016) provides asymptotic upper bound for $m$.  Given the role of the kernels in our setting ($k_x$ to approximate $f^\dagger$, $k_z$ to approximate $Ef$ for $f\sim\mathcal{GP}(0,k_x)$) we expect similar results, although a proof will require additional effort.
>
> ### (pt 3). experiments for the selection of $m$
>
> As our approximate inference experiments mainly focused on NN-based models (where we find the hyperparameter selection procedure to be robust), here we provide additional experiments for choosing m in random feature approximation. Specifically, we revisit the experiment used to generate Figure 6(d) (1d simulation, abs design, approximate RBF kernel, $N=1000,\alpha=0.5$), this time varying $m\in\\{50, 100, 200, 400, 800, 1600\\}$. For each m we choose other hyperparameters using the validation statistics, and report best the validation statistics, test MSE, CI coverage rate and average CI length. We also report results for the exact kernel for comparison. The result is as follows:
>
> | m         | 50     | 100    | 200    | 400    | 800    |   1600 | (exact) |
> | --------- | ------ | ------ | ------ | ------ | ------ | ------ | ----- |
> | val.stats ($\times 10^{-3}$) | 4.1 (3.3) | 3.2 (2.3) | 3.0 (1.9) | 2.9 (2.1) | 2.5 (2.0) | 3.2 (1.9) | 3.1 (1.3) |
> | MSE       | .027 (.010)  | .022 (.003) | .025 (.009) | .025 (.007) | .031 (.018) |  .041 (.031) | .031 (.012) |
> | CI Cvg.   | 0.97 (0.04) | 1.00 (0.01) | 0.99 (0.02) | 0.96 (0.03)  | 0.90 (0.07)  |  0.89 (0.09) | 0.89 (0.13) |
> | CI Len.   | 0.47 (0.08) | 0.41 (0.17)  | 0.36 (0.02)  | 0.35 (0.03)  | 0.31 (0.07)  | 0.32 (0.09) | 0.31 (0.07) |
>
> As we can see, for $m> 100$, the choice of $m$ does not have a significant impact, and for all values of $m$ the validation statistics correlates with test performance, although randomness in the repeatedly generated dataset has a larger impact.  We note that, in line with (pt 1), optimization is somewhat more unstable when $m=1600$, but it is mitigated by the search over a grid of other hyperparameters (so optimization only has to succeed once in a region of relatively good hyperparameters); alternatively we can simply repeat the optimization process for several times and choose the best run according to the validation statistics.
>
> ### (pt 4). sensitivity to the data distribution
>
> To study sensitivity to the data distribution, we further repeat the above experiment using the sine design (thus changing $p(x,y)$), and vary $\alpha\in\\{0.05,0.5\\}$ for both designs (changing $p(z,x)$). The experiments lead to the same findings.
>
>
> ## Q4. Sensitivity to the choice of kernels
>
> Thanks for the suggestions. We will make this more clear in the main text.
>
>
> ## References
> - C. E. Rasmussen, and C. K. Williams, “Gaussian Processes for Machine Learning,” Cambridge University Press, 2006.
> - Rudi, A. and Rosasco, L., (2016). Generalization properties of learning with random features. *arXiv preprint arXiv:1602.04474*.

---

> > ### Comment · Reviewer_km2d · 2021-08-14
> > **comment**
> >
> > Thanks for your response. I feel my comments have been adequately addressed.

---

### Official Review · Reviewer_hAiR · 2021-07-15

**Rating:** 7
**Confidence:** 3

**Summary:**

The paper studies a quasi-Bayesian approach to instrumental variable (IV) regression, based on using an empirical estimate of the ‘likelihood’ to obtain a Gibbs posterior. Such an approach has been studied in earlier literature due to the difficulty of incorporating moment conditions in IV via a likelihood. The paper pursues a Gaussian process approach and proves some theory, namely that the resulting posterior puts most of its mass on functions (almost) satisfying the moment constraint. The paper provides a computation approach based on the randomized prior trick and illustrates the method on various simulated datasets.

**Limitations And Societal Impact:**

Yes

**Main Review:**

I think the paper is interesting and well written in parts (though a bit dense at times), and features a nice mixture of theory and methodology. The approach of using a quasi-likelihood is not new, and the paper’s main contribution in this direction is to develop a choice of quasi-likelihood for which more efficient computation can be performed. This is a reasonably novel idea, cleverly combining several different existing ideas in the instrumental variable (IV) setting. I work on Bayesian methods rather than IV methods, so it was not entirely clear to me what was previously known here. I also comment more on the theory since I am more familiar with this.

A main contribution is to prove some theory, namely (C1) and (C2) in the paper. (C1) was fine, but (C2), was not clear to me either mathematically or in words and needs to be looked at. The proofs follow from tools in Bayesian nonparametrics as have been applied in this setting by Kato (Annals of Statistics 2013). Saying that, his proofs are very technical and applying them to new settings requires effort, so I think the paper has a suitable technical level for this venue.

The method is illustrated in simulations where it shows promising empirical performance. The simulations are broadly fine, but do have a few issues I think could be improved.

Overall, I think the paper is interesting and relevant. If the theoretical results can be satisfactorily cleared up (especially Proposition 3.1) and better explained, this will make a solid contribution.

Main comments:
1.	Theorem 3.1 and its discussion need to be clarified. What does $\mathbb{E}^2$ mean? If I understand correctly, this result says that the posterior concentrates on functions for which the moment restriction (1) ‘almost holds’. If you were Bayesian, you would try to put a prior that directly enforces this condition rather than hope your posterior concentrates on it. Why can such a restriction not be enforced with quasi-posterior probability one?

2.	Theorem 3.1: what values of $\tau_n$ are provided by your theory for concrete examples? E.g. Matern prior and Holder/Sobolev smooth truth. The results are currently quite abstract and not interpretable.
3.	Proposition 3.1:

(a)	Why is $f \sim \Pi$ from the prior? Where is this used? The conclusion doesn’t involve the prior and the prior is only specified as a GP, so does this result hold for any function $f$ drawn from any GP?

(b)	$\hat{E}_n$ is an operator taking $x$ valued functions to $z$ valued functions, so it’s not at all clear what the LHS means here.

(c)	A general notational improvement throughout the paper would be to clarify what you are taking the expectation over. e.g. $\mathbb{E}_X$ for taking the expectation of $X$, etc.

(d)	Make precise under what probability distribution $\to^p$ is.

(e)	This looks like a kind of ‘uniform law of large numbers’ since the $\hat{E}_n$ is random and depends on the same random variables as the sum. Empirical process theory tells us that such a result holds if the function class is Glivenko-Cantelli but not generally otherwise. However, I don’t see such a condition. Can you please clarify (maybe it’s implicit from $f \sim \Pi$)?

(f)	Remark 3.3 and the interpretation of Proposition 3.1 don’t make sense to me. Also, if you want to relate this to the log-likelihood, then wouldn’t it make sense to study the limit uniformly over $f$ rather than pointwise? Please carefully explain what is going on.

4.	Proposition 4.2:  Can you discuss this result a bit more. If I understand it correctly, this say the algorithm won’t get stuck at a local minimum (likely due to the stochastic part), but it doesn’t give any quantitative bounds on convergence, correct?
5.	Simulations:

(a)	I think it would be clearer to present your data in table form rather than as boxplots (one picture is fine).

(b)	What are you plotting credible intervals of? The goal is to estimate the whole function $f$. Are you doing it at a point?

(c)	You should provide the average credible interval lengths, otherwise the coverage is not informative.

(d)	10 repetitions of the data is small and you should do more if possible. If this is computationally prohibitive, please say so and give some indication of the run time.


Minor points:
-	p3 Maybe define the tensor product $\otimes$
-	p3 Could you please explain the equation on l92 a bit better
-	Assumption 3.2 One typically needs the assumption $m\tau_m^2 \to \infty$ for these kind of results. Has this been forgotten?
-	Assumption 3.2: mention that this is a standard condition in the GP literature. You should really cite van der Vaart & van Zanten (Annals of Statistics 2008) here since they first applied this theory to GP’s.
-	Theorem 3.1 I guess $\tau_{\sqrt{n}}$ should be $\tau_n$ everywhere (it’s indexed by an integer)?
- Theorem 3.1 is not a contraction rate, which involves the posterior putting most of its mass about the ‘true’ function, which is not what you are studying here.
- THeorem 3.1: specify what assumptions you have put on the ‘true’ $f$ generating the data (taken over the expectation $\mathbb{E}_{D_n}$. Do you assume the condition (1) holds in reality?
- Proposition 4.2: $S_m$, $\mu_m$ should be $S$, $\mu$?
-	P19 What is $err_{n,f}$?

############################################

Note: scored changed from 6 to 7 after author response.


**Time Spent Reviewing:**

5

---

> ### Author Response · Authors · 2021-08-09
> **Response 1/2**
>
> Thank you for the appreciation of our contributions and constructive comments. We apologize for the clarity issues, espeically around Proposition 3.1, and will incorporate your feedback in the final version. Below we clarify on your questions.
>
> ## Clarification on Theorem 3.1
>
> ### (Q1, part 1) What does $\mathbb E^2$ mean
> $\mathbb{E}^2(f(\mathbf{x})-\mathbf{y}\mid\mathbf{z})$ means $\big(\mathbb{E}(f(\mathbf{x})-\mathbf{y}\mid\mathbf{z})\big)^2$, the square of the conditional expectation random variable.
>
> ### (Q1, part 2) Prior should directly enforce Eq.(1)
>
> 1. As the moment condition Eq.(1) needs to be estimated from data, it is not possible to choose a prior that places all its mass on functions satisfying such conditions, without using the data.
> 2. Still, it is true that an ideal prior should place as much mass on such functions as possible.  In our analysis the decay rate of $\{\tau_m\}$ (see Assumption 3.2) quantifies this property: by Eq.(21) in the appendix and the continuity of conditional expectation w.r.t. the sup norm, we have $\Pi(\{f : \left|\mathbb{E}(f(\mathbf{x}) - \mathbf{y}\mid\mathbf{z})\right|\le 2\tau_m\text{ a.s. }[p(z)]\})\ge \exp\left(-m\tau_m^2\right)$; in words, when $\tau_m$ has a fast decay rate, the prior places higher mass on functions where Eq.(1) holds approximately up to a given error.  And as discussed in Remark A.2, the decay rate is usually the fastest when the regularity of the GP prior matches the regularity of the true function $f_0$, so in summary, a good prior places more mass on functions where (1) approximately holds.
>
> ### (Q1, part 3) Why such a restriction is not enforced with posterior probability one
>
> We're not sure about the exact meaning of this question. If you are referring to the posterior [\^1], then Theorem 3.1 already shows that, as $n$ increases, the posterior assigns probability mass approaching 1 to the set of functions where the moment conditions are enforced with vanishing error. If there was a typo and you were referring to the prior probability, see our response to (Q1, part 2) above.
>
> [\^1]: we omit the prefix “quasi-” below for conciseness.
>
> ### Q2. Values of $\tau_n$ for concrete examples
>
> Lemma A.1 and A.3 provide values of $\tau_n$ under varying combinations of prior and ground truth smoothness assumptions.  When $\mathcal{X}=[0,1]^d$ and the ground truth is in the Sobolev and Holder spaces of order $\beta$, i.e., $H^\beta(\mathcal X) \cap C^\beta(\mathcal X)$, the optimal Matern prior leads to $\tau_n^2 \asymp n^{-\frac{2\beta}{2\beta+d}}$ (Remark A.2).
>
> ## Clarification on Proposition 3.1 and (C2)
>
> ### (Q3.a-d) Precise statement of the proposition
>
> The statement of the proposition should be as follows:
>
> **Proposition 3.1.** The scaled log quasi-likelihood estimate $\ell_n(f) := d_n^2(\hat{E}_n f - \hat{b})$ satisfies
>
> $$
> \\Pi\\left(\\left\\{f : \\forall\\delta>0,\\lim_{n\\rightarrow\\infty} \\mathbb{P}\_{\\mathcal{D}^{(n)}} \\left(\\left|\\ell_n(f) - \\mathbb{E}_{\\mathbf{z}\\sim p(z)}(\\mathbb{E}(f(\\mathbf{x})-\\mathbf{y}\\mid\\mathbf{z})^2)\\right|>\\delta\\right)=0\\right\\}\\right)=1.
> $$
>
> This is what we proved in Appendix B.3 [\^2].
> In the above, $d_n$ and $\hat{b}$ are defined in L87-L92.  And since the condition on $f$ is the definition of convergence in ($\mathbb{P}_{\mathcal{D}^{(n)}}$-)probability, the original text in the proposition is correct. (The original formula erroneously dropped the RKHS regularizer in the definition of $d_n$; we apologize for the confusion.)
>
> We hope the above clarification addresses your questions (a-d). For (a), also note that the proposition can only be stated w.r.t.~the GP prior $\Pi$, as opposed to arbitrary GP priors, because our assumptions (in particular, 3.1 and 3.2) are made around this GP.
>
> [\^2]: We proved convergence of $\sqrt{\ell_n(f)}$.  Convergence of $\ell_n(f)$ follows from the continuous mapping theorem for convergence in probability.
>
> ### (Q3.e) Where is the Glivenko-Cantelli class condition
>
> You are right. This is because we only consider a subset with GP prior probability 1.  Intuitively, as shown in Eq.(18) and Eq.(20) in appendix, this subset can be approximated by $\bar{J}_m\mathcal{H}_1$ which is a G-C class; and the approximation error $\tau_m$ can be made arbitrarily small.  The proof of the proposition characterizes this intuition.
>
> ### (Q3.f, part 1) Uniform bound over $f$
>
> 1. Lemma B.2 provides a similar statement to Proposition 3.1 that holds uniformly, for a large proportion of functions and a fixed estimation error. Here we provide its following corollary for convenience:
>
> **Corollary.** For any $c>0,M\in\mathbb N$, let $\Theta_{\infty,M,c} := \{f: \forall m\ge M, c \cdot f\in\Theta_m\}$. Then under the hyperparameter choices in Theorem 3.1, we have, for any $\epsilon>0$,
> $$
> \mathbb{P}\_{\mathcal D^{(n)}}\left(\forall f\in\Theta_{\infty,M,c}, \left |\sqrt{\ell_n(f)} - \sqrt{\ell_{gt}(f)}\right| \le \epsilon \sqrt{\ell_{gt}(f)} + r_{n}\right) \rightarrow 1,
> $$
>
> where $\Theta_m$ is defined in Theorem A.1, $\ell_{gt}(f) = \mathbb E(\mathbb{E}(f(\mathbf x)-\mathbf y\mid\mathbf z)^2)$, $r_n = C \gamma_n \tau_{[\sqrt{n}]}$ where the constant $C$ depends on $c$, and $\gamma_n$ is a sequence with arbitrarily slow growth.
>
> 2. It is intuitive that the above corollary should apply to "almost all" functions in the GP prior (although for varying $c$ and $M$), since we have $\Pi(\Theta_m)\rightarrow 1$ (Eq.(20)); and by Assumption 3.2 and L562 we also have $f^\dagger\in \Theta_{\infty,1,c}$ for some $c$.  Proposition 3.1 formalizes this intuition by showing that a similar result holds for functions in a set with prior probability 1.
>
> **Proof of the Corollary.** It suffices to consider the case $c=1$. By Lemma B.2, we can choose $r\in(0,1/2)$ so that for any $n> M^2$, on the event $B_n(r,L)$ we have $|\sqrt{\ell_n(f)}-\sqrt{\ell_{gt}(f)}|\le \epsilon \sqrt{\ell_{gt}}(f) + r_{n,[\sqrt{n}]}^{(1)}$ uniformly for all $f\in\Theta_{\infty,M,c}$. For the choice of hyperparameters and $L$ in Theorem 3.1, we have shown that $r_{n,[\sqrt{n}]}^{(1)}\precsim \sqrt{\gamma_n}\tau_{[\sqrt{n}]}$ (L662, L666), and   $\mathbb{P}\_{\mathcal{D}^{(n)}} B_{n}(1/3,L)\rightarrow 1$ (L670). Now $B_n(r,L)=B_n(1/3,L)\cap A_n(r)$ (L616) also has $\mathbb{P}\_{\mathcal D^{(n)}}$ probability converges to 1 by Lemma B.3, which concludes the proof.
>
> ### (Q3.f, part 2) Interpretation of the proposition, connection to (C2)
>
> 1. The proposition allows us to compare the asymptotic behavior of estimated log-likelihood on a finite number of functions, over which the convergence result holds uniformly.  For functions with a similar level of smoothness (e.g., similar asymptotics of shifted small ball probability), comparing likelihood is similar to comparing posterior density for finite-dimensional models, as the (quasi-)likelihood is the Radon-Nikodym density w.r.t. the prior measure. Thus it provides information on what kind of functions are favored by the posterior.
>
> 2. We clarify on the example given in Remark 3.3:
>     consider three functions $f_1,f_2,f_3$ in the probability-1 subset defined above, such that $f_1$ and $f_2$ satisfies Eq.(1) (i.e., a “valid solution”), and $\mathbb{E}_{p(z)}(\mathbb{E}(f_3(\mathbf{x})-\mathbf{y}\mid\mathbf{z})^2)=a>0$ (an "invalid solution").  The likelihood is proportional to $\exp(-\lambda^{-1}n\ell_n(f))$, and Proposition 3.1 shows that the negative log quasi-likelihood for $f_1,f_2$ satisfies $\lambda^{-1}n\ell_n(f_1), \lambda^{-1}n\ell_n(f_2)=o(\lambda^{-1}n)$, whereas $\lambda^{-1}n\ell_n(f_3) \asymp a\cdot \lambda^{-1}n$.  Thus *as* $n$ *increases, the quasi-posterior density (w.r.t. prior) of the “invalid solution”* $f_3$ *quickly become distinguishable from that of* $f_1$ *or* $f_2$, *whereas distinguishing between* $f_1$ *and* $f_2$ *is more difficult*.
>
> 3. The above example provides a preliminary characterization of (C2), if we consider a function $f$ to be “excluded” from the posterior when its likelihood is smaller than a certain threshold.  By inspecting the proof, or from the above corollary, we can see the threshold can be chosen as $\exp(-M_n\lambda^{-1}n\tau_{[\sqrt{n}]}^2)$, where $M_n$ is any slowly increasing sequence.
>
> ### (Q3.f, part 3) Remark 3.3
>
> We have clarified on the first half of Remark 3.3 above. Regarding the remaining part, as the proposition shows that the quasi-likelihood estimate is asymptotically equivalent to the violation of the GMM conditions Eq.(1), and in light of the interpretation of quasi-likelihood in (Q3.f, part 2, bullet point (1)), we expect that the quasi-posterior should have a larger spread when more functions in the prior have a smaller violation of Eq.(1), which happens when the instrument is weak.  The behavior is not trivial: bootstrap 2SLS may behave differently, in light of Remark 3.4 and past work on bootstrapped linear 2SLS [36-38].  Nonetheless, it is true that this discussion is informal and only aimed at providing intuition, as we will further clarify in the revision.

---

> > ### Author Response · Authors · 2021-08-09
> > **Response 2/2**
> >
> >
> > ## Other Main Comments
> >
> > ### Q4. Is it correct that Proposition 4.2 doesn’t give quantative bounds
> >
> > This is correct; we state a qualitative result for simplicity.
> >
> > ### Q5. Simulations
> >
> > 1. The revision will provide tables for the full results.  We are plotting average CI coverage (i.e. coverage rate averaged over test inputs).
> > 2. For all but the image experiment, we have run 10 additional replications, so the total number of replications is now 20.  There is no qualitative change to our reported results.  A representative subset of results is presented in the table below, where we report the median of the average CI coverage as well as the 25% and 75% percentile.  We only include the bootstrapped KIV baseline as it is  most competitive; results of all methods and datasets will be added to the revision.
> >
> > | avg. CI cvg.  | sin (RBF), $$n=200,\alpha=0.5$$ | sin (RBF), $$n=1000,\alpha=0.5$$ | sin (RBF), $$n=1000,\alpha=0.05$$ | abs (Matern 3/2), $$n=1000,\alpha=0.5$$ | demand, $$n=1000$$   |
> > | ------------- | ------------------------------- | -------------------------------- | --------------------------------- | --------------------------------------- | -------------------- |
> > | KIV-Bootstrap | 1.000 (0.762, 1.000)            | 0.947 (0.668, 1.000)             | 0.615 (0.502, 0.670)              | 0.988 (0.932, 1.000)                    | 0.859 (0.825, 0.877) |
> > | QB (ours)     | 1.000 (1.000, 1.000)            | 1.000 (1.000, 1.000)             | 0.857 (0.782, 1.000)              | 1.000 (0.985, 1.000)                    | 0.933 (0.926, 0.952) |
> >
> > 3. We have calculated the average CI lengths in all experiments.  A representative subset of results is presented in the table below, where we report the mean and standard deviation, and for now you may refer to Appendix D.4 for a qualitative understanding about the other experiments.  All results will be added to the revision.
> >
> > | avg. CI width | sin (RBF), $$n=200,\alpha=0.5$$ | sin (RBF), $$n=1000,\alpha=0.5$$ | sin (RBF), $$n=1000,\alpha=0.05$$ | abs (Matern 3/2), $$n=1000,\alpha=0.5$$ | demand, $$n=1000$$ |
> > | ------------- | ------------------------------- | -------------------------------- | --------------------------------- | --------------------------------------- | ------------------ |
> > | KIV-Bootstrap | $$0.25\pm 0.06$$                | $$0.18\pm 0.04$$                 | $$0.33\pm 0.10$$                  | $$0.36\pm 0.10$$                        |  $$0.14\pm 0.01$$             |
> > | QB (ours)     | $$0.49\pm 0.10$$                | $$0.31\pm 0.07$$                 | $$0.57\pm 0.07$$                  | $$0.48\pm 0.06$$                        |  $$0.26\pm 0.04$$ |
> >
> > 4. The run time for the neural network-based experiments can be found in L935-944; for the closed-form kernel methods please refer to our response to Reviewer rhpF.
> >
> >
> > ## Minor Comments
> >
> > We agree with most of the comments and will incorporate them in revision.  Below we clarify on a few points.
> >
> > ### Q6. Explaination of the equation on L92
> >
> > $\langle g, n\hat{C}\_{zz} g\rangle_{\mathcal I} = \langle g, S_z^* S_z g\rangle_{\mathcal I} = (S_z g)^\top (S_z g) = \sum_{i=1}^n g(z_i)^2$. For the first equality please refer to our response to reviewer km2d, Q3-i.
> >
> >
> > ### Q7. $\tau_{\sqrt{n}}$ should be $\tau_n$
> >
> > The correct statement should be $\tau_m$ with $m=[\sqrt{n}]$ (rounded to the nearest integer).  The rate is slower than $\tau_n$ as we need to account for estimation error of the likelihood; similar issues exist in [7, 21].  Although as we have stated, our analysis is crude and a better rate should be possible (e.g., with further assumptions on the regularity of $k_z$, and on relations between the GP prior and the conditional expectation operator [17, 18].)
> >
> > ### Q8. Theorem 3.1 is not a contraction rate
> >
> > 1. Note that by Eq.(1), $\mathbb{E}(f(\mathbf{x})-\mathbf{y}\mid\mathbf{z}) = \mathbb{E}(f(\mathbf{x})-f^\dagger(\mathbf{x})\mid\mathbf{z})$ a.s., and thus Theorem 3.1 is a contraction rate in the semi-norm $d(f,f'):=\mathbb{E}_{p(z)}\big(\mathbb{E}(f(\mathbf{x})-f'(\mathbf{x})\mid\mathbf{z})^2\big)$.  For the kernelized IV method we build upon, theoretical guarantees have only been developed around this seminorm.
> > 2. We agree that contraction rate in $L_2$ or sup norm would be more intuitive, but due to the ill-posed nature of the IV problem, this will require additional assumptions between the GP prior and the conditional expectation operator.  We intend to leave such refined analyses as future work.
> >
> > ### Q9. Conditions on $f^\dagger$ in Theorem 3.1, whether they hold in reality
> >
> > We assume the existence of one $f^\dagger$ which satisfies Eq.(1), and can be well approximated by our prior (Assumption 3.2).  (1) is the standard assumption in the nonparametric IV literature, and a more intuitive sufficient condition is that the instrument $\mathbf{z}$ is independent of the confounded residual $\mathbf{u}$, which can be verified using domain knowledge in many applications (e.g. clinical trial non-compliance).  However, there are also many cases where the chosen “instrument” may not exactly satisfy (1), and there is a vast literature on detecting and utilizing of such invalid instruments, see the references below.
> >
> > ### Q10. What is $err_{n,f}$
> >
> > We apologize for the omission; it is the set defined in Eq.(12).
> >
> > ## References
> >
> > * Hahn, Jinyong, and Jerry Hausman. "Estimation with valid and invalid instruments." *Annales d'Economie et de Statistique* (2005): 25-57.
> > * Murray, Michael P. "Avoiding invalid instruments and coping with weak instruments." *Journal of economic Perspectives* 20.4 (2006): 111-132.
> > * Hartford, Jason S., et al. "Valid causal inference with (some) invalid instruments." *International Conference on Machine Learning*. PMLR, 2021.

---

> > > ### Comment · Reviewer_hAiR · 2021-08-12
> > > **Response to authors**
> > >
> > > Thank you for your detailed response. I am correspondingly raising my score to 7 in view of these points being included in the final version. I have only one minor follow up point:
> > >
> > > -	Proposition 3.1: these “Prior-a.s.” statements can be deceptively weak for infinite-dimensional priors like GPs. For example Doob’s consistency theorem states that a prior is consistent for $\Pi$-almost every $f$, but one can construct priors which are inconsistent for ‘most’ parameters in a natural topological sense. It’s worth putting a sentence after Prop 3.1 that one should be careful about null sets for infinite-dimensional priors since these can be very large in other ways (e.g. topologically) and contain many relevant functions/parameters. This has historically caused some confusion in the Bayesian literature. For discussion on this, see e.g.
> > >
> > > Diaconis and Freedman (1986). On the consistency of Bayes estimates. Annals of Statistics.

---

> > > > ### Author Response · Authors · 2021-08-13
> > > > **Thanks for the prompt response**
> > > >
> > > > Thanks for the prompt response.  We will include the points in our original response in the revision, as well as a discussion on the null set in Proposition 3.1.

---

### Decision · Program_Chairs · 2021-09-27

**Decision:**

Accept (Poster)

**Comment:**

The paper investigates the problem of instrumental variable (IV) regression and proposes a novel quasi-Bayesian approach to it. The IV regression is an important problem in economics, social science, and epidemiology which has recently gained traction in the machine learning community as well. However, as the authors mentioned, we still lack satisfactory way of qualifying the uncertainty in this setting and Bayesian modelling is non-trivial in this case because of the inability to specify full likelihood. Due to the non-standard setting of this work, it thus provides an important step towards a full probabilistic treatment of IV regression and will certainly inspire future works in the Bayesian ML community.

There is a consensus among the expert reviewers that the paper makes a sold technical and methodological contribution. All reviewers also feel that their concerns have been addressed adequately. Upon reading the paper, I do agree with Reviewer hAiR and Reviewer km2d that the paper is quite difficult for the non-expert readers to follow, especially because of the technical nature of this work. I encourage the authors to improve the writing and to make it more accessible to the broader audience. This is a minor point, but can increase the impact of this paper.

I thus recommend this paper for publication at NeurIPS2021.

**Remark**: I feel that the related work section can also be improved. Here are some references that might helpful in improving this section:

- [1] R. Zhang, M. Imaizumi, B. Schölkopf, and K. Muandet, “Maximum Moment Restriction for Instrumental Variable Regression,” arXiv:2010.07684 [cs], Oct. 2020, arXiv: 2010.07684.

- [2] K. Muandet, W. Jitkrittum, J. Kubler, "Kernel Conditional Moment Test via Maximum Moment Restriction", UAI2020.

- [3] A. Mastouri, Y. Zhu, L. Gultchin, A. Korba, R. Silva, M. J. Kusner, A. Gretton, K. Muandet: "Proximal Causal Learning with Kernels: Two-Stage Estimation and Moment Restriction", ICML2021.

- [4] R. Singh, M. Sahani, and A. Gretton, “Kernel Instrumental Variable Regression”, NeurIPS2019.

For relevant discussions, see, e.g., related work section in [4], Section 6 in [1] and Section 5 & Appendix B1 in [2]. Section 3.3 in [3] is definitely relevant to the second paragraph L243-L250 of this submission.